# The Benefits of Implicit Regularization from SGD in Least Squares Problems

**Difan Zou**[*]
University of California, Los Angeles
knowzou@cs.ucla.edu

**Jingfeng Wu**[*]
Johns Hopkins University
uuujf@jhu.edu

**Vladimir Braverman**
Johns Hopkins University
vova@cs.jhu.edu

**Quanquan Gu**
University of California, Los Angeles
qgu@cs.ucla.edu

**Dean P. Foster**
Amazon
dean@foster.net

**Sham M. Kakade**
University of Washington & Microsoft Research
sham@cs.washington.edu

## Abstract

Stochastic gradient descent (SGD) exhibits strong algorithmic regularization effects in practice, which has been hypothesized to play an important role in the generalization of modern machine learning approaches. In this work, we seek to understand these issues in the simpler setting of linear regression (including both underparameterized and overparameterized regimes), where our goal is to make sharp instance-based comparisons of the implicit regularization afforded by (unregularized) average SGD with the explicit regularization of ridge regression. For a broad class of least squares problem instances (that are natural in high-dimensional settings), we show: (1) for every problem instance and for every ridge parameter, (unregularized) SGD, when provided with *logarithmically* more samples than that provided to the ridge algorithm, generalizes no worse than the ridge solution (provided SGD uses a tuned constant stepsize); (2) conversely, there exist instances (in this wide problem class) where optimally-tuned ridge regression requires *quadratically* more samples than SGD in order to have the same generalization performance. Taken together, our results show that, up to the logarithmic factors, the generalization performance of SGD is always no worse than that of ridge regression in a wide range of overparameterized problems, and, in fact, could be much better for some problem instances. More generally, our results show how algorithmic regularization has important consequences even in simpler (overparameterized) convex settings.

## 1 Introduction

Deep neural networks often exhibit powerful generalization in numerous machine learning applications, despite being *overparameterized*. It has been conjectured that the optimization algorithm itself, e.g., *stochastic gradient descent* (SGD), implicitly regularizes such overparameterized models [29]; here, (unregularized) overparameterized models could admit numerous global and local minima (many of which generalize poorly [29, 21]), yet SGD tends to find solutions that generalize well, even in the absence of explicit regularizers [22, 29, 19]. This regularizing effect due to the choice of the optimization algorithm is often referred to as *implicit regularization* [22].

---

[*]Equal Contribution

35th Conference on Neural Information Processing Systems (NeurIPS 2021).

Before moving to the non-convex regime, we may hope to start by understanding this effect in the (overparameterized) convex regime. At least for linear models, there is a growing body of evidence suggesting that the implicit regularization of SGD is closely related to an explicit, $\ell_2$-type of (ridge) regularization [25]. For example, (multi-pass) SGD for linear regression converges to the *minimum-norm interpolator*, which corresponds to the limit of the ridge solution with a vanishing penalty [29, 14]. Tangential evidence for this also comes from examining gradient descent, where a continuous time (gradient flow) analysis shows how the optimization path of gradient descent is (pointwise) closely connected to an explicit, $\ell_2$-regularization [24, 1]. Similar results [2] have been further extended to SGD, where a (early-stopped) continuous-time SGD is demonstrated to perform similarly to ridge regression with certain regularization parameters.

However, as of yet, a precise comparison between the implicit regularization afforded by SGD and the explicit regularization of ridge regression (in terms of the *generalization performance*) is still lacking, especially when the hyperparameters (e.g., stepsize for SGD and regularization parameter for ridge regression) are allowed to be tuned. This motivates the central question in this work:

> *How does the generalization performance of SGD compare with that of ridge regression in least square problems?*

In particular, even in the arguably simplest setting of linear regression, we seek to understand if/how SGD behaves differently from using an explicit $\ell_2$-regularizer, with a particular focus on the overparameterized regime.

**Our Contributions.** Due to recent advances on sharp, *instance-dependent* excess risks bounds of both (single-pass) SGD and ridge regression for overparameterized least square problems [26, 30], a nearly complete answer to the above question is now possible using these tools. In this work, we deliver an *instance-based* risk comparison between SGD and ridge regression in several interesting settings, including one-hot distributed data and Gaussian data. In particular, for a broad class of least squares problem instances that are natural in high-dimensional settings, we show that

- For every problem instance and for every ridge parameter, (unregularized) SGD, when provided with *logarithmically* more samples than that provided to ridge regularization, generalizes no worse than the ridge solution, provided SGD uses a tuned constant stepsize.
- Conversely, there exist instances in our problem class where optimally-tuned ridge regression requires *quadratically* more samples than SGD to achieve the same generalization performance.

Quite strikingly, the above results show that, up to some logarithmic factors, the generalization performance of SGD is always no worse than that of ridge regression in a wide range of overparameterized least square problems, and, in fact, could be much better for some problem instances. As a special case (for the above two claims), our problem class includes a setting in which: (i) the signal-to-noise is bounded and (ii) the eigenspectrum decays at a polynomial rate $1/i^\alpha$, for $0 \leq \alpha \leq 1$ (which permits a relatively fast decay). This one-sided near-domination phenomenon (in these natural overparameterized problem classes) could further support the preference for the implicit regularization brought by SGD over explicit ridge regularization.

Several novel technical contributions are made to make the above risk comparisons possible. For the one-hot data, we derive similar risk upper bound of SGD and risk lower bound of ridge regression. For the Gaussian data, while a sharp risk bound of SGD is borrowed from [30], we prove a sharp lower bound of ridge regression by adapting the proof techniques developed in [26, 7]. By carefully comparing these upper and lower bound results (and exhibiting particular instances to show that our sample size inflation bounds are sharp), we are able to provide nearly complete conditions that characterize when SGD generalizes better than ridge regression.

**Notation.** For two functions $f(x) \geq 0$ and $g(x) \geq 0$ defined on $x > 0$, we write $f(x) \lesssim g(x)$ if $f(x) \leq c \cdot g(x)$ for some absolute constant $c > 0$; we write $f(x) \gtrsim g(x)$ if $g(x) \lesssim f(x)$; we write $f(x) \approx g(x)$ if $f(x) \lesssim g(x) \lesssim f(x)$. For a vector $\mathbf{w} \in \mathbb{R}^d$ and a positive semidefinite matrix $\mathbf{H} \in \mathbb{R}^{d \times d}$, we denote $\|\mathbf{w}\|_{\mathbf{H}} := \sqrt{\mathbf{w}^\top \mathbf{H} \mathbf{w}}$.

## 2 Related Work

In terms of making sharp risk comparisons with ridge, the work of [10] shows that OLS (after a PCA projection is applied to the data) is instance-wise competitive with ridge on fixed design problems.

The insights in our analysis are draw from this work, though there are a number of technical challenges in dealing with the random design setting. We start with a brief discussion of the technical advances in the analysis of ridge regression and SGD, and then briefly overview more related work comparing SGD to explicit norm-based regularization.

**Excess Risk Bounds for Ridge Regression.** In the underparameterized regime, the excess risk bounds for ridge regression has been well-understood [16]. In the overparameterized regime, a large body of works [12, 15, 28, 27] focused on characterizing the excess risk of ridge regression in the asymptotic regime where both the sample size $N$ and dimension $d$ go to infinite and $d/N \to \gamma$ for some finite $\gamma$. More recently, Bartlett et al. [7] developed sharp non-asymptotic risk bounds for ordinary least square in the overparameterized setting, which are further extended to ridge regression by Tsigler and Bartlett [26]. These bounds have additional interest because they are instance-dependent, in particular, depending on the data covariance spectrum. The risk bounds of ridge regression derived in Tsigler and Bartlett [26] is highly nontrivial in the overparameterized setting as it holds when the ridge parameter equals to zero or even being negative. This line of results build one part of the theoretical tools for this paper.

**Excess Risk Bounds for SGD.** Risk bounds for one-pass, constant-stepsize (average) SGD have been derived in the finite dimensional case [4, 9, 17, 18, 11, 1]. Very recently, the work of [30] extends these analyses, providing sharp *instance-dependent* risk bound applicable to the overparameterized regime; here, Zou et al. [30] provides nearly matching upper and lower excess risk bounds for constant-stepsize SGD, which are sharply characterized in terms of the full eigenspectrum of the population covariance matrix. This result plays a pivotal role in our paper.

**Implicit Regularization of SGD vs. Explicit Norm-based Regularization.** For least square problems, multi-pass SGD converges to the minimum-norm solution [22, 29, 14], which is widely cited as (one of) the implicit bias of SGD. However, in more general settings, e.g., convex but non-linear models, a (distribution-independent) norm-based regularizer is no longer sufficient to characterize the optimization behavior of SGD [3, 8, 23]. Those discussions, however, exclude the possibility of *hyperparameter tuning*, e.g., stepsize for SGD and penalty strength for ridge regression, and are not instance-based, either. Our aim in this paper is to provide instance-based excess risk comparison between the optimally tuned (one-pass) SGD and the optimally tuned ridge regression.

## 3 Problem Setup and Preliminaries

We seek to compare the generalization ability of SGD and ridge algorithms for *least square problems*. We use $\mathbf{x} \in \mathcal{H}$ to denote a feature vector in a (separable) Hilbert space $\mathcal{H}$. We use $d$ to refer to the dimensionality of $\mathcal{H}$, where $d = \infty$ if $\mathcal{H}$ is infinite-dimensional. We use $y \in \mathbb{R}$ to denote a response that is generated by

$$y = \langle \mathbf{x}, \mathbf{w}^* \rangle + \xi,$$

where $\mathbf{w}^* \in \mathcal{H}$ is an unknown true model parameter and $\xi \in \mathbb{R}$ is the model noise. The following regularity assumption is made throughout the paper.

**Assumption 3.1** (Well-specified noise). *The second moment of* $\mathbf{x}$, *denoted by* $\mathbf{H} := \mathbb{E}[\mathbf{x}\mathbf{x}^\top]$, *is strictly positive definite and has finite trace. The noise* $\xi$ *is independent of* $\mathbf{x}$ *and satisfies*

$$\mathbb{E}[\xi] = 0, \quad and \quad \mathbb{E}[\xi^2] = \sigma^2.$$

In order to characterize the interplay between $\mathbf{w}^*$ and $\mathbf{H}$ in the excess risk bound, we introduce:

$$\mathbf{H}_{0:k} := \sum_{i=1}^k \lambda_i \mathbf{v}_i \mathbf{v}_i^\top, \quad \text{and} \quad \mathbf{H}_{k:\infty} := \sum_{i>k} \lambda_i \mathbf{v}_i \mathbf{v}_i^\top,$$

where $\{\lambda_i\}_{i=1}^\infty$ are the eigenvalues of $\mathbf{H}$ sorted in non-increasing order and $\mathbf{v}_i$'s are the corresponding eigenvectors. Then we define

$$\|\mathbf{w}\|_{\mathbf{H}_{0:k}^{-1}}^2 = \sum_{i \leq k} \frac{(\mathbf{v}_i^\top \mathbf{w})^2}{\lambda_i}, \quad \|\mathbf{w}\|_{\mathbf{H}_{k:\infty}}^2 = \sum_{i>k} \lambda_i (\mathbf{v}_i^\top \mathbf{w})^2.$$

The least squares problem is to estimate the true parameter $\mathbf{w}^*$. Assumption 3.1 implies that $\mathbf{w}^*$ is the unique solution that minimizes the *population risk*:

$$L(\mathbf{w}^*) = \min_{\mathbf{w} \in \mathcal{H}} L(\mathbf{w}), \quad \text{where } L(\mathbf{w}) := \frac{1}{2} \mathbb{E}_{(\mathbf{x},y)\sim\mathcal{D}}\big[(y - \langle \mathbf{w}, \mathbf{x} \rangle)^2\big]. \tag{3.1}$$

Moreover we have that $L(\mathbf{w}^*) = \sigma^2$. For an estimation $\mathbf{w}$ found by some algorithm, e.g., SGD or ridge regression, its performance is measured by the *excess risk*, $L(\mathbf{w}) - L(\mathbf{w}^*)$.

**Constant-Stepsize SGD with Tail-Averaging.** We consider the constant-stepsize SGD with tail-averaging [4, 17, 18, 30]: at the $t$-th iteration, a fresh example $(\mathbf{x}_t, y_t)$ is sampled independently from the data distribution, and SGD makes the following update on the current estimator $\mathbf{w}_{t-1} \in \mathcal{H}$,

$$\mathbf{w}_t = \mathbf{w}_{t-1} + \gamma \cdot \big(y_t - \langle \mathbf{w}_{t-1}, \mathbf{x}_t \rangle\big)\mathbf{x}_t, \ t = 1, 2, \ldots, \qquad \mathbf{w}_0 = 0,$$

where $\gamma > 0$ is a constant stepsize. After $N$ iterations (which is also the number of samples observed), SGD outputs the tail-averaged iterates as the final estimator:

$$\mathbf{w}_{\mathrm{sgd}}(N; \gamma) := \frac{2}{N} \sum_{t=N/2}^{N-1} \mathbf{w}_t.$$

In the underparameterized setting ($d < N$), constant-stepsize SGD with tail-averaging is known for achieving minimax optimal rate for least squares [17, 18]. More recently, Zou et al. [30] investigate the performance of constant-stepsize SGD with tail-averaging in the overparameterized regime ($d > N$), and establish *instance-dependent*, nearly-optimal excess risk bounds under mild assumptions on the data distribution. Notably, results from [30] cover underparameterized cases ($d < N$) as well.

**Ridge Regression.** Given $N$ i.i.d. samples $\{(\mathbf{x}_i, y_i)\}_{i=1}^N$, let us denote $\mathbf{X} := [\mathbf{x}_1, \ldots, \mathbf{x}_N]^\top \in \mathbb{R}^{N \times d}$ and $\mathbf{y} := [y_1, \ldots, y_N]^\top \in \mathbb{R}^d$. Then ridge regression outputs the following estimator for the true parameter [25]:

$$\mathbf{w}_{\mathrm{ridge}}(N; \lambda) := \arg\min_{\mathbf{w} \in \mathcal{H}} \|\mathbf{X}\mathbf{w} - \mathbf{y}\|_2^2 + \lambda \|\mathbf{w}\|_2^2, \tag{3.2}$$

where $\lambda$ (which could possibly be negative) is a regularization parameter. We remark that the ridge regression estimator takes the following two equivalent form:

$$\mathbf{w}_{\mathrm{ridge}}(N; \lambda) = (\mathbf{X}^\top \mathbf{X} + \lambda \mathbf{I}_d)^{-1} \mathbf{X}^\top \mathbf{y} = \mathbf{X}^\top (\mathbf{X}\mathbf{X}^\top + \lambda \mathbf{I}_N)^{-1} \mathbf{y}. \tag{3.3}$$

The first expression is useful in the classical, underparameterized setting ($d < N$) [16]; and the second expression is more useful in the overparameterized setting ($d > N$) where the empirical covariance $\mathbf{X}^\top \mathbf{X}$ is usually not invertible [20, 26]. As a final remark, when $\lambda = 0$, ridge estimator reduces to the *ordinary least square estimator* (OLS) [13].

**Generalizable Regime.** In the following sections we will make instance-based risk comparisons between SGD and ridge regression. To make the comparison meaningful, we focus on regime where SGD and ridge regression are "generalizable", i.e, the SGD and the ridge regression estimators, with the optimally-tuned hypeparameters, can achieve excess risk that is smaller than the optimal population risk, i.e., $\sigma^2$. The formal mathematical definition is as follows.

**Definition 1** (Generalizability). *Consider an algorithm* Alg *and a least squares problem instance* P. *Let* Alg$(n, \boldsymbol{\theta})$ *be the output of the algorithm when provided with $n$ i.i.d. samples from the problem instance* P, *and a set of hyperparameters $\boldsymbol{\theta}$ (that could be a function on $n$). Then we say that the algorithm* Alg *with sample size $n$ and hyperparameters configuration $\boldsymbol{\theta}$ is* generalizable *on problem instance* P, *if*

$$\mathbb{E}_{\mathrm{Alg,P}}[L\big(\mathrm{Alg}(n, \boldsymbol{\theta})\big)] - L(\mathbf{w}^*) \le \sigma^2,$$

*where the expectation is over the randomness of* Alg *and data drawn from the problem instance* P.

Clearly, the generalizable regime is defined by conditions on both the sample size, hyperparameter configuration, the problem instance, and the algorithm. For example, in the $d$-dimensional setting with $\|\mathbf{w}^*\|_2 = O(1)$, the ordinary least squares (OLS) solution (ridge regression with $\lambda = 0$), i.e., $\mathbf{w}_{\mathrm{ridge}}(N; 0)$ has $\mathcal{O}(d\sigma^2/N)$ excess risk, then we can say that the ridge regression with regularization parameter $\lambda = 0$ and sample size $N = \omega(d)$ is in the generalizable regime on all problem instances in $d$-dimension with $\|\mathbf{w}^*\|_2 = O(1)$.

**Sample Inflation vs. Risk Inflation Comparisons.** This work characterizes the *sample inflation* of SGD, i.e., bounding the required sample size of SGD to achieve an instance-based comparable excess risk as ridge regression (which is essentially the notion of Bahadur statistical efficiency [5, 6]). Another natural comparison would be examining the *risk inflation* of SGD, examining the instance-based increase in risk for any fixed sample size. Our preference for the former is due to the relative instability of the risk with respect to the sample size (in some cases, given a slightly different sample size, the risk could rapidly change.).

# 4 Warm-Up: One-Hot Least Squares Problems

Let us begin with a simpler data distribution, the *one-hot* data distribution. (inspired by settings where the input distribution is sparse). In detail, assume each input vector $\mathbf{x}$ is sampled from the set of natural basis $\{\mathbf{e}_1, \mathbf{e}_2, \ldots, \mathbf{e}_d\}$ according to the data distribution given by $\mathbb{P}\{\mathbf{x} = \mathbf{e}_i\} = \lambda_i$, where $0 < \lambda_i \leq 1$ and $\sum_i \lambda_i = 1$. The class of one-hot least square instances is completely characterized by the following problem set:

$$\big\{ (\mathbf{w}^*; \lambda_1, \cdots, \lambda_d) : \ \mathbf{w}^* \in \mathcal{H}, \ \textstyle\sum_i \lambda_i = 1, \ 1 \geq \lambda_1 \geq \lambda_2 \geq \cdots > 0 \big\}.$$

Clearly the population data covariance matrix is $\mathbf{H} = \operatorname{diag}(\lambda_1, \ldots, \lambda_d)$. The next two theorems give an instance-based sample inflation comparisons for this problem class.

**Theorem 4.1** (Instance-wise comparison, one-hot data). *Let $\mathbf{w}_{\mathrm{sgd}}(N; \gamma)$ and $\mathbf{w}_{\mathrm{ridge}}(N; \lambda)$ be the solutions found by SGD and ridge regression when using $N$ training examples. Then for any one-hot least square problem instance such that the ridge regression solution is generalizable and any $\lambda$, there exists a choice of stepsize $\gamma^*$ for SGD such that*

$$L\big[\mathbf{w}_{\mathrm{sgd}}(N_{\mathrm{sgd}}; \gamma^*)\big] - L(\mathbf{w}^*) \lesssim L\big[\mathbf{w}_{\mathrm{ridge}}(N_{\mathrm{ridge}}; \lambda)\big] - L(\mathbf{w}^*) < \sigma^2,$$

*provided the sample size of SGD satisfies*

$$N_{\mathrm{sgd}} \geq N_{\mathrm{ridge}}.$$

Theorem 4.1 suggests that for *every* one-hot problem instance, when provided with the same or more number of samples, the SGD solution with a properly tuned stepsize generalizes at most constant times worse than the optimally tuned ridge regression solution. In other words, with the same number of samples, SGD is *always* competitive with ridge regression.

**Theorem 4.2** (Best-case comparison, one-hot data). *There exists an one-hot least square problem instance satisfying $\|\mathbf{w}^*\|_{\mathbf{H}}^2 = \sigma^2$, and a SGD solution with constant stepsize and sample size $N_{\mathrm{sgd}}$, such that for any ridge regression solution with sample size*

$$N_{\mathrm{ridge}} \leq \frac{N_{\mathrm{sgd}}^2}{\log^2(N_{\mathrm{sgd}})},$$

*it holds that,*

$$L\big[\mathbf{w}_{\mathrm{ridge}}(N_{\mathrm{ridge}}; \lambda)\big] - L(\mathbf{w}^*) \gtrsim L\big[\mathbf{w}_{\mathrm{sgd}}(N_{\mathrm{sgd}}; \gamma^*)\big] - L(\mathbf{w}^*).$$

Theorem 4.2 shows that for some one-hot least square instance, ridge regression, even with the optimally-tuned regularization, needs at least (nearly) quadratically more samples than that provided to SGD, in order to compete with the optimally-tuned SGD. In other words, ridge regression could be much worse than SGD for one-hot least squares problems.

**Remark 4.3.** *The above two results together indicate a* superior *performance of the implicit regularization of SGD in comparison with the explicit regularization of ridge regression, for one-hot least squares problems. This is not the only case that SGD is always no worse than ridge estimator. In fact, we will next turn to compare SGD with ridge regression for the class of Gaussian least square instances, where both SGD and ridge regression exhibit richer behaviors but SGD still exhibits superiority over the ridge estimator.*

# 5 Gaussian Least Squares Problems

In this section, we consider least squares problems with a Gaussian data distribution. In particular, assume the population distribution of the input vector $\mathbf{x}$ is Gaussian[2], i.e., $\mathbf{x} \sim \mathcal{N}(\mathbf{0}, \mathbf{H})$. We further make the following regularity assumption for simplicity:

**Assumption 5.1.** $\mathbf{H}$ *is strictly positive definite and has a finite trace.*

Gaussian least squares problems are completely characterized by the following problem set $\big\{ (\mathbf{w}^*; \mathbf{H}) : \ \mathbf{w}^* \in \mathcal{H} \big\}.$

The next theorem give an instance-based sample inflation comparison between SGD and ridge regression for Gaussian least squares instances.

---

[2]We restrict ourselves to the Gaussian distribution for simplicity. Our results hold under more general assumptions, e.g., $\mathbf{H}^{-1/2}\mathbf{x}$ has sub-Gaussian tail and independent components [7] and is symmetrically distributed.

**Theorem 5.1** (Instance-wise comparison, Gaussian data). *Let $\mathbf{w}_{\mathrm{sgd}}(N; \gamma)$ and $\mathbf{w}_{\mathrm{ridge}}(N; \lambda)$ be the solutions found by SGD and ridge regression respectively. Then under Assumption 5.1, for any Gaussian least square problem instance such that the ridge regression solution is generalizable and any $\lambda$, there exists a choice of stepsize $\gamma^*$ for SGD such that*

$$L\big[\mathbf{w}_{\mathrm{sgd}}(N_{\mathrm{sgd}}; \gamma^*)\big] - L(\mathbf{w}^*) \lesssim L\big[\mathbf{w}_{\mathrm{ridge}}(N_{\mathrm{ridge}}; \lambda)\big] - L(\mathbf{w}^*),$$

*provided the sample size of SGD satisfies*

$$N_{\mathrm{sgd}} \geq (1 + R^2) \cdot \kappa(N_{\mathrm{ridge}}) \cdot \log(a) \cdot N_{\mathrm{ridge}},$$

*where*

$$\kappa(n) = \frac{\mathrm{tr}(\mathbf{H})}{n\lambda_{\min\{n,d\}}}, \quad R^2 = \frac{\|\mathbf{w}^*\|_{\mathbf{H}}^2}{\sigma^2}, \quad a = \kappa(N_{\mathrm{ridge}})R\sqrt{N}.$$

Note that the result in Theorem 5.1 holds for arbitrary $\lambda$. Then this theorem provides a sufficient condition for SGD such that it provably performs no worse than optimal ridge regression solution (i.e., ridge regression with optimal $\lambda$). Besides, we would also like to point out that the SGD stepsize $\gamma^*$ in Theorem 5.1 is only a function of the regularization parameter $\lambda$ and $\mathrm{tr}(\mathbf{H})$, which can be easily estimated from training dataset without knowing the exact formula of $\mathbf{H}$.

Different from the one-hot case, here the required sample size for SGD depends on two important quantities: $R^2$ and $\kappa(N_{\mathrm{ridge}})$. In particular, $R^2 = \|\mathbf{w}^*\|_{\mathbf{H}}^2/\sigma^2$ can be understood as the *signal-to-noise* ratio. The quantity $\kappa(N_{\mathrm{ridge}})$ characterizes the flatness of the eigenspectrum of $\mathbf{H}$ in the top $N_{\mathrm{ridge}}$-dimensional subspace, which clearly satisfies $\kappa(N_{\mathrm{ridge}}) \geq 1$. Let us further explain why we have the dependencies on $R^2$ and $\kappa(N_{\mathrm{ridge}})$ in the condition of the sample inflation for SGD.

A large $R^2$ emphasizes the problem hardness is more from the numerical optimization instead of from the statistic learning. In particular, let us consider a special case where $\sigma = 0$ and $R^2 = \infty$, i.e., there is no noise in the least square problem, and thus solving it is purely a numerical optimization issue. In this case, ridge regression with $\lambda = 0$ achieves *zero* population risk so long as the observed data can span the whole parameter space, but constant stepsize SGD in general suffers a non-zero risk in finite steps, thus cannot be competitive with the risk of ridge regression, which is as predicted by Theorem 5.1. From a learning perspective, a constant or even small $R^2$ is more interesting.

To explain why the dependency on $\kappa(N_{\mathrm{ridge}})$ is unavoidable, we can consider a 2-d dimensional example where

$$\mathbf{H} = \begin{pmatrix} 1 & 0 \\ 0 & \frac{1}{N_{\mathrm{ridge}} \cdot \kappa(N_{\mathrm{ridge}})} \end{pmatrix}, \quad \mathbf{w}^* = \begin{pmatrix} 0 \\ N_{\mathrm{ridge}} \cdot \kappa(N_{\mathrm{ridge}}) \end{pmatrix}.$$

It is commonly known that for this problem, ridge regression with $\lambda = 0$ can achieve $\mathcal{O}(\sigma^2/N_{\mathrm{ridge}})$ excess risk bound [13]. However, this problem is rather difficult for SGD since it is hard to learn the second coordinate of $\mathbf{w}^*$ using gradient information (the gradient in the second coordinate is quite small). In fact, in order to accurately learn $\mathbf{w}^*[2]$, SGD requires at least $\Omega(1/\lambda_2) = \Omega\big(N_{\mathrm{ridge}}\kappa(N_{\mathrm{ridge}})\big)$ iterations/samples, which is consistent with our theory.

Then from Theorem 5.1 it can be observed that when the signal-to-noise ratio is nearly a constant, i.e., $R^2 = \Theta(1)$, and the eigenspectrum of $\mathbf{H}$ does not decay too fast so that $\kappa(N_{\mathrm{ridge}}) \leq \mathrm{polylog}(N_{\mathrm{ridge}})$, SGD provably generalizes no worse than ridge regression, provided with logarithmically more samples than that provided to ridge regression. More specifically, the following corollary gives a family of problem instances that are in this regime.

**Corollary 5.1.** *Under the same conditions as Theorem 5.1, let $N_{\mathrm{ridge}}$ be the sample size of ridge regression. Consider the problem instance that satisfies $R^2 = \Theta(1)$, $d = O(N_{\mathrm{ridge}})$, and $\lambda_i = 1/i^\alpha$ for some $\alpha \leq 1$, then SGD, with a tuned stepsize $\gamma^*$, provably generalizes no worse than any ridge regression solution in the generalizable regime if*

$$N_{\mathrm{sgd}} \geq \log^2(N_{\mathrm{ridge}}) \cdot N_{\mathrm{ridge}}.$$

We would like to further point out that the comparison made in Corollary 5.1 concerns the worst-case result regarding $\mathbf{w}^*$ (from the perspective of SGD), while SGD could perform much better if $\mathbf{w}^*$ has a nice structure. For example, considering the same setting in Corollary 5.1 but assuming that the ground truth $\mathbf{w}^*$ is drawn from a prior distribution that is rotation invariant, SGD can be no worse than ridge regression provided the same or larger sample size. We formally state this result in the following corollary.

**Corollary 5.2.** *Under the same conditions as Corollary 5.1, let $N_{\mathrm{ridge}}$ be the sample size of ridge regression. Consider the problem instance with random and rotation invariant $\mathbf{w}^*$, then SGD with a tuned stepsize $\gamma^*$ provably generalizes no worse than any ridge regression solution in the generalizable regime if*

$$N_{\mathrm{sgd}} \geq N_{\mathrm{ridge}}.$$

The next theorem shows that, in fact, for some instances, SGD could perform much better than ridge regression, as for the one-hot least square problems.

**Theorem 5.2** (Best-case comparison, Gaussian data)**.** *There exists a Gaussian least square problem instance satisfying $R^2 = 1$ and $\kappa(N_{\mathrm{sgd}}) = \Theta(1)$, and an SGD solution with a constant stepsize and sample size $N_{\mathrm{sgd}}$, such that for any ridge regression solution (i.e., any $\lambda$) with sample size*

$$N_{\mathrm{ridge}} \leq \frac{N_{\mathrm{sgd}}^2}{\log^2(N_{\mathrm{sgd}})},$$

*it holds that,*

$$L\big[\mathbf{w}_{\mathrm{ridge}}(N_{\mathrm{ridge}};\lambda)\big] - L(\mathbf{w}^*) \gtrsim L\big[\mathbf{w}_{\mathrm{sgd}}(N_{\mathrm{sgd}};\gamma^*)\big] - L(\mathbf{w}^*).$$

Besides the instance-wise comparison, it is also interesting to see under what condition SGD can provably outperform ridge regression, i.e., achieving comparable or smaller excess risk using the *same* number of samples. The following theorem shows that this occurs when the signal-to-noise ratio $R^2$ is a constant and there is only a small fraction of $\mathbf{w}^*$ living in the tail eigenspace of $\mathbf{H}$.

**Theorem 5.3** (SGD outperforms ridge regression, Gaussian data)**.** *Let $N_{\mathrm{ridge}}$ be sample size of ridge regression and $k^* = \min\big\{k : \lambda_k \leq \frac{\mathrm{tr}(\mathbf{H})}{N_{\mathrm{ridge}}\log(N_{\mathrm{ridge}})}\big\}$, then if $R^2 = \Theta(1)$, and*

$$\sum_{i=k^*+1}^{N_{\mathrm{ridge}}} \lambda_i(\mathbf{w}^*[i])^2 \lesssim \frac{k^*\|\mathbf{w}^*\|_{\mathbf{H}}^2}{N_{\mathrm{ridge}}},$$

*for any ridge regression solution that is generalizable and any $\lambda$, there exists a choice of stepsize $\gamma^*$ for SGD such that*

$$L\big[\mathbf{w}_{\mathrm{sgd}}(N_{\mathrm{sgd}};\gamma^*)\big] - L(\mathbf{w}^*) \lesssim L\big[\mathbf{w}_{\mathrm{ridge}}(N_{\mathrm{ridge}};\lambda)\big] - L(\mathbf{w}^*)$$

*provided the sample size of SGD satisfies*

$$N_{\mathrm{sgd}} \geq N_{\mathrm{ridge}}.$$

**Experiments.** We perform experiments on Gaussian least square problem. We consider 6 problem instances, which are the combinations of 2 different covariance matrices $\mathbf{H}$: $\lambda_i = i^{-1}$ and $\lambda_i = i^{-2}$; and 3 different true model parameter vectors $\mathbf{w}^*$: $\mathbf{w}^*[i] = 1$, $\mathbf{w}^*[i] = i^{-1}$, and $\mathbf{w}^*[i] = i^{-10}$. Figure 1 compares the required sample sizes of ridge regression and SGD that lead to the same population risk on these 6 problem instances, where the hyperparameters (i.e., $\gamma$ and $\lambda$) are fine-tuned to achieve the best performance. We have two key observations: (1) in terms of the worst problem instance for SGD (i.e., $\mathbf{w}^*[i] = 1$), its sample size is only worse than ridge regression up to nearly constant factors (the curve is nearly linear); and (2) SGD can significantly outperform ridge regression when the true model $\mathbf{w}^*$ mainly lives in the head eigenspace of $\mathbf{H}$ (i.e., $\mathbf{w}^*[i] = i^{-10}$). The empirical observations are pretty consistent with our theoretical findings and again demonstrate the benefit of the implicit regularization of SGD.

## 6 An Overview of the Proof

In this section, we will sketch the proof of main Theorems for Gaussian least squares problems. Recall that we aim to show that provided certain number of training samples, SGD is guaranteed to generalize better than ridge regression. Therefore, we will compare the risk *upper bound* of SGD [30] with the risk *lower bound* of ridge regression [26][3]. In particular, we first provide the following informal lemma summarizing the aforementioned risk bounds of SGD and ridge regression.

---

[3]The lower bound of ridge regression in our paper is a tighter variant of the lower bound in Tsigler and Bartlett [26] since we consider Gaussian case and focus on the expected excess risk. Tsigler and Bartlett [26] studied the sub-Gaussian case and established a high-probability risk bound.

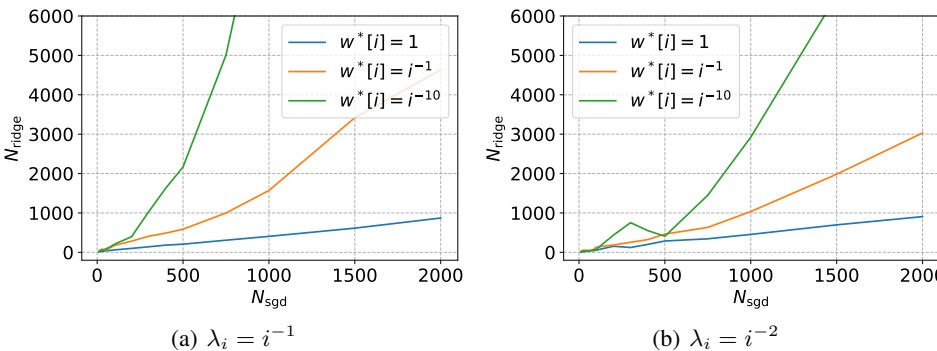

(a) $\lambda_i = i^{-1}$                                                (b) $\lambda_i = i^{-2}$

Figure 1: Sample size comparison between SGD and ridge regression, where the stepsize $\gamma$ and regularization parameter $\lambda$ are fine-tuned to achieve the best performance. The problem dimension is $d = 200$ and the variance of model noise is $\sigma^2 = 1$. We consider 6 combinations of 2 different covariance matrices and 3 different ground truth model vectors. The plots are averaged over 20 independent runs.

**Lemma 6.1** (Risk bounds of SGD and ridge regression, informal). *Suppose Assumptions 3.1 and 5.1 hold and $\gamma \le 1/\operatorname{tr}(\mathbf{H})$, then SGD has the following risk upper bound for arbitrary $k_1, k_2 \in [d]$,*

$$\text{SGDRisk} \lesssim \underbrace{\frac{1}{\gamma^2 N_{\text{sgd}}^2} \cdot \left\| \exp(-N_{\text{sgd}}\gamma\mathbf{H})\mathbf{w}^* \right\|^2_{\mathbf{H}_{0:k_1}^{-1}} + \left\| \mathbf{w}^* \right\|^2_{\mathbf{H}_{k_1:\infty}}}_{\text{SGDBiasBound}}$$

$$+ \underbrace{(1 + R^2)\sigma^2 \cdot \left( \frac{k_2}{N_{\text{sgd}}} + N_{\text{sgd}}\gamma^2 \sum_{i>k_2} \lambda_i^2 \right)}_{\text{SGDVarianceBound}}. \tag{6.1}$$

*Additionally, ridge regression has the following risk lower bound for a constant $\widetilde{\lambda}$, depending on $\lambda$, $N_{\text{ridge}}$, and $\mathbf{H}$, and $k^* = \min\{k : N_{\text{ridge}}\lambda_k \lesssim \widetilde{\lambda}\}$*

$$\text{RidgeRisk} \gtrsim \underbrace{\left( \frac{\widetilde{\lambda}}{N_{\text{ridge}}} \right)^2 \|\mathbf{w}^*\|^2_{\mathbf{H}_{0:k^*}^{-1}} + \|\mathbf{w}^*\|^2_{\mathbf{H}_{k^*:\infty}}}_{\text{RidgeBiasBound}} + \underbrace{\sigma^2 \cdot \left( \frac{k^*}{N_{\text{ridge}}} + \frac{N_{\text{ridge}}}{\widetilde{\lambda}^2} \sum_{i>k^*} \lambda_i^2 \right)}_{\text{RidgeVarianceBound}}. \tag{6.2}$$

We first highlight some useful observations in Lemma 6.1.

1. SGD has a condition on the stepsize: $\gamma \le 1/\operatorname{tr}(\mathbf{H})$, while ridge regression has no condition on the regularization parameter $\lambda$.

2. Both the upper bound of SGD and the lower bound of ridge regression can be decomposed into two parts corresponding to the head and tail eigenspaces of $\mathbf{H}$. Furthermore, for the upper bound of SGD, the decomposition is arbitrary ($k_1$ and $k_2$ are arbitrary), while for the lower bound of ridge estimator, the decomposition is fixed (i.e., $k^*$ is fixed).

3. Regarding the SGDBiasBound and SGDVarianceBound, performing the transformation $N \to \alpha N$ and $\gamma \to \alpha^{-1}\gamma$ will decrease SGDVarianceBound by a factor of $\alpha$ while the SGDBiasBound remains unchanged.

Based on the above useful observations, we can now interpret the proof sketch for Theorems 5.1, 5.2, and 5.3. We will first give the sketch for Theorem 5.3 and then prove Theorem 5.2 for the ease of presentation. We would like to emphasize that the calculation in the proof sketch may not be the sharpest since they are presented for the ease of exposition. A preciser and sharper calculation can be found in Appendix.

**Proof Sketch of Theorem 5.1.** In order to perform instance-wise comparison, we need to take care of all possible $\mathbf{w}^* \in \mathcal{H}$. Therefore, by Observation 2, we can simply pick $k_1 = k_2 = k^*$ in the upper

bound (6.1). Then it is clear that if setting $\gamma = \widetilde{\lambda}^{-1}$ and $N_{\text{sgd}} = N_{\text{ridge}}$, we have

$$\text{SGDBiasBound} \leq \text{RidgeBiasBound}$$

$$\text{SGDVarianceBound} = (1 + R^2) \cdot \text{RidgeVarianceBound}.$$

Then by Observation 3, enlarging $N_{\text{sgd}}$ by $(1 + R^2)$ times suffices to guarantee

$$\text{SGDBiasBound} + \text{SGDVarianceBound} \leq \text{RidgeBiasBound} + \text{RidgeVarianceBound}.$$

On the other hand, according to Observation 1, there is an upper bound on the feasible stepsize of SGD: $\gamma \leq 1/\operatorname{tr}(\mathbf{H})$. Therefore, the above claim only holds when $\widetilde{\lambda} \geq \operatorname{tr}(\mathbf{H})$.

When $\widetilde{\lambda} \leq \operatorname{tr}(\mathbf{H})$, the stepsize $\widetilde{\lambda}^{-1}$ is no longer feasible and instead, we will use the largest possible stepsize: $\gamma = 1/\operatorname{tr}(\mathbf{H})$. Besides, note that we assume ridge regression solution is in the generalizable regime, then it holds that $k^* \leq N_{\text{ridge}}$ since otherwise we have

$$\text{RidgeRisk} \gtrsim \text{RidgeVarianceBound} \geq \sigma^2.$$

Then again we set $k_1 = k_2 = k^*$ in SGDBiasBound and SGDVarianceBound. Applying the choice of stepsize $\gamma = 1/\operatorname{tr}(\mathbf{H})$ and sample size

$$N_{\text{sgd}} = \frac{\log(R^2 N_{\text{ridge}})}{\gamma \lambda_{k^*}} \leq N_{\text{ridge}} \cdot \kappa(N_{\text{ridge}}) \cdot \log(R^2 N_{\text{ridge}}),$$

we get

$$\begin{aligned}
\text{SGDBiasBound} &\leq \frac{(1 - N_{\text{sgd}} \gamma \lambda_{k^*})^{N_{\text{sgd}}}}{\gamma^2 N_{\text{sgd}}^2 \lambda_{k^*}^2} \cdot \|\mathbf{w}^*\|_{\mathbf{H}_{0:k^*}}^2 + \|\mathbf{w}^*\|_{\mathbf{H}_{k^*:\infty}}^2 \\
&\leq \frac{\sigma^2}{N_{\text{ridge}}} + \|\mathbf{w}^*\|_{\mathbf{H}_{k^*:\infty}}^2 \\
&\leq \text{RidgeBiasBound} + \text{RidgeVarianceBound}. \qquad (6.3)
\end{aligned}$$

Moreover, we can also get the following bound on SGDVarianceBound,

$$\begin{aligned}
\text{SGDVarianceBound} &\leq (1 + R^2)\sigma^2 \cdot \left( \frac{k^*}{N_{\text{ridge}}} + \frac{\log(R^2 N_{\text{ridge}})}{\lambda_{k^*} \operatorname{tr}(\mathbf{H})} \sum_{i > k^*} \lambda_i^2 \right) \\
&\leq (1 + R^2) \log(R^2 N_{\text{ridge}}) \cdot \text{RidgeVarianceBound},
\end{aligned}$$

where in the second inequality we use the fact that

$$\frac{N_{\text{ridge}}}{\widetilde{\lambda}^2} \geq \frac{1}{\lambda_{k^*} \widetilde{\lambda}} \geq \frac{1}{\lambda_{k^*} \operatorname{tr}(\mathbf{H})}.$$

Therefore by Observation 3 again we can enlarge $N_{\text{sgd}}$ properly to ensure that SGDVarianceBound remains unchanged and SGDVarianceBound $\leq$ RidgeVarianceBound. Then combining this and (6.3) we can get

$$\text{SGDBiasBound} + \text{SGDVarianceBound} \leq 2 \cdot \text{RidgeBiasBound} + 2 \cdot \text{RidgeVarianceBound},$$

which completes the proof.

**Proof Sketch of Theorem 5.3.** Now we will investigate in which regime SGD will generalizes no worse than ridge regression when provided with same training sample size. For simplicity in the proof we assume $R^2 = 1$. First note that we only need to deal with the case where $\widetilde{\lambda} \leq \operatorname{tr}(\mathbf{H})$ by the proof sketch of Theorem 5.1.

Unlike the instance-wise comparison that consider all possible $\mathbf{w}^* \in \mathcal{H}$, in this lemma we only consider the set of $\mathbf{w}^*$ that SGD performs well. Specifically, as we have shown in the proof of Theorem 5.1, in the worst-case comparison (in terms of $\mathbf{w}^*$), we require SGD to be able to learn the first $k^*$ (where $k^* \leq N_{\text{ridge}}$) coordinates of $\mathbf{w}^*$ in order to be competitive with ridge regression, while SGD with sample size $N_{\text{sgd}}$ can only be guaranteed to learn the first $k_{\text{sgd}}^*$ coordinates of $\mathbf{w}^*$, where $k_{\text{sgd}}^* = \min\{k : N_{\text{ridge}} \lambda_k \leq \operatorname{tr}(\mathbf{H})\}$. Therefore, in the instance-wise comparison we need to enlarge $N_{\text{sgd}}$ to $N_{\text{ridge}} \cdot \kappa(N_{\text{ridge}})$ to guarantee the learning of the top $k^*$ coordinates of $\mathbf{w}^*$.

However, this is not required for some good $\mathbf{w}^*$'s that have small components in the $k^*_{\text{sgd}}$-$k^*$ coordinates. In particular, as assumed in the theorem, we have $\sum_{i=\widehat{k}+1}^{N_{\text{ridge}}} \lambda_i(\mathbf{w}^*[i])^2 \leq \widehat{k} \|\mathbf{w}^*\|_{\mathbf{H}}^2 / N_{\text{ridge}}$, where $\widehat{k} := \min\{k : \lambda_k N_{\text{sgd}} \leq \text{tr}(\mathbf{H}) \cdot \log(N_{\text{sgd}})\}$ satisfies $\widehat{k} \leq k^*_{\text{sgd}} \leq k^*$. Then let $k_1 = \widehat{k}$ in SGDBiasBound, we have

$$\text{SGDBiasBound} = \frac{1}{\gamma^2 N_{\text{ridge}}^2} \cdot \big\| \exp(-N_{\text{ridge}}\gamma\mathbf{H})\mathbf{w}^* \big\|_{\mathbf{H}_{0:\widehat{k}}^{-1}}^2 + \|\mathbf{w}^*\|_{\mathbf{H}_{\widehat{k}:\infty}}^2$$

$$\leq (1 - N_{\text{ridge}}\gamma\lambda_{\widehat{k}})^{N_{\text{ridge}}} \cdot \|\mathbf{w}^*\|_{\mathbf{H}_{0:k^*}}^2 + \|\mathbf{w}^*\|_{\mathbf{H}_{\widehat{k}:\infty}}^2$$

$$\overset{(i)}{\leq} \frac{R^2\sigma^2(\widehat{k}+1)}{N_{\text{ridge}}} + \|\mathbf{w}^*\|_{\mathbf{H}_{k^*:\infty}}^2$$

$$\leq 2 \cdot \text{RidgeVarBound} + \text{RidgeBiasBound}.$$

where $(i)$ is due to the condition that $\sum_{i=\widehat{k}+1}^{N_{\text{ridge}}} \lambda_i(\mathbf{w}^*[i])^2 \leq \widehat{k} \|\mathbf{w}^*\|_{\mathbf{H}}^2 / N_{\text{ridge}}$. Moreover, it is easy to see that given $N_{\text{sgd}} = N_{\text{ridge}}$ and $\gamma = 1/\text{tr}(\mathbf{H}) \leq 1/\widetilde{\lambda}$, we have SGDVarianceBound $\leq 2 \cdot$ RidgeVarianceBound. As a consequence we can get

$\text{SGDBiasBound} + \text{SGDVarianceBound} \leq 3 \cdot \text{RidgeBiasBound} + 3 \cdot \text{RidgeVarianceBound}.$

**Proof Sketch of Theorem 5.2.** We will consider the best $\mathbf{w}^*$ for SGD, which only has nonzero entry in the first coordinate. For example, consider a true model parameter vector with $\mathbf{w}^*[1] = 1$ and $\mathbf{w}^*[i] = 0$ for $i \geq 2$ and a problem instance whose spectrum of $\mathbf{H}$ has a flat tail with $\sum_{i \geq N_{\text{ridge}}} \lambda_i^2 = \Theta(1)$ and $\sum_{i \geq 2} \lambda_i^2 = \Theta(1)$. Then according to Lemma 6.1, we can set the stepsize as $\gamma = \Theta(\log(N_{\text{sgd}})/N_{\text{sgd}})$ and get

$$\text{SGDRisk} \lesssim \text{SGDBiasBound} + \text{SGDVarianceBound}$$

$$= O\left(\frac{1}{N_{\text{sgd}}} + \frac{\log^2(N_{\text{sgd}})}{N_{\text{sgd}}}\right) = O\left(\frac{\log^2(N_{\text{sgd}})}{N_{\text{sgd}}}\right).$$

For ridge regression, according to Lemma 6.1 we have

$$\text{RidgeRisk} \gtrsim \text{RidgeBiasBound} + \text{RidgeVarianceBound}$$

$$= \Omega\left(\frac{\widetilde{\lambda}^2}{N_{\text{ridge}}^2} + \frac{N_{\text{ridge}}}{\widetilde{\lambda}^2}\right) \qquad \text{since } \sum_{i \geq k^*} \lambda_i^2 = \Theta(1)$$

$$= \Omega\left(\frac{1}{N_{\text{ridge}}^{1/2}}\right). \qquad \text{by the fact that } a + b \geq \sqrt{ab}$$

Therefore, it is evident that ridge regression is guaranteed to be worse than SGD if $N_{\text{ridge}} \leq N_{\text{sgd}}^2 / \log^2(N_{\text{sgd}})$. This completes the proof.

# 7  Conclusions

We conduct an instance-based risk comparison between SGD and ridge regression for a broad class of least square problems. We show that SGD is always no worse than ridge regression provided logarithmically more samples. On the other hand, there exist some instances where even optimally-tuned ridge regression needs quadratically more samples to compete with SGD. This separation in terms of sample inflation between SGD and ridge regression suggests a provable benefit of implicit regularization over explicit regularization for least squares problems. In the future, we will explore the benefits of implicit regularization for learning other linear models and potentially nonlinear models.

## Acknowledgments and Disclose of Funding

We would like to thank the anonymous reviewers and area chairs for their helpful comments. DZ is supported by the Bloomberg Data Science Ph.D. Fellowship. JW is supported in part by NSF CAREER grant 1652257. VB is supported in part by NSF CAREER grant 1652257, ONR Award N00014-18-1-2364 and the Lifelong Learning Machines program from DARPA/MTO. QG is supported in part by the National Science Foundation awards IIS-1855099 and IIS-2008981. SK acknowledges funding from the National Science Foundation under Award CCF-1703574. The views and conclusions contained in this paper are those of the authors and should not be interpreted as representing any funding agencies.

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
