&= \Omega\bigg(\frac{1}{N_{\text{ridge}}^{1/2}}\bigg). \qquad \text{by the fact that } a + b \ge \sqrt{ab}
\end{aligned}
$$

Therefore, it is evident that ridge regression is guaranteed to be worse than SGD if $N_{\text{ridge}} \le N_{\text{sgd}}^2/\log^2(N_{\text{sgd}})$. This completes the proof.

# 7 Conclusions

We conduct an instance-based risk comparison between SGD and ridge regression for a broad class of least square problems. We show that SGD is always no worse than ridge regression provided logarithmically more samples. On the other hand, there exist some instances where even optimally-tuned ridge regression needs quadratically more samples to compete with SGD. This separation in terms of sample inflation between SGD and ridge regression suggests a provable benefit of implicit regularization over explicit regularization for least squares problems. In the future, we will explore the benefits of implicit regularization for learning other linear models and potentially nonlinear models.

## Acknowledgments and Disclose of Funding

We would like to thank the anonymous reviewers and area chairs for their helpful comments. DZ is supported by the Bloomberg Data Science Ph.D. Fellowship. JW is supported in part by NSF CAREER grant 1652257. VB is supported in part by NSF CAREER grant 1652257, ONR Award N00014-18-1-2364 and the Lifelong Learning Machines program from DARPA/MTO. QG is supported in part by the National Science Foundation awards IIS-1855099 and IIS-2008981. SK acknowledges funding from the National Science Foundation under Award CCF-1703574. The views and conclusions contained in this paper are those of the authors and should not be interpreted as representing any funding agencies.

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

# A Proof of One-hot Least Squares

## A.1 Excess risk bound of SGD

In this part we will mainly follow the proof technique in Zou et al. [30] that is developed to sharply characterize the excess risk bound for SGD (with tail-averaging) when the data distribution has a nice finite fourth-moment bound. However, such condition does not hold for the one-hot case so that their results cannot be directly applied here.

Before presenting the detailed proofs, we first introduce some notations and definitions that will be repeatedly used in the subsequent analysis. Let $\mathbf{H} = \mathbb{E}[\mathbf{x}\mathbf{x}^\top]$ be the covariance of data distribution. It is easy to verify that $\mathbf{H}$ is a diagonal matrix with eigenvalues $\lambda_1, \ldots, \lambda_d$. Let $\mathbf{w}_t$ be the $t$-th iterate of the SGD, we define $\boldsymbol{\eta}_t := \mathbf{w}_t - \mathbf{w}^*$ as the centered SGD iterate. Then we define $\boldsymbol{\eta}_t^{\text{bias}}$ and $\boldsymbol{\eta}_t^{\text{variance}}$ as the bias error and variance error respectively, which are described by the following update rule:

$$\boldsymbol{\eta}_t^{\text{bias}} = \big(\mathbf{I} - \gamma\mathbf{x}_t\mathbf{x}_t^\top\big)\boldsymbol{\eta}_{t-1}^{\text{bias}}, \qquad \boldsymbol{\eta}_0^{\text{bias}} = \boldsymbol{\eta}_0,$$
$$\boldsymbol{\eta}_t^{\text{variance}} = \big(\mathbf{I} - \gamma\mathbf{x}_t\mathbf{x}_t^\top\big)\boldsymbol{\eta}_{t-1}^{\text{bias}} + \gamma\xi_t\mathbf{x}_t, \qquad \boldsymbol{\eta}_0^{\text{variance}} = \mathbf{0}. \tag{A.1}$$

Accordingly, we can further define the bias covariance $\mathbf{B}_t$ and variance covariance $\mathbf{C}_t$ as follows

$$\mathbf{B}_t = \mathbb{E}[\boldsymbol{\eta}_t^{\text{bias}} \otimes \boldsymbol{\eta}_t^{\text{bias}}], \qquad \mathbf{C}_t = \mathbb{E}[\boldsymbol{\eta}_t^{\text{variance}} \otimes \boldsymbol{\eta}_t^{\text{variance}}].$$

Regarding these two covariance matrices, the following lemma mathematically characterizes the upper bounds of the diagonal entries of $\mathbf{B}_t$ and $\mathbf{C}_t$.

**Lemma A.1.** *Under Assumptions 3.1, let $\bar{\mathbf{B}}_t = \text{diag}(\mathbf{B}_t)$ and $\bar{\mathbf{C}}_t = \text{diag}(\mathbf{C}_t)$, then if the stepsize satisfies $\gamma \leq 1$, we have*

$$\bar{\mathbf{B}}_t \preceq (\mathbf{I} - \gamma\mathbf{H})\bar{\mathbf{B}}_{t-1}, \qquad \bar{\mathbf{C}}_t \preceq (\mathbf{I} - \gamma\mathbf{H})\bar{\mathbf{C}}_{t-1} + \gamma^2\sigma^2\mathbf{H}.$$

*Proof.* According to (A.1), we have

$$\mathbf{B}_t = \mathbb{E}[\boldsymbol{\eta}_t^{\text{bias}} \otimes \boldsymbol{\eta}_t^{\text{bias}}] = \mathbb{E}\big[(\mathbf{I} - \gamma\mathbf{x}_t\mathbf{x}_t^\top)\boldsymbol{\eta}_{t-1}^{\text{bias}} \otimes (\mathbf{I} - \gamma\mathbf{x}_t\mathbf{x}_t^\top)\boldsymbol{\eta}_{t-1}^{\text{bias}}\big]$$
$$= \mathbf{B}_{t-1} - \gamma\mathbf{H}\mathbf{B}_{t-1} - \gamma\mathbf{B}_{t-1}\mathbf{H} + \gamma^2\mathbb{E}[\mathbf{x}_t\mathbf{x}_t^\top\mathbf{B}_{t-1}\mathbf{x}_t\mathbf{x}_t^\top]. \tag{A.2}$$

Note that $\mathbf{x}_t = \mathbf{e}_i$ with probability $\lambda_i$, then we have

$$\mathbb{E}[\mathbf{x}_t\mathbf{x}_t^\top\mathbf{B}_{t-1}\mathbf{x}_t\mathbf{x}_t^\top] = \sum_i \lambda_i \cdot \mathbf{e}_i\mathbf{e}_i^\top\mathbf{B}_{t-1}\mathbf{e}_i\mathbf{e}_i^\top$$
$$= \sum_i \lambda_i \cdot \mathbf{e}_i^\top\mathbf{B}_{t-1}\mathbf{e}_i \cdot \mathbf{e}_i\mathbf{e}_i^\top$$
$$= \bar{\mathbf{B}}_{t-1}\mathbf{H}.$$

Plugging the above equation into (A.2) gives

$$\mathbf{B}_t = \mathbf{B}_{t-1} - \gamma\mathbf{H}\mathbf{B}_{t-1} - \gamma\mathbf{B}_{t-1}\mathbf{H} + \gamma^2\bar{\mathbf{B}}_{t-1}\mathbf{H}.$$

Then if only look at the diagonal entries of both sides, we have

$$\bar{\mathbf{B}}_t = \bar{\mathbf{B}}_{t-1} - 2\gamma\mathbf{H}\bar{\mathbf{B}}_{t-1} + \gamma^2\mathbf{H}\bar{\mathbf{B}}_{t-1} \preceq (\mathbf{I} - \gamma\mathbf{H})\bar{\mathbf{B}}_{t-1},$$

where in the first equation we use the fact that $\text{diag}(\mathbf{H}\mathbf{B}) = \text{diag}(\mathbf{B}\mathbf{H}) = \mathbf{H}\bar{\mathbf{B}}$ and the inequality follows from the fact that both $\bar{\mathbf{B}}_t$ and $\mathbf{H}$ are diagonal and $\gamma \leq 1$.

Similarly, regarding $\mathbf{C}_t$ the following holds according to (A.1),

$$\mathbf{C}_t = \mathbb{E}\big[(\mathbf{I} - \gamma\mathbf{x}_t\mathbf{x}_t^\top)\boldsymbol{\eta}_{t-1}^{\text{variance}} \otimes (\mathbf{I} - \gamma\mathbf{x}_t\mathbf{x}_t^\top)\boldsymbol{\eta}_{t-1}^{\text{variance}}\big] + \gamma^2\mathbb{E}[\xi_t^2\mathbf{x}_t\mathbf{x}_t^\top],$$

where we use the fact that $\mathbb{E}[\xi_t|\mathbf{x}_t] = 0$. Similar to deriving the bound for $\bar{\mathbf{B}}_t$, we have

$$\text{diag}\big(\mathbb{E}\big[(\mathbf{I} - \gamma\mathbf{x}_t\mathbf{x}_t^\top)\boldsymbol{\eta}_t^{\text{variance}} \otimes (\mathbf{I} - \gamma\mathbf{x}_t\mathbf{x}_t^\top)\boldsymbol{\eta}_t^{\text{variance}}\big]\big) \preceq (\mathbf{I} - \gamma\mathbf{H})\bar{\mathbf{C}}_{t-1}.$$

Besides, under Assumption 3.1 we also have $\mathbb{E}[\xi_t^2\mathbf{x}_t\mathbf{x}_t^\top] = \sigma^2\mathbf{H}$, which is a diagonal matrix. Based on these two results, we can get the following upper bound for $\bar{\mathbf{C}}_t$,

$$\bar{\mathbf{C}}_t = \text{diag}\big(\mathbb{E}\big[(\mathbf{I} - \gamma\mathbf{x}_t\mathbf{x}_t^\top)\boldsymbol{\eta}_{t-1}^{\text{variance}} \otimes (\mathbf{I} - \gamma\mathbf{x}_t\mathbf{x}_t^\top)\boldsymbol{\eta}_{t-1}^{\text{variance}}\big] + \gamma^2\mathbb{E}[\xi_t^2\mathbf{x}_t\mathbf{x}_t^\top]\big)$$
$$\preceq (\mathbf{I} - \gamma\mathbf{H})\bar{\mathbf{C}}_{t-1} + \gamma^2\sigma^2\mathbf{H}.$$

This completes the proof. $\qquad\square$

**Lemma A.2** (Lemmas D.1 & D.2 in Zou et al. [30]). *Let $\bar{\mathbf{w}}_{N:2N}$ be the output of tail-averaged SGD, then if the stepsize satisfied $\gamma \leq 1/\lambda_1$, it holds that*

$$\mathbb{E}[L(\bar{\mathbf{w}}_{N:2N})] - L(\mathbf{w}^*) \lesssim \text{SGDBias} + \text{SGDVariance},$$

*where*

$$\text{SGDBias} \leq \frac{1}{N^2} \sum_{t=0}^{N-1} \sum_{k=t}^{N-1} \left\langle (\mathbf{I} - \gamma\mathbf{H})^{k-t}\mathbf{H}, \mathbf{B}_{N+t} \right\rangle$$

$$\text{SGDVariance} \leq \frac{1}{N^2} \sum_{t=0}^{N-1} \sum_{k=t}^{N-1} \left\langle (\mathbf{I} - \gamma\mathbf{H})^{k-t}\mathbf{H}, \mathbf{C}_{N+t} \right\rangle$$

**Lemma A.3.** *Under Assumptions 3.1, if the stepsize satisfies $\gamma \leq 1$ and set $\mathbf{w}_0 = \mathbf{0}$, then*

$$\mathbb{E}[L(\bar{\mathbf{w}}_{N:2N})] - L(\mathbf{w}^*) \leq 2 \cdot \text{bias} + 2 \cdot \text{variance},$$

*where*

$$\text{bias} \lesssim \frac{1}{N^2\gamma^2} \cdot \left\| (\mathbf{I} - \gamma\mathbf{H})^{N/2}\mathbf{w}^* \right\|_{\mathbf{H}_{0:k_1}^{-1}} + \left\| (\mathbf{I} - \gamma\mathbf{H})^{N/2}\mathbf{w}^* \right\|_{\mathbf{H}_{k_1:\infty}}^2$$

$$\text{variance} \lesssim \sigma^2 \cdot \left( \frac{k_2}{N} + N\gamma^2 \sum_{i>k_2} \lambda_i^2 \right)$$

*for arbitrary $k_1, k_2 \in [d]$.*

*Proof.* The first conclusion of this theorem can be directly proved via Young's inequality.

Note that $\mathbf{H}$ is a diagonal matrix, and thus $(\mathbf{I} - \gamma\mathbf{H})^{k-t}$ is also a diagonal matrix for all $k$ and $t$. Therefore, by Lemma A.2, it is clear that in order to calculate the upper bound of the bias and variance error, it suffices to consider the diagonal entries of $\mathbf{B}_{N+t}$ and $\mathbf{C}_{N+t}$, denoted by $\bar{\mathbf{B}}_{N+t}$ and $\bar{\mathbf{C}}_{N+t}$ (which are obtained by setting all non-diagonal entries of $\mathbf{B}_{N+t}$ and $\mathbf{C}_{N+t}$ as zero). Then by Young's inequality, Lemma A.2 implies that

$$\text{bias} \leq \frac{1}{N^2} \sum_{t=0}^{N-1} \sum_{k=t}^{N-1} \left\langle (\mathbf{I} - \gamma\mathbf{H})^{k-t}\mathbf{H}, \bar{\mathbf{B}}_{N+t} \right\rangle$$

$$\text{variance} \leq \frac{1}{N^2} \sum_{t=0}^{N-1} \sum_{k=t}^{N-1} \left\langle (\mathbf{I} - \gamma\mathbf{H})^{k-t}\mathbf{H}, \bar{\mathbf{C}}_{N+t} \right\rangle. \tag{A.3}$$

Now we are ready to precisely calculate the above two bounds. In particular, by Lemma A.1 we have

$$\bar{\mathbf{B}}_t \preceq (\mathbf{I} - \gamma\mathbf{H})\bar{\mathbf{B}}_{t-1} \preceq (\mathbf{I} - \gamma\mathbf{H})^t\mathbf{B}_0, \tag{A.4}$$

$$\bar{\mathbf{C}}_t \preceq (\mathbf{I} - \gamma\mathbf{H})\bar{\mathbf{C}}_{t-1} \preceq \sum_{s=0}^{t-1} \sigma^2\gamma^2(\mathbf{I} - \gamma\mathbf{H})^s\mathbf{H} = \sigma^2\gamma\big(\mathbf{I} - (\mathbf{I} - \gamma\mathbf{H})^t\big), \tag{A.5}$$

where in the second inequality we use the fact that $\mathbf{C}_0 = \boldsymbol{\eta}_t^{\text{variance}} \otimes \boldsymbol{\eta}_t^{\text{variance}} = \mathbf{0}$. Then plugging (A.4) into (A.3) gives

$$\text{bias} \leq \frac{1}{N^2} \sum_{t=0}^{N-1} \sum_{k=t}^{N-1} \langle (\mathbf{I} - \gamma\mathbf{H})^{k-t}\mathbf{H}, (\mathbf{I} - \gamma\mathbf{H})^{N+t}\mathbf{B}_0 \rangle$$

$$= \frac{1}{N^2} \left\langle \sum_{k=0}^{N-1-t} (\mathbf{I} - \gamma\mathbf{H})^k\mathbf{H}, \sum_{t=0}^{N-1} (\mathbf{I} - \gamma\mathbf{H})^{N+t}\mathbf{B}_0 \right\rangle$$

$$\leq \frac{1}{N^2} \left\langle \sum_{k=0}^{N-1} (\mathbf{I} - \gamma\mathbf{H})^k\mathbf{H}, \sum_{t=0}^{N-1} (\mathbf{I} - \gamma\mathbf{H})^{N+t}\mathbf{B}_0 \right\rangle$$

$$= \frac{1}{N^2\gamma^2} \left\langle \mathbf{I} - (\mathbf{I} - \gamma\mathbf{H})^N, \mathbf{H}^{-1}(\mathbf{I} - \gamma\mathbf{H})^N\big(\mathbf{I} - (\mathbf{I} - \gamma\mathbf{H})^N\big)\mathbf{B}_0 \right\rangle$$

$$= \frac{1}{N^2\gamma^2} \left\langle (\mathbf{I} - \gamma\mathbf{H})^N\big[\mathbf{I} - (\mathbf{I} - \gamma\mathbf{H})^N\big]^2\mathbf{H}^{-1}, \mathbf{B}_0 \right\rangle \tag{A.6}$$

Note that $(1-x)^N \geq \min\{0, 1 - Nx\}$ for all $x \in [0, 1]$. Then for all $i$ we have

$$\big[1 - (1 - \gamma\lambda_i)^N\big]^2 \lambda^{-1} \leq \min\left\{\frac{1}{\lambda_i}, N^2\gamma^2\lambda_i\right\}$$

where we use the fact that $\gamma \leq 1 \leq 1/\lambda_i$ for all $i$. This further implies that

$$\big[\mathbf{I} - (\mathbf{I} - \gamma\mathbf{H})^N\big]^2 \mathbf{H}^{-1} \preceq \mathbf{H}_{0:k}^{-1} + N^2\gamma^2\mathbf{H}_{k:\infty}$$

for all $k \in [d]$. Plugging the above results into (A.6) leads to

$$\text{bias} \leq \frac{1}{N^2\gamma^2} \cdot \big\langle \mathbf{H}_{0:k}^{-1}, (\mathbf{I} - \gamma\mathbf{H})^N \mathbf{B}_0 \big\rangle + \big\langle \mathbf{H}_{k:\infty}, (\mathbf{I} - \gamma\mathbf{H})^N \mathbf{B}_0 \big\rangle \qquad\text{(A.7)}$$

for all $k \in [d]$. Further note that $\mathbf{B}_0 = (\mathbf{w}_0 - \mathbf{w}^*) \otimes (\mathbf{w}_0 - \mathbf{w}^*) = \mathbf{w}^* \otimes \mathbf{w}^*$ as we pick $\mathbf{w}_0 = \mathbf{0}$. Thus (A.7) implies that

$$\text{bias} \leq \frac{1}{N^2\gamma^2} \cdot \big\| (\mathbf{I} - \gamma\mathbf{H})^{N/2}\mathbf{w}^* \big\|_{\mathbf{H}_{0:k}^{-1}} + \big\| (\mathbf{I} - \gamma\mathbf{H})^{N/2}\mathbf{w}^* \big\|_{\mathbf{H}_{k:\infty}}^2.$$

Then we will deal with the variance error. Plugging (A.5) into (A.3) gives

$$\begin{aligned}
\text{variance} &\leq \frac{\sigma^2\gamma}{N^2} \sum_{t=0}^{N-1} \sum_{k=t}^{N-1} \big\langle (\mathbf{I} - \gamma\mathbf{H})^{k-t}\mathbf{H}, \mathbf{I} - (\mathbf{I} - \gamma\mathbf{H})^{N+t} \big\rangle \\
&\leq \frac{\sigma^2\gamma}{N^2} \sum_{t=0}^{N-1} \Big\langle \sum_{k=0}^{N-1} (\mathbf{I} - \gamma\mathbf{H})^k\mathbf{H}, \mathbf{I} - (\mathbf{I} - \gamma\mathbf{H})^{N+t} \Big\rangle \\
&= \frac{\sigma^2}{N^2} \sum_{t=0}^{N-1} \Big\langle \mathbf{I} - (\mathbf{I} - \gamma\mathbf{H})^N, \mathbf{I} - (\mathbf{I} - \gamma\mathbf{H})^{N+t} \Big\rangle \\
&\leq \frac{\sigma^2}{N} \Big\langle \mathbf{I} - (\mathbf{I} - \gamma\mathbf{H})^{2N}, \mathbf{I} - (\mathbf{I} - \gamma\mathbf{H})^{2N} \Big\rangle.
\end{aligned}$$

We then use the inequality $(1 - x)^N \geq \min\{0, 1 - xN\}$ again and thus the above inequality further leads to

$$\begin{aligned}
\text{variance} &\leq \frac{\sigma^2}{N} \cdot \sum_i \min\{1, 4N^2\gamma^2\lambda_i^2\} \\
&\leq \frac{4\sigma^2}{N} \cdot \Big(k + N^2\gamma^2 \sum_{i>k} \lambda_i^2\Big)
\end{aligned}$$

for any $k \in [d]$. $\qquad\square$

## A.2 Excess risk bound of ridge regression

**Lemma A.4.** *Let $\mathbf{X} \in \mathbb{R}^{N \times d}$ be the training data matrix and $\mathbf{w}_{\text{ridge}}(N; \lambda)$ be the solution of ridge regression with parameter $\lambda$ and sample size $N$, then for any $\lambda > 0$*

$$\mathbb{E}[L(\mathbf{w}_{\text{ridge}}(N; \lambda))] - L(\mathbf{w}^*) = \text{bias} + \text{variance},$$

*where*

$$\text{bias} = \lambda^2 \cdot \mathbb{E}\big[\mathbf{w}^{*\top}(\mathbf{X}^\top\mathbf{X} + \lambda\mathbf{I})^{-1}\mathbf{H}(\mathbf{X}^\top\mathbf{X} + \lambda\mathbf{I})^{-1}\mathbf{w}^*\big]$$
$$\text{variance} = \sigma^2 \cdot \mathbb{E}\big[\text{tr}\big((\mathbf{X}^\top\mathbf{X} + \lambda\mathbf{I})^{-1}\mathbf{X}^\top\mathbf{X}(\mathbf{X}^\top\mathbf{X} + \lambda\mathbf{I})^{-1}\mathbf{H}\big)\big],$$

*where the expectations are taken over the randomness of the training data matrix $\mathbf{X}$.*

*Proof.* Recall that the solution of ridge regression takes form

$$\mathbf{w}_{\text{ridge}}(N; \lambda) = (\mathbf{X}^\top\mathbf{X} + \lambda\mathbf{I})^{-1}\mathbf{X}^\top\mathbf{y},$$

where $\mathbf{X}$ is the data matrix and $\mathbf{y}$ is the response vector. Then according to the definition of the loss function $L(\mathbf{w})$, we have

$$
\begin{aligned}
\mathbb{E}[L(\mathbf{w}_{\mathrm{ridge}}(N;\lambda))] &= \mathbb{E}\Big[\big(y - \langle \mathbf{w}_{\mathrm{ridge}}(N;\lambda), \mathbf{x}\rangle\big)^2\Big] \\
&= \mathbb{E}\Big[\big(\langle \mathbf{w}^*, \mathbf{x}\rangle - \langle \mathbf{w}_{\mathrm{ridge}}(N;\lambda), \mathbf{x}\rangle\big)^2\Big] + \mathbb{E}\Big[\big(y - \langle \mathbf{w}^*, \mathbf{x}\rangle\big)^2\Big] \\
&\quad + 2\mathbb{E}\Big[\big(\langle \mathbf{w}^*, \mathbf{x}\rangle - \langle \mathbf{w}_{\mathrm{ridge}}(N;\lambda), \mathbf{x}\rangle\big) \cdot \big(y - \langle \mathbf{w}^*, \mathbf{x}\rangle\big)\Big] \\
&= \mathbb{E}[\|\mathbf{w}_{\mathrm{ridge}}(N;\lambda) - \mathbf{w}^*\|_{\mathbf{H}}^2] + L(\mathbf{w}^*),
\end{aligned}
$$

where the last equation is by Assumption 3.1. Then regarding $\mathbb{E}[\|\mathbf{w}_{\mathrm{ridge}}(N;\lambda) - \mathbf{w}^*\|_{\mathbf{H}}^2]$, let $\boldsymbol{\xi} = \mathbf{y} - \mathbf{X}\mathbf{w}^*$ be the model noise vector, we have

$$
\begin{aligned}
\mathbb{E}[\|\mathbf{w}_{\mathrm{ridge}}(N;\lambda) - \mathbf{w}^*\|_{\mathbf{H}}^2] &= \mathbb{E}\big[\big\|(\mathbf{X}^\top\mathbf{X} + \lambda\mathbf{I})^{-1}\mathbf{X}^\top\mathbf{y} - \mathbf{w}^*\big\|_{\mathbf{H}}^2\big] \\
&= \mathbb{E}\big[\big\|(\mathbf{X}^\top\mathbf{X} + \lambda\mathbf{I})^{-1}\mathbf{X}^\top(\mathbf{X}\mathbf{w}^* + \boldsymbol{\xi}) - \mathbf{w}^*\big\|_{\mathbf{H}}^2\big] \\
&= \underbrace{\mathbb{E}\big[\big\|(\mathbf{X}^\top\mathbf{X} + \lambda\mathbf{I})^{-1}\mathbf{X}^\top\mathbf{X}\mathbf{w}^* - \mathbf{w}^*\big\|_{\mathbf{H}}^2\big]}_{\text{bias}} + \underbrace{\mathbb{E}\big[\big\|(\mathbf{X}^\top\mathbf{X} + \lambda\mathbf{I})^{-1}\mathbf{X}^\top\boldsymbol{\xi}\big\|_{\mathbf{H}}^2\big]}_{\text{variance}}.
\end{aligned}
$$

where in the last inequality we again apply Assumption 3.1 that $\mathbb{E}[\boldsymbol{\xi}|\mathbf{X}] = \mathbf{0}$. More specifically, the bias error can be reformulated as

$$
\begin{aligned}
\text{bias} &= \mathbb{E}\big[\big\|\big((\mathbf{X}^\top\mathbf{X} + \lambda\mathbf{I})^{-1}\mathbf{X}^\top\mathbf{X} - \mathbf{I}\big)\mathbf{w}^*\big\|_{\mathbf{H}}^2\big] \\
&= \lambda^2 \mathbb{E}\big[\big\|(\mathbf{X}^\top\mathbf{X} + \lambda\mathbf{I})^{-1}\mathbf{w}\big\|_{\mathbf{H}}^2\big] \\
&= \lambda^2 \mathbb{E}\big[\mathbf{w}^{*\top}(\mathbf{X}^\top\mathbf{X} + \lambda\mathbf{I})^{-1}\mathbf{H}(\mathbf{X}^\top\mathbf{X} + \lambda\mathbf{I})^{-1}\mathbf{w}^*\big].
\end{aligned}
$$

In terms of the variance error, note that by Assumption 3.1 we have $\mathbb{E}[\boldsymbol{\xi}\boldsymbol{\xi}^\top|\mathbf{X}] = \sigma^2\mathbf{I}$, then

$$
\begin{aligned}
\text{variance} &= \mathbb{E}\big[\big\|(\mathbf{X}^\top\mathbf{X} + \lambda\mathbf{I})^{-1}\mathbf{X}^\top\boldsymbol{\epsilon}\big\|_{\mathbf{H}}^2\big] \\
&= \mathbb{E}\big[\operatorname{tr}\big((\mathbf{X}^\top\mathbf{X} + \lambda\mathbf{I})^{-1}\mathbf{X}^\top\boldsymbol{\xi}\boldsymbol{\xi}^\top\mathbf{X}(\mathbf{X}^\top\mathbf{X} + \lambda\mathbf{I})^{-1}\mathbf{H}\big)\big] \\
&= \sigma^2 \cdot \mathbb{E}\big[\operatorname{tr}\big((\mathbf{X}^\top\mathbf{X} + \lambda\mathbf{I})^{-1}\mathbf{X}^\top\mathbf{X}(\mathbf{X}^\top\mathbf{X} + \lambda\mathbf{I})^{-1}\mathbf{H}\big)\big].
\end{aligned}
$$

$\square$

**Lemma A.5.** *The solution of ridge regression with sample size $N$ and regularization parameter $\lambda$ satisfies*

$$
\mathbb{E}[L(\mathbf{w}_{\mathrm{ridge}}(N;\lambda))] - L(\mathbf{w}^*) = \mathrm{RidgeBias} + \mathrm{RidgeVariance},
$$

*where*

$$
\begin{aligned}
\mathrm{RidgeBias} &\gtrsim \max\left\{\sum_i (1-\lambda_i)^N \cdot \lambda_i\mathbf{w}^*[i]^2, \sum_{i=1}^{k^*}\frac{\lambda^2\lambda_i\mathbf{w}^*[i]^2}{(N\lambda_i + \lambda)^2} + \sum_{i>k^*}\lambda_i\mathbf{w}^*[i]^2\right\} \\
\mathrm{RidgeVariance} &\gtrsim \sigma^2 \cdot \left(\sum_{i=1}^{k^*}\frac{N\lambda_i^2}{(N\lambda_i + \lambda)^2} + \sum_{i>k^*}\frac{N\lambda_i^2}{(1+\lambda)^2}\right),
\end{aligned}
$$

*where $k^* = \min\{k : N\lambda_k \le 1\}$.*

*Proof.* In the one-hot case, it is easy to verify that $\mathbf{X}^\top\mathbf{X} = \sum_{i=1}^n \mathbf{x}_i\mathbf{x}_i^\top$ is a diagonal matrix. Let $\mu_1, \mu_2, \ldots, \mu_d$ be the eigenvalues of $\mathbf{X}^\top\mathbf{X}$ corresponding to the eigenvectors $\mathbf{e}_1, \mathbf{e}_2, \ldots, \mathbf{e}_d$ respectively. Then by Lemma A.4, we have the following results for the bias and variance errors of ridge regression.

$$
\begin{aligned}
\mathrm{RidgeBias} &= \lambda^2 \cdot \mathbb{E}\big[\mathbf{w}^{*\top}(\mathbf{X}^\top\mathbf{X} + \lambda\mathbf{I})^{-1}\mathbf{H}(\mathbf{X}^\top\mathbf{X} + \lambda\mathbf{I})^{-1}\mathbf{w}^*\big] \\
&= \lambda^2 \sum_i \mathbb{E}_{\mu_i}\left[\frac{\lambda_i\mathbf{w}^*[i]^2}{(\mu_i + \lambda)^2}\right],
\end{aligned} \tag{A.8}
$$

where the expectation in the first equation is taken over the training data $\mathbf{X}$ and in the second inequality the expectation is equivalently taken over the eigenvalues $\mu_1, \ldots, \mu_d$. Since $\mathbf{x}_i$ can only take on natural basis, the eigenvalue $\mu_i$ can be understood as the number of training data that equals $\mathbf{e}_i$. Note that the probability of sampling $\mathbf{e}_i$ is $\lambda_i$, then we can get that $\mu_i$ has a marginal distribution $\mathrm{Binom}(N, \lambda_i)$, where $N$ is the sample size. Then in terms of each expectation in (A.8), we first have

$$\mathbb{E}_{\mu_i}\left[\frac{\lambda_i \mathbf{w}^*[i]^2}{(\mu_i + \lambda)^2}\right] \geq \frac{\lambda_i \mathbf{w}^*[i]^2}{(\mathbb{E}[\mu_i] + \lambda)^2} = \frac{\lambda_i \mathbf{w}^*[i]^2}{(N\lambda_i + \lambda)^2},$$

where the first inequality is by applying Jensen's inequality to the convex function $f(x) = 1/(x+\lambda)^2$. On the other hand, we also have

$$\mathbb{E}_{\mu_i}\left[\frac{\lambda_i \mathbf{w}^*[i]^2}{(\mu_i + \lambda)^2}\right] \geq \frac{\lambda_i \mathbf{w}^*[i]^2}{\lambda^2} \cdot \mathbb{P}(\mu_i = 0) = \frac{\lambda_i \mathbf{w}^*[i]^2}{\lambda^2} \cdot (1 - \lambda_i)^N.$$

Therefore, combining the above two lower bounds, we can get the following lower bound on the bias error by (A.8)

$$\mathrm{RidgeBias} = \lambda^2 \sum_i \mathbb{E}_{\mu_i}\left[\frac{\lambda_i \mathbf{w}^*[i]^2}{(\mu_i + \lambda)^2}\right] \geq \sum_i \max\left\{\frac{\lambda^2 \lambda_i \mathbf{w}^*[i]^2}{(N\lambda_i + \lambda)^2}, \lambda_i \mathbf{w}^*[i]^2 \cdot (1 - \lambda_i)^N\right\}. \quad \text{(A.9)}$$

Therefore, a trivial lower bound on the bias error of ridge regression is

$$\mathrm{RidgeBias} \geq \sum_i (1 - \lambda_i)^N \cdot \lambda_i \mathbf{w}^*[i]^2.$$

Additionally, note that $(1 - \lambda_i)^N \geq 0.25$ if $\lambda_i \leq 1/N$ and $N \geq 2$. Then let $k^* = \min\{k : N\lambda_k \leq 1\}$, (A.9) further leads to

$$\mathrm{RidgeBias} \geq \sum_{i=1}^{k^*} \frac{\lambda^2 \lambda_i \mathbf{w}^*[i]^2}{(N\lambda_i + \lambda)^2} + 0.25 \cdot \sum_{i > k^*} \lambda_i \mathbf{w}^*[i]^2.$$

This completes the proof of the lower bound of the bias error.

By Lemma A.4, we have

$$\mathrm{RidgeVariance} = \sigma^2 \cdot \mathbb{E}\big[\,\mathrm{tr}\big((\mathbf{X}^\top \mathbf{X} + \lambda\mathbf{I})^{-1}\mathbf{X}^\top \mathbf{X}(\mathbf{X}^\top \mathbf{X} + \lambda\mathbf{I})^{-1}\mathbf{H}\big)\big]$$

$$= \sigma^2 \cdot \sum_i \mathbb{E}_{\mu_i}\left[\frac{\lambda_i \mu_i}{(\mu_i + \lambda)^2}\right], \quad \text{(A.10)}$$

Regarding the variance error, we cannot use the similar approach since the function $g(x) = x/(x+\lambda)^2$ is no longer convex. Instead, we will directly make use of property of the binomial distribution of $\mu_i$ to prove the desired bound. In particular, note that $\mu_i \sim \mathrm{binom}(N, \lambda_i)$, by Bernstein inequality, we have

$$\mathbb{P}(|\mu_i - N\lambda_i| \leq t) \geq 1 - 2\exp\left(-\frac{t^2}{2(N\lambda_i + t/3)}\right).$$

If $N\lambda_i \geq 6$, by set $t = \sqrt{3N\lambda_i}$, we have

$$\mathbb{P}\big(\mu_i \in \big[N\lambda_i - \sqrt{3N\lambda_i}, N\lambda_i + \sqrt{3N\lambda_i}\big]\big) \geq 1 - 2e^{-1} \geq 0.2,$$

which further implies that

$$\mathbb{P}\big(\mu_i \in \big[0.25N\lambda_i, 2N\lambda_i\big]\big) \geq 0.2,$$

where we use the fact that $\sqrt{3N\lambda_i} \leq 0.75N\lambda_i$ if $N\lambda_i > 6$. Therefore, in this case, we can get

$$\mathbb{E}_{\mu_i}\left[\frac{\lambda_i \mu_i}{(\mu_i + \lambda)^2}\right] \geq 0.2\min\left\{\frac{0.25N\lambda_i^2}{(0.25N\lambda_i + \lambda)^2}, \frac{2N\lambda_i^2}{(2N\lambda_i + \lambda)^2}\right\} \geq \frac{0.05N\lambda_i^2}{(N\lambda_i + \lambda)^2}. \quad \text{(A.11)}$$

Then we consider the case that $N\lambda_i < 6$. In particular, we have

$$\mathbb{E}_{\mu_i}\left[\frac{\lambda_i \mu_i}{(\mu_i + \lambda)^2}\right] \geq \frac{\lambda_i}{(1 + \lambda)^2} \cdot \mathbb{P}(\mu_i = 1). \quad \text{(A.12)}$$

Note that $\mu_i$ follows $\mathrm{Binom}(N, \lambda_i)$ distribution, which implies that

$$\mathbb{P}(\mu_i = 1) = N\lambda_i(1 - \lambda_i)^{N-1} \geq N\lambda_i\big(1 - \frac{6}{N}\big)^{N-1} \geq e^{-6}N\lambda_i.$$

Plugging this into (A.12) gives

$$\mathbb{E}_{\mu_i}\left[\frac{\lambda_i\mu_i}{(\mu_i + \lambda)^2}\right] \geq \frac{e^{-6}N\lambda_i^2}{(1 + \lambda)^2}. \tag{A.13}$$

Therefore, let $k^* = \min\{k : N\lambda_k \leq 1\}$, then for all $i \leq k^*$, combining (A.11) and (A.13) gives

$$\mathbb{E}_{\mu_i}\left[\frac{\lambda_i\mu_i}{(\mu_i + \lambda)^2}\right] \geq \frac{e^{-6}N\lambda_i^2}{(N\lambda_i + \lambda)^2}.$$

For all $i > k^*$, we can directly apply (A.13) to get the lower bound. Therefore, according to (A.10), the variance error can be lower bounded as follows,

$$\mathrm{RidgeVariance} = \sigma^2 \cdot \sum_i \mathbb{E}_{\mu_i}\left[\frac{\lambda_i\mu_i}{(\mu_i + \lambda)^2}\right]$$

$$\geq e^{-6}\sigma^2 \cdot \left(\sum_{i=1}^{k^*} \frac{N\lambda_i^2}{(N\lambda_i + \lambda)^2} + \sum_{i>k^*} \frac{N\lambda_i^2}{(1 + \lambda)^2}\right).$$

This completes the proof of the lower bound of the variance error. $\qquad\square$

## A.3   Proof of Theorem 4.1

*Proof.* In the beginning, we first recall the excess risk upper bound of SGD (see Lemma A.3) and excess risk lower bound of ridge (see Lemma A.3) as follows,

$$\mathbb{E}[L(\mathbf{w}_{\mathrm{sgd}}(N_{\mathrm{sgd}}; \gamma))] - L(\mathbf{w}^*) \leq 2 \cdot \mathrm{SGDBias} + 2 \cdot \mathrm{SGDVariance},$$

where

$$\mathrm{SGDBias} \lesssim \frac{1}{N^2\gamma^2} \cdot \left\|(\mathbf{I} - \gamma\mathbf{H})^{N/2}\mathbf{w}^*\right\|_{\mathbf{H}_{0:k_1}^{-1}} + \left\|(\mathbf{I} - \gamma\mathbf{H})^{N/2}\mathbf{w}^*\right\|_{\mathbf{H}_{k_1:\infty}}^2$$

$$\mathrm{SGDVariance} \lesssim \sigma^2 \cdot \left(\frac{k_2}{N} + N\gamma^2 \sum_{i>k_2} \lambda_i^2\right) \tag{A.14}$$

for arbitrary $k_1, k_2 \in [d]$.

$$\mathbb{E}[L(\mathbf{w}_{\mathrm{ridge}}(N; \lambda))] - L(\mathbf{w}^*) = \mathrm{RidgeBias} + \mathrm{RidgeVariance},$$

where

$$\mathrm{RidgeBias} \gtrsim \max\left\{\sum_i (1 - \lambda_i)^N \cdot \lambda_i\mathbf{w}^*[i]^2, \sum_{i=1}^{k^*} \frac{\lambda^2\lambda_i\mathbf{w}^*[i]^2}{(N\lambda_i + \lambda)^2} + \sum_{i>k^*} \lambda_i\mathbf{w}^*[i]^2\right\}$$

$$\mathrm{RidgeVariance} \gtrsim \sigma^2 \cdot \left(\sum_{i=1}^{k^*} \frac{N\lambda_i^2}{(N\lambda_i + \lambda)^2} + \sum_{i>k^*} \frac{N\lambda_i^2}{(1 + \lambda)^2}\right), \tag{A.15}$$

where $k^* = \min\{k : N\lambda_k \leq 1\}$.

Next, we will show that the excess risk of SGD can be provably upper bounded (up to constant factors) by the excess risk of ridge regression respectively, given the sample size of ridge regression $N_{\mathrm{ridge}}$ (which we will use $N$ in the remaining proof for simplicity). In particular, we consider two cases regarding different $\lambda$: **Case I** $\lambda < 1$ and **Case II** $\lambda \geq 1$.

For **Case I**, (A.15) gives the following bias lower bound for ridge regression,

$$\text{RidgeBias} \gtrsim \sum_i (1 - \lambda_i)^N \cdot \lambda_i \mathbf{w}^*[i]^2$$

$$\gtrsim \sum_{i > k^*} \lambda_i \mathbf{w}^*[i]^2$$

$$\text{RidgeVariance} \gtrsim \sigma^2 \cdot \left( \sum_{i=1}^{k^*} \frac{N\lambda_i^2}{(N\lambda_i + \lambda)^2} + \sum_{i > k^*} \frac{N\lambda_i^2}{(1+\lambda)^2} \right)$$

$$\overset{(i)}{\approx} \sigma^2 \cdot \left( \frac{k^*}{N} + N \sum_{i > k^*} \lambda_i^2 \right),$$

where in $(i)$ we use the fact that $N\lambda_i + \lambda \approx N\lambda_i$ for all $i \leq k^*$.

Then let $R^2 = \|\mathbf{w}^*\|_2^2 / \sigma^2$ denotes the signal-to-noise ratio, let's consider the following configuration for SGD:

$$N_{\text{sgd}} = N, \quad \gamma = 1.$$

Then by (A.14) and setting $k_1 = 0$ and $k_2 = k^*$, we get

$$\text{SGDBias} \lesssim \sum_i (1 - \lambda_i)^N \cdot \lambda_i \mathbf{w}^*[i]^2$$

$$\text{SGDVariance} \lesssim \sigma^2 \cdot \left( \frac{k^*}{N_{\text{sgd}}} + N_{\text{sgd}}\gamma^2 \sum_{i > k^*} \lambda_i^2 \right)$$

$$\overset{(i)}{\lesssim} \sigma^2 \cdot \left( \frac{k^*}{N} + N \sum_{i > k^*} \lambda_i^2 \right).$$

Therefore, given such choice of $N_{\text{sgd}}$ and $\gamma$, we have

$$\mathbb{E}[L(\mathbf{w}_{\text{sgd}}(N_{\text{sgd}}; \gamma))] - L(\mathbf{w}^*) \lesssim \text{SGDBias} + \text{SGDVariance}$$

$$\lesssim \sum_i (1 - \lambda_i)^N \cdot \lambda_i \mathbf{w}^*[i]^2 + \sigma^2 \cdot \left( \frac{k^*}{N} + N \sum_{i > k^*} \lambda_i^2 \right)$$

$$\lesssim \text{RidgeBias} + \text{RidgeVariance}$$

$$= \mathbb{E}[L(\mathbf{w}_{\text{ridge}(N; \lambda)})] - L(\mathbf{w}^*).$$

For **Case II**, we can define $\widetilde{k}^* = \min\{k : N\lambda_k \leq \lambda\}$, then (A.15) implies

$$\text{RidgeBias} \gtrsim \sum_{i=1}^{k^*} \frac{\lambda^2 \lambda_i \mathbf{w}^*[i]^2}{(N\lambda_i + \lambda)^2} + \sum_{i > k^*} \lambda_i \mathbf{w}^*[i]^2$$

$$\overset{(i)}{\approx} \sum_{i=1}^{\widetilde{k}^*} \frac{\lambda^2 \mathbf{w}^*[i]^2}{N^2 \lambda_i} + \sum_{i > \widetilde{k}^*} \lambda_i \mathbf{w}^*[i]^2$$

$$\text{RidgeVariance} \gtrsim \sigma^2 \cdot \left( \sum_{i=1}^{k^*} \frac{N\lambda_i^2}{(N\lambda_i + \lambda)^2} + \sum_{i > k^*} \frac{N\lambda_i^2}{(1+\lambda)^2} \right)$$

$$\overset{(ii)}{\approx} \sigma^2 \cdot \left( \frac{\widetilde{k}^*}{N} + \frac{N}{\lambda^2} \sum_{i > \widetilde{k}^*} \lambda_i^2 \right),$$

where $(i)$ and $(ii)$ are due to the fact that for every $i \leq k^*$, we have

$$\frac{1}{(N\lambda_i + \lambda)^2} \approx \begin{cases} \frac{1}{N^2 \lambda_i} & i \leq \widetilde{k}^* \\ \frac{1}{\lambda^2} & \widetilde{k}^* < i \leq k^*. \end{cases}$$

Therefore, we can apply the following configuration for SGD:

$$N_{\mathrm{sgd}} = N, \quad \gamma = 1/\lambda.$$

Then by (A.14) and set $k_1 = k_2 = \widetilde{k}^*$, we have

$$\mathbb{E}[L(\mathbf{w}_{\mathrm{sgd}}(N_{\mathrm{sgd}}; \gamma))] - L(\mathbf{w}^*)$$
$$\lesssim \mathrm{SGDBias} + \mathrm{SGDVariance}$$
$$\lesssim \sum_{i=1}^{\widetilde{k}^*} \frac{(1 - \gamma\lambda_i)^{N_{\mathrm{sgd}}} \mathbf{w}^*[i]^2}{\lambda_i N_{\mathrm{sgd}}^2 \gamma^2} + \sum_{i > \widetilde{k}^*} \lambda_i \mathbf{w}^*[i]^2 + \sigma^2 \cdot \left( \frac{\widetilde{k}^*}{N_{\mathrm{sgd}}} + N_{\mathrm{sgd}} \gamma^2 \sum_{i > \widetilde{k}^*} \lambda_i^2 \right)$$
$$\approx \sum_{i=1}^{\widetilde{k}^*} \frac{\lambda^2 \mathbf{w}^*[i]^2}{\lambda_i N^2} + \sum_{i > \widetilde{k}^*} \lambda_i \mathbf{w}^*[i]^2 + \sigma^2 \cdot \left( \frac{\widetilde{k}^*}{N} + \frac{N}{\lambda^2} \sum_{i > \widetilde{k}^*} \lambda_i^2 \right)$$
$$\lesssim \mathrm{RidgeBias} + \mathrm{RidgeVariance}$$
$$= \mathbb{E}[L(\mathbf{w}_{\mathrm{ridge}}(N; \lambda))] - L(\mathbf{w}^*).$$

Combining the results for these two cases completes the proof. $\square$

## A.4 Proof of Theorem 4.2

*Proof.* For simplicity we define $N := N_{\mathrm{sgd}}$ in the proof.

- The data covariance matrix $\mathbf{H}$ has the following spectrum

$$\lambda_i = \begin{cases} \frac{\log(N)}{N^{1/2}} & i = 1, \\ \frac{1 - \log(N)/N^{1/2}}{N} & 1 < i \leq N, \\ 0 & N < i \leq d \end{cases}$$

- The true parameter $\mathbf{w}^*$ is given by

$$\mathbf{w}^*[i] = \begin{cases} \sigma \cdot \sqrt{\frac{N^{1/2}}{\log(N)}} & i = 1, \\ 0 & 1 < i \leq d. \end{cases}$$

Then it is easy to verify that $\mathrm{tr}(\mathbf{H}) = 1$. For SGD, we consider setting the stepsize as $\gamma^* = N^{-1/2}$. Then by Lemma A.3 and choosing $k_1 = 1$, we have the following on the bias error of SGD,

$$\mathrm{SGDBias} \lesssim \sum_{i=1}^{k^*} \frac{(1 - \gamma\lambda_i)^{N_{\mathrm{sgd}}} \mathbf{w}^*[i]^2}{\lambda_i N_{\mathrm{sgd}}^2 \gamma^2} + \sum_{i > k^*} \lambda_i \mathbf{w}^*[i]^2 \lesssim \frac{(1 - \log(N)/N)^N \sigma^2}{\log^2(N)} \lesssim \frac{\sigma^2}{N}.$$

For variance error, we can pick $k_2 = 1$ and get

$$\mathrm{SGDVariance} \lesssim \sigma^2 \cdot \left( \frac{1}{N} + N\gamma^2 \sum_{i > 1} \lambda_i^2 \right) \lesssim \sigma^2 \left( \frac{1}{N} + \sum_{i > 1} \lambda_i^2 \right) \approx \frac{\sigma^2}{N}.$$

Now let us characterize the excess risk of ridge regression. In terms of the bias error, by Lemma A.5 we have

$$\mathrm{RidgeBias} \gtrsim \sum_{i=1}^{k^*} \frac{\lambda^2 \lambda_i \mathbf{w}^*[i]^2}{(N_{\mathrm{ridge}}\lambda_i + \lambda)^2} + \sum_{i > k^*} \lambda_i \mathbf{w}^*[i]^2 \approx \frac{\lambda^2 \sigma^2}{(N_{\mathrm{ridge}} \log(N)/N^{1/2} + \lambda)^2}, \quad (A.16)$$

where $k^* = \min\{k : N_{\mathrm{ridge}}\lambda_k \leq 1\}$. Then it is clear for ridge regression we must have $\lambda \lesssim N_{\mathrm{ridge}} \log(N)/N^{1/2}$ since otherwise $\mathrm{RidgeBias} \gtrsim \sigma^2 \gtrsim \mathrm{SGDRisk}$. Regarding the variance, we have

$$\mathrm{RidgeVariance} \gtrsim \sigma^2 \cdot \left( \sum_{i=1}^{k^*} \frac{N_{\mathrm{ridge}}\lambda_i^2}{(N_{\mathrm{ridge}}\lambda_i + \lambda)^2} + \sum_{i > k^*} \frac{N_{\mathrm{ridge}}\lambda_i^2}{(1 + \lambda)^2} \right).$$

Then we will consider two cases: (1) $N_{\text{ridge}} \lesssim N$ and (2) $N_{\text{ridge}} \gtrsim N$. In the first case we can get $k^* = 1$ and then

$$\text{RidgeVariance} \gtrsim \sigma^2 \cdot \left( \frac{N_{\text{ridge}} \log^2(N)/N^2}{(N_{\text{ridge}} \log(N)/N + \lambda)^2} + \frac{N_{\text{ridge}}}{N^2(1+\lambda)^2} \right) \geq \frac{N_{\text{ridge}} \sigma^2}{N^2(1+\lambda^2)}.$$

In this case, we can get $k^* = 1$ and thus

$$\text{RidgeVariance} \gtrsim \sigma^2 \cdot \frac{N_{\text{ridge}} \log^2(N)/N^2}{(N_{\text{ridge}} \log(N)/N + \lambda)^2} \overset{(i)}{\approx} \frac{\sigma^2}{N_{\text{ridge}}} \overset{(ii)}{\gtrsim} \frac{\sigma^2}{N},$$

where $(i)$ is due to we require $\lambda \lesssim N_{\text{ridge}} \log(N)/N^{1/2}$ to guarantee vanishing bias error and $(ii)$ is due to in this case we have $N_{\text{ridge}} \lesssim N$. As a result, ridge regression cannot achieve smaller excess risk than SGD in this case.

In the second case we can get $k^* = N$ and then

$$\begin{aligned}
\text{RidgeVariance} &\gtrsim \sigma^2 \cdot \left( \frac{N_{\text{ridge}} \log^2(N)/N^2}{(N_{\text{ridge}} \log(N)/N + \lambda)^2} + \frac{(k^* - 1) \cdot N_{\text{ridge}}/N^2}{(N_{\text{ridge}}/N + \lambda^2)} \right) \\
&\gtrsim \sigma^2 \cdot \frac{N N_{\text{ridge}}}{N_{\text{ridge}}^2 + N^2 \lambda^2},
\end{aligned} \tag{A.17}$$

where the second inequality is due to $k^* = N$. We will again consider two cases: (a) $N_{\text{ridge}} \gtrsim N\lambda$ and (b) $N_{\text{ridge}} \lesssim N\lambda$. Regarding Case (a) we have

$$\text{RidgeVariance} \geq \frac{N\sigma^2}{N_{\text{ridge}}},$$

and it is clear that for all $N_{\text{ridge}} \lesssim N^2$ we have $\text{RidgeVariance} \gtrsim \sigma^2/N \gtrsim \text{SGDRisk}$. Regarding Case (b), combining the lower bounds of bias (A.16) and variance (A.17) of ridge regression, we get

$$\text{RidgeRisk} \gtrsim \sigma^2 \cdot \left( \frac{\lambda^2 N}{N_{\text{ridge}}^2 \log^2(N)} + \frac{N_{\text{ridge}}}{N\lambda^2} \right) \gtrsim \frac{\sigma^2}{N_{\text{ridge}}^{1/2} \log(N)},$$

where the first inequality follows from the fact that $\lambda \lesssim N_{\text{ridge}} \log(N)/N^{1/2}$ and $N_{\text{ridge}} \lesssim N\lambda$, and the second inequality is by Cauchy-Schwartz inequality. This further suggests that $\text{RidgeRisk} \lesssim \sigma^2/N \lesssim \text{SGDRisk}$ if $N_{\text{ridge}} \leq N^2/\log^2(N)$, which completes the proof. $\qquad\square$

# B Proof of Gaussian Least Squares

## B.1 Excess risk bounds of SGD and ridge regression

We first recall the excess risk bounds for SGD (with tail averaging) and ridge regression as follows.

### SGD with tail averaging

**Theorem B.1** (Extension of Theorem 5.1 in Zou et al. [30]). *Consider SGD with tail-averaging with initialization $\mathbf{w}_0 = \mathbf{0}$. Suppose Assumption 5.1 holds and the stepsize satisfies $\gamma \lesssim 1/\text{tr}(\mathbf{H})$. Then the excess risk can be upper bounded as follows,*

$$\mathbb{E}[L(\mathbf{w}_{\text{sgd}}(N; \gamma))] - L(\mathbf{w}^*) \leq \text{SGDBias} + \text{SGDVariance},$$

*where*

$$\text{SGDBias} \lesssim \frac{1}{\gamma^2 N^2} \cdot \left\| (\mathbf{I} - \gamma\mathbf{H})^N \mathbf{w}^* \right\|_{\mathbf{H}_{0:k_1}^{-1}}^2 + \left\| (\mathbf{I} - \gamma\mathbf{H})^N \mathbf{w}^* \right\|_{\mathbf{H}_{k_1:\infty}}^2$$

$$\text{SGDVariance} \lesssim \frac{\sigma^2 + \|\mathbf{w}^*\|_{\mathbf{H}}^2}{N} \cdot \left( k_2 + N^2 \gamma^2 \sum_{i > k_2} \lambda_i^2 \right).$$

*where $k_1, k_2 \in [d]$ are arbitrary.*

This theorem is a simple extension of Theorem 5.1 in Zou et al. [30]. In particular, we observe that though the original theorem is stated for some particular $k^*$ and $k^\dagger$, based on the proof, their results hold for arbitrary $k_1$ and $k_2$, as stated in Theorem B.1.

**Ridge regression.** See Appendix C for a proof of the following theorem.

**Theorem B.2** (Extension of Lemmas 2 & 3 in Tsigler and Bartlett [26]). *Suppose Assumption 5.1 holds. Let $\lambda \geq 0$ be the regularization parameter, $n$ be the training sample size and $\widehat{\mathbf{w}}_{\mathrm{ridge}}(N; \lambda)$ be the output of ridge regression. Then*

$$\mathbb{E}\big[L(\mathbf{w}_{\mathrm{ridge}}(N; \lambda))\big] - L(\mathbf{w}^*) = \mathrm{RidgeBias} + \mathrm{RidgeVariance},$$

*and there is some absolute constant $b > 1$, such that for*

$$k^*_{\mathrm{ridge}} := \min\left\{ k : b\lambda_{k+1} \leq \frac{\lambda + \sum_{i>k} \lambda_i}{n} \right\},$$

*the following holds:*

$$\mathrm{RidgeBias} \gtrsim \left( \frac{\lambda + \sum_{i > k^*_{\mathrm{ridge}}} \lambda_i}{N} \right)^2 \cdot \|\mathbf{w}^*\|^2_{\mathbf{H}^{-1}_{0:k^*_{\mathrm{ridge}}}} + \|\mathbf{w}^*\|^2_{\mathbf{H}_{k^*_{\mathrm{ridge}}:\infty}},$$

$$\mathrm{RidgeVariance} \gtrsim \sigma^2 \cdot \left\{ \frac{k^*_{\mathrm{ridge}}}{N} + \frac{N \sum_{i > k^*_{\mathrm{ridge}}} \lambda_i^2}{\left( \lambda + \sum_{i > k^*_{\mathrm{ridge}}} \lambda_i \right)^2} \right\}.$$

## B.2 Proof of Theorem 5.1

*Proof.* For simplicity, let us fix $N := N_{\mathrm{ridge}}$ and $k := k_{\mathrm{ridge}}$, we will next locate $\gamma$ such that the risk of SGD competes with that of Ridge. Denote $\widetilde{\lambda} := \lambda + \sum_{i>k} \lambda_i$. Then

$$\mathrm{RidgeRisk} = \mathrm{RidgeBias} + \mathrm{RidgeVariance}$$

$$\gtrsim \left( \frac{\widetilde{\lambda}}{N} \right)^2 \|\mathbf{w}^*\|^2_{\mathbf{H}^{-1}_{0:k}} + \|\mathbf{w}^*\|^2_{\mathbf{H}_{k:\infty}} + \frac{\sigma^2}{N} \left( k + \left( \frac{N}{\widetilde{\lambda}} \right)^2 \sum_{i>k} \lambda_i^2 \right).$$

Then for SGD we can set

$$N_{\mathrm{sgd}} = (1 + R^2) \cdot N \cdot (1 \vee \kappa \log a),$$

where

$$\kappa := \frac{\mathrm{tr}(\mathbf{H})}{N\lambda_N}, \qquad a = \frac{\mathrm{tr}(\mathbf{H})}{\lambda + \sum_{i>N} \lambda_i} \wedge (\kappa R \sqrt{N}) = \frac{\mathrm{tr}(\mathbf{H})}{\lambda + \sum_{i>N} \lambda_i} \wedge \frac{\mathrm{tr}(\mathbf{H})R}{\sqrt{N}\lambda_N}.$$

Next we discuss two cases:

**Case I, $\widetilde{\lambda} \cdot (1 \vee \kappa \log a) \geq \mathrm{tr}(\mathbf{H})$.** For SGD, let us set $k_{\mathrm{sgd}} = k$ and that

$$\gamma = \frac{1}{(1 + R^2) \cdot \widetilde{\lambda} \cdot (1 \vee \kappa \log a)} \leq \frac{1}{\mathrm{tr}(\mathbf{H})},$$

then

$$N_{\mathrm{sgd}} \cdot \gamma = \frac{N}{\widetilde{\lambda}}.$$

Thus we obtain that

$$\mathrm{SGDRisk} \lesssim \frac{(1 - \gamma\lambda_k)^{2N_{\mathrm{sgd}}}}{(\gamma N_{\mathrm{sgd}})^2} \|\mathbf{w}^*\|^2_{\mathbf{H}^{-1}_{0:k}} + \|\mathbf{w}^*\|^2_{\mathbf{H}_{k:\infty}} + \frac{(1 + R^2)\sigma^2}{N_{\mathrm{sgd}}} \left( k + (\gamma N_{\mathrm{sgd}})^2 \sum_{i>k} \lambda_i^2 \right)$$

$$= \frac{(1 - \gamma\lambda_k)^{2N_{\mathrm{sgd}}}}{\left( N/\widetilde{\lambda} \right)^2} \|\mathbf{w}^*\|^2_{\mathbf{H}^{-1}_{0:k}} + \|\mathbf{w}^*\|^2_{\mathbf{H}_{k:\infty}} + \frac{\sigma^2}{N(1 \vee \kappa \log a)} \left( k + \left( N/\widetilde{\lambda} \right)^2 \sum_{i>k} \lambda_i^2 \right)$$

$$\leq \left( \frac{\widetilde{\lambda}}{N} \right)^2 \|\mathbf{w}^*\|^2_{\mathbf{H}^{-1}_{0:k}} + \|\mathbf{w}^*\|^2_{\mathbf{H}_{k:\infty}} + \frac{\sigma^2}{N} \left( k + \left( \frac{N}{\widetilde{\lambda}} \right)^2 \sum_{i>k} \lambda_i^2 \right)$$

$$\lesssim \mathrm{RidgeRisk}.$$

**Case II, $\widetilde{\lambda} \cdot (1 \vee \kappa \log a) < \operatorname{tr}(\mathbf{H})$.** For SGD, let us set $k_{\text{sgd}} = k$ and that

$$\gamma = \frac{1}{(1 + R^2) \cdot \operatorname{tr}(\mathbf{H})} \leq \frac{1}{\operatorname{tr}(\mathbf{H})},$$

then

$$N_{\text{sgd}} \cdot \gamma = \frac{N \cdot (1 \vee \kappa \log a)}{\operatorname{tr}(\mathbf{H})} \leq \frac{N}{\widetilde{\lambda}}.$$

We obtain that

$\operatorname{SGDRisk} \leq \operatorname{SGDBias} + \operatorname{SGDVariance}$

$$\lesssim \frac{(1 - \gamma \lambda_k)^{2N_{\text{sgd}}}}{(\gamma N_{\text{sgd}})^2} \|\mathbf{w}^*\|_{\mathbf{H}_{0:k}^{-1}}^2 + \|\mathbf{w}^*\|_{\mathbf{H}_{k:\infty}}^2 + \frac{(1 + R^2)\sigma^2}{N_{\text{sgd}}} \left( k + (\gamma N_{\text{sgd}})^2 \sum_{i>k} \lambda_i^2 \right)$$

$$\leq \frac{(1 - \gamma \lambda_k)^{2N_{\text{sgd}}}}{(\gamma N_{\text{sgd}})^2} \|\mathbf{w}^*\|_{\mathbf{H}_{0:k}^{-1}}^2 + \|\mathbf{w}^*\|_{\mathbf{H}_{k:\infty}}^2 + \frac{\sigma^2}{N(1 \vee \kappa \log a)} \left( k + \left( N/\widetilde{\lambda} \right)^2 \sum_{i>k} \lambda_i^2 \right)$$

$$\leq \frac{(1 - \gamma \lambda_k)^{2N_{\text{sgd}}}}{(\gamma N_{\text{sgd}})^2} \|\mathbf{w}^*\|_{\mathbf{H}_{0:k}^{-1}}^2 + \|\mathbf{w}^*\|_{\mathbf{H}_{k:\infty}}^2 + \frac{\sigma^2}{N} \left( k + \left( \frac{N}{\widetilde{\lambda}} \right)^2 \sum_{i>k} \lambda_i^2 \right).$$

The second and the third terms match those of ridge error. As for the first term, notice that by the choice of $\gamma$ and that $\lambda_k \geq \lambda_N$, we have that

$$\frac{(1 - \gamma \lambda_k)^{N_{\text{sgd}}}}{\gamma N_{\text{sgd}}} \leq \left( 1 - \frac{\lambda_N}{(1 + R^2) \cdot \operatorname{tr}(\mathbf{H})} \right)^{N_{\text{sgd}}} \cdot \frac{1}{\gamma N_{\text{sgd}}}$$

$$= \left( 1 - \frac{1}{(1 + R^2) \cdot N \cdot \kappa} \right)^{(1 + R^2) \cdot N \cdot (1 \vee \kappa \log a)} \cdot \frac{\operatorname{tr}(\mathbf{H})}{N \cdot (1 \vee \kappa \log a)}$$

$$\leq \left( 1 - \frac{1}{(1 + R^2) \cdot N \cdot \kappa} \right)^{(1 + R^2) \cdot N \cdot \kappa \log a} \cdot \frac{\operatorname{tr}(\mathbf{H})}{N}$$

$$\leq \frac{1}{a} \cdot \frac{\operatorname{tr}(\mathbf{H})}{N} = \frac{(\lambda + \sum_{i>N} \lambda_i) \vee (\sqrt{N} \lambda_N / R)}{\operatorname{tr}(\mathbf{H})} \cdot \frac{\operatorname{tr}(\mathbf{H})}{N}$$

$$\leq \frac{\lambda + \sum_{i>k} \lambda_i}{N} \vee \frac{\lambda_k}{R \cdot \sqrt{N}} = \frac{\widetilde{\lambda}}{N} \vee \frac{\lambda_k}{R \cdot \sqrt{N}}.$$

If $\frac{(1 - \gamma \lambda_k)^{N_{\text{sgd}}}}{\gamma N_{\text{sgd}}} \leq \frac{\widetilde{\lambda}}{N}$, then

$$\operatorname{SGDRisk} \lesssim \frac{(1 - \gamma \lambda_k)^{2N_{\text{sgd}}}}{(\gamma N_{\text{sgd}})^2} \|\mathbf{w}^*\|_{\mathbf{H}_{0:k}^{-1}}^2 + \|\mathbf{w}^*\|_{\mathbf{H}_{k:\infty}}^2 + \frac{\sigma^2}{N} \left( k + \left( \frac{N}{\widetilde{\lambda}} \right)^2 \sum_{i>k} \lambda_i^2 \right)$$

$$\leq \left( \frac{\widetilde{\lambda}}{N} \right)^2 \|\mathbf{w}^*\|_{\mathbf{H}_{0:k}^{-1}}^2 + \|\mathbf{w}^*\|_{\mathbf{H}_{k:\infty}}^2 + \frac{\sigma^2}{N} \left( k + \left( \frac{N}{\widetilde{\lambda}} \right)^2 \sum_{i>k} \lambda_i^2 \right)$$

$$\lesssim \operatorname{RidgeRisk}.$$

If $\frac{(1 - \gamma \lambda_k)^{N_{\text{sgd}}}}{\gamma N_{\text{sgd}}} \leq \frac{\lambda_k}{R \cdot \sqrt{N}}$, then

$$\frac{(1 - \gamma \lambda_k)^{2N_{\text{sgd}}}}{(\gamma N_{\text{sgd}})^2} \|\mathbf{w}^*\|_{\mathbf{H}_{0:k}^{-1}}^2 \leq \frac{\lambda_k^2}{R^2 \cdot N} \|\mathbf{w}^*\|_{\mathbf{H}_{0:k}^{-1}}^2 \leq \frac{\|\mathbf{w}^*\|_{\mathbf{H}}^2}{R^2 \cdot N} \leq \frac{\sigma^2}{N},$$

and

$$\operatorname{SGDRisk} \lesssim \frac{(1 - \gamma \lambda_k)^{2N_{\text{sgd}}}}{(\gamma N_{\text{sgd}})^2} \|\mathbf{w}^*\|_{\mathbf{H}_{0:k}^{-1}}^2 + \|\mathbf{w}^*\|_{\mathbf{H}_{k:\infty}}^2 + \frac{\sigma^2}{N} \left( k + \left( \frac{N}{\widetilde{\lambda}} \right)^2 \sum_{i>k} \lambda_i^2 \right)$$

$$\leq \frac{\sigma^2}{N} + \|\mathbf{w}^*\|_{\mathbf{H}_{k:\infty}}^2 + \frac{\sigma^2}{N} \left( k + \left( \frac{N}{\widetilde{\lambda}} \right)^2 \sum_{i>k} \lambda_i^2 \right)$$

$$\lesssim 2 \cdot \operatorname{RidgeRisk}.$$

These complete the proof. $\qquad\square$

## B.3 Proof of Corollary 5.1

*Proof.* By Theorem 5.1, we only need to verify that $\kappa(N_{\text{ridge}}) \lesssim \log(N_{\text{ridge}})$. Recall that $\lambda_i = 1/i^\alpha$ for $0 < \alpha \leq 1$, and $d \lesssim N_{\text{ridge}}$. For $\alpha = 1$, then

$$\text{tr}(\mathbf{H}) = \sum_{i=1}^{d} i^{-\alpha} \lesssim \log d \lesssim \log(N_{\text{ridge}}),$$

thus

$$\kappa(N_{\text{ridge}}) = \frac{\text{tr}(\mathbf{H})}{N_{\text{ridge}} \lambda_{\min\{d, N_{\text{ridge}}\}}} \lesssim \frac{\log(N_{\text{ridge}})}{N_{\text{ridge}} \cdot N_{\text{ridge}}^{-1}} = \log(N_{\text{ridge}}).$$

For $\alpha < 1$, then

$$\text{tr}(\mathbf{H}) = \sum_{i=1}^{d} i^{-\alpha} \lesssim d^{1-\alpha} \lesssim N_{\text{ridge}}^{1-\alpha},$$

thus

$$\kappa(N_{\text{ridge}}) = \frac{\text{tr}(\mathbf{H})}{N_{\text{ridge}} \lambda_{\{N_{\text{ridge}}, d\}}} \lesssim \frac{N_{\text{ridge}}^{1-\alpha}}{N_{\text{ridge}} \cdot N_{\text{ridge}}^{-\alpha}} = 1.$$

$\square$

## B.4 Proof of Corollary 5.2

*Proof.* Note that given random $\mathbf{w}^*$, the expected risk considered in our paper will be including the expectation over both random data $\mathbf{x}$ and random ground-truth $\mathbf{w}^*$. Since the distribution of $\mathbf{w}^*$ is rotation invariant, the expectation of $\mathbf{w}^*[i]$ will be the same for all $i \in [d]$. Therefore, let $B = \mathbb{E}[(\mathbf{w}^*[i])^2]$, the following holds according to (6.2)

$$\text{RidgeRisk} \gtrsim \left(\frac{\widetilde{\lambda}}{N_{\text{ridge}}}\right)^2 \cdot \mathbb{E}\big[\|\mathbf{w}^*\|_{\mathbf{H}_{0:k^*}^{-1}}^2\big] + \mathbb{E}\big[\|\mathbf{w}^*\|_{\mathbf{H}_{k^*:\infty}}^2\big] + \sigma^2 \cdot \left(\frac{k^*}{N_{\text{ridge}}} + \frac{N_{\text{ridge}}}{\widetilde{\lambda}^2} \sum_{i>k^*} \lambda_i^2\right)$$

$$= B\left(\frac{\widetilde{\lambda}}{N_{\text{ridge}}}\right)^2 \cdot \sum_{i=1}^{k^*} i^\alpha + B \cdot \sum_{i=k^*+1} i^{-\alpha} + \sigma^2 \cdot \left(\frac{k^*}{N_{\text{ridge}}} + \frac{N_{\text{ridge}}}{\widetilde{\lambda}^2} \sum_{i>k^*} \lambda_i^2\right)$$

where $k^* = \min\{k : N_{\text{ridge}} \lambda_k \leq \widetilde{\lambda}\}$. Then note that $\lambda_i = i^{-\alpha}$, we have $k^* = (N_{\text{ridge}}/\widetilde{\lambda})^{1/\alpha}$, which implies that

$$\text{RidgeRisk} \gtrsim B\left(\frac{\widetilde{\lambda}}{N_{\text{ridge}}}\right)^2 \cdot (k^*)^{1+\alpha} + B \cdot \big[d^{1-\alpha} - (k^*)^{1-\alpha}\big] + \sigma^2 \cdot \left(\frac{k^*}{N_{\text{ridge}}} + \frac{N_{\text{ridge}}}{\widetilde{\lambda}^2} \sum_{i>k^*} \lambda_i^2\right)$$

$$\gtrsim N_{\text{ridge}}^{1-\alpha} \cdot B$$

where we use the fact that $d = \Theta(N)$. Note that constant SNR $R = \Theta(1)$ implies that

$$\sigma^2 \asymp B \sum_{i=1}^{d} \lambda_i \asymp N_{\text{ridge}}^{1-\alpha} B.$$

Then by (6.1) and set $N_{\text{sgd}} = N_{\text{ridge}} = N$ and $k_1 = k_2 = N_{\text{ridge}}$, we have

$$\text{SGDRisk} \lesssim \frac{1}{\gamma^2 N_{\text{sgd}}^2} \cdot \mathbb{E}\big[\|\exp(-N_{\text{sgd}}\gamma\mathbf{H})\mathbf{w}^*\|_{\mathbf{H}_{0:k_1}^{-1}}^2\big] + \mathbb{E}\big[\|\mathbf{w}^*\|_{\mathbf{H}_{k_1:\infty}}^2\big]$$

$$+ (1+R^2)\sigma^2 \cdot \left(\frac{k_2}{N_{\text{sgd}}} + N_{\text{sgd}}\gamma^2 \sum_{i>k_2} \lambda_i^2\right)$$

$$= \frac{1}{\gamma^2 N^2} \cdot \mathbb{E}\big[\|\mathbf{w}^*\|_{\mathbf{H}_{0:N}^{-1}}^2\big] + \mathbb{E}\big[\|\mathbf{w}^*\|_{\mathbf{H}_{N:d}}^2\big] + BN^{1-\alpha} \cdot \left(1 + N\gamma^2 \sum_{i>N} \lambda_i^2\right).$$

Note that we have

$$\mathbb{E}\big[\|\mathbf{w}^*\|^2_{\mathbf{H}^{-1}_{0:N}}\big] = BN^{1+\alpha}, \ \mathbb{E}\big[\|\mathbf{w}^*\|^2_{\mathbf{H}_{N:d}}\big] = BN^{1-\alpha}.$$

Then we can set $\gamma \asymp 1/\operatorname{tr}(\mathbf{H}) \asymp N^{\alpha-1}$ and get

$$\text{SGDRisk} \lesssim \frac{B}{N^{2\alpha}} \cdot N^{1+\alpha} + BN^{1-\alpha} + BN^{1-\alpha} \cdot \left(1 + N\gamma^2 \sum_{i>N} \lambda_i^2\right)$$

$$\lesssim BN^{1-\alpha}$$

$$\lesssim \text{RidgeRisk}.$$

This implies that SGD can be no worse than ridge regression as long as provided same or larger sample size, which completes the proof.

$\square$

### B.5  Proof of Theorem 5.2

*Proof.* For simplicity we fix $N := N_{\mathrm{sgd}}$. Let us consider the following problem instance:

- The data covariance matrix $\mathbf{H}$ has the following spectrum

$$\lambda_i = \begin{cases} 1 & i = 1, \\ \frac{1}{N \log N} & 1 < i \le N^2, \\ 0 & N^2 < i \le d \end{cases}$$

where we require the dimension $d \ge N^2$. We note that $\operatorname{tr}(\mathbf{H}) = 1 + N/\log N \asymp N/\log N$.

- The true parameter $\mathbf{w}^*$ is given by

$$\mathbf{w}^*[i] = \begin{cases} \sigma & i = 1, \\ 0 & 1 < i \le d. \end{cases}$$

Then for SGD, we choose stepsize as $\gamma = \log(N)/(2N) \le 1/\operatorname{tr}(\mathbf{H})$. By Lemma B.1, we have the following excess risk bound for $\mathbf{w}_{\mathrm{sgd}}(N; \gamma^*)$,

$$L\big[\mathbf{w}_{\mathrm{sgd}}(N; \gamma)\big] - L(\mathbf{w}^*) \le \text{SGDBias} + \text{SGDVariance},$$

where

$$\text{SGDBias} \lesssim \sigma^2 \cdot \frac{(1-\gamma)^N}{(\gamma N)^2} \lesssim \sigma^2 \cdot \log^2 N \cdot \left(1 - \frac{\log N}{2N}\right)^N \lesssim \frac{\sigma^2 \log^2 N}{N^2} \lesssim \frac{\sigma^2}{N},$$

$$\text{SGDVariance} \lesssim \frac{\sigma^2}{N} \cdot \left(1 + (N\gamma)^2 \sum_{i>1} \lambda_i^2\right) \asymp \frac{\sigma^2}{N},$$

where we use the fact that $\sum_{i>1} \lambda_i^2 = \frac{1}{\log^2 N}$. This implies that SGD with sample size $N$ achieves at most $\mathcal{O}(\sigma^2/N)$ excess risk on this example.

Then we calculate the excess risk lower bound of ridge regression. By Lemma B.2 and let $\widetilde{\lambda} = \lambda + \sum_{i > k^*_{\mathrm{ridge}}} \lambda_i$, we have

$$L\big[\mathbf{w}_{\mathrm{ridge}}(N; \lambda)\big] - L(\mathbf{w}^*) = \text{RidgeBias} + \text{RidgeVariance}$$

$$\gtrsim \sigma^2 \cdot \left(\frac{\widetilde{\lambda}^2}{N^2_{\mathrm{ridge}}} + \frac{k^*_{\mathrm{ridge}}}{N_{\mathrm{ridge}}} + \frac{N_{\mathrm{ridge}} \sum_{i>k^*_{\mathrm{ridge}}} \lambda_i^2}{\widetilde{\lambda}^2}\right).$$

If $k^*_{\mathrm{ridge}} > N$, then

$$L\big[\mathbf{w}_{\mathrm{ridge}}(N; \lambda)\big] - L(\mathbf{w}^*) \gtrsim \frac{\sigma^2 k^*_{\mathrm{ridge}}}{N_{\mathrm{ridge}}} \ge \frac{\sigma^2 N}{N_{\mathrm{ridge}}} \ge \frac{\sigma^2}{N}, \quad \text{for } N_{\mathrm{ridge}} < \frac{N^2}{\log^2 N}.$$

If $k^*_{\text{ridge}} \leq N$, then $\sum_{i>k^*_{\text{ridge}}} \lambda_i^2 \geq \sum_{N<i\leq N^2} \frac{1}{N^2 \log^2 N} \eqsim \frac{1}{\log^2 N}$, which implies that

$$L\big[\mathbf{w}_{\text{ridge}}(N; \lambda)\big] - L(\mathbf{w}^*) \gtrsim \sigma^2 \cdot \left( \frac{\widetilde{\lambda}^2}{N_{\text{ridge}}^2} + \frac{N_{\text{ridge}}}{\widetilde{\lambda}^2} \cdot \frac{1}{\log^2 N} \right)$$

$$\geq \frac{\sigma^2}{N_{\text{ridge}}^{1/2} \log N}$$

$$\geq \frac{\sigma^2}{N}, \quad \text{for } N_{\text{ridge}} < \frac{N^2}{\log^2 N}.$$

To sum up, we have show that

$$L\big[\mathbf{w}_{\text{ridge}}(N_{\text{ridge}}; \lambda)\big] - L(\mathbf{w}^*) \gtrsim \frac{\sigma^2}{N} \gtrsim L\big[\mathbf{w}_{\text{sgd}}(N; \lambda)\big] - L(\mathbf{w}^*), \quad \text{for } N_{\text{ridge}} < \frac{N^2}{\log^2 N}.$$

This completes the proof. $\qquad\square$

## B.6 Proof of Theorem 5.3

*Proof.* The proof of Theorem 5.3 is similar to that of Theorem 5.1. In particular, we still consider two cases: (1) $\lambda \gtrsim \text{tr}(\mathbf{H})$ and (2) $\lambda \lesssim \text{tr}(\mathbf{H})$. For the first case, we can use the identical proof in Theorem 5.1 and get that SGD with sample size $N_{\text{sgd}} \eqsim (1 + R^2) \cdot N_{\text{ridge}}$ to achieve better excess risk than ridge regression. Note that we have assumed $R^2 = \Theta(1)$, therefore, we can claim that SGD outperforms ridge regression, as long as the sample size is at least in the same order of $N_{\text{ridge}}$.

For the second case that $\lambda \lesssim \text{tr}(\mathbf{H})$, for simplicity we denote $N := N_{\text{ridge}}$ and we can directly set $\gamma = 1/\text{tr}(\mathbf{H})$ and $N_{\text{sgd}} = N$. Let $k^* = \min\big\{ k : \lambda_k \leq \frac{\text{tr}(\mathbf{H})\log(N)}{N} \big\}$, then by the definition of $k^*_{\text{ridge}}$ in Lemma B.2 and the assumption that ridge regression is in the generalizable regime, we have $k^* \leq k^*_{\text{ridge}} \leq N_{\text{ridge}}$. Therefore, applying Lemma B.1 with $k_1 = k^*$, we have the following bound on the effective bias of SGD,

$$\text{SGDBias} \lesssim \sum_{i=1}^{k^*} \frac{(1 - \gamma\lambda_i)^N (\mathbf{w}^*[i])^2}{\lambda_i \gamma^2 N^2} + \sum_{i>k^*} \lambda_i (\mathbf{w}[i])^2$$

$$\lesssim \sum_{i=1}^{k^*} \frac{\left(1 - \frac{\log(N)}{N}\right)^N (\mathbf{w}^*[i])^2}{\lambda_i N^2} + \sum_{i>k^*} \lambda_i (\mathbf{w}[i])^2$$

$$\lesssim \frac{\|\mathbf{w}^*\|_{\mathbf{H}}^2}{N} + \sum_{i>k^*} \lambda_i (\mathbf{w}[i])^2.$$

Then by our assumption that

$$\sum_{i=k^*}^{N_{\text{ridge}}} \lambda_i \big(\mathbf{w}[i]\big)^2 \lesssim \frac{k^* \|\mathbf{w}^*\|_{\mathbf{H}}^2}{N},$$

we further have

$$\text{SGDBias} \lesssim \frac{\|\mathbf{w}^*\|_{\mathbf{H}}^2}{N} + \sum_{i>k^*} \lambda_i (\mathbf{w}[i])^2$$

$$\lesssim \sum_{i>k^*_{\text{ridge}}} \lambda_i (\mathbf{w}[i])^2 + \frac{(k^*_{\text{ridge}} + 1)\|\mathbf{w}^*\|_{\mathbf{H}}^2}{N},$$

where in the second inequality we use the fact that $k^* \leq k^*_{\text{ridge}} \leq N_{\text{ridge}}$. Regarding the variance of SGD, applying Lemma B.1 with $k_2 = k^*_{\text{ridge}}$ gives

$$\text{SGDVariance} \lesssim (\sigma^2 + \|\mathbf{w}^*\|_{\mathbf{H}}^2) \cdot \left( \frac{k^*_{\text{ridge}}}{N} + \frac{N}{(\text{tr}(\mathbf{H})^2)} \cdot \sum_{i \geq k^*_{\text{ridge}}} \lambda_i^2 \right)$$

$$\lesssim (\sigma^2 + \|\mathbf{w}^*\|_{\mathbf{H}}^2) \cdot \left( \frac{k^*_{\text{ridge}}}{N} + \frac{N}{(\lambda + \sum_{i>k^*_{\text{ridge}}} \lambda_i)^2} \cdot \sum_{i \geq k^*_{\text{ridge}}} \lambda_i^2 \right),$$

where the last inequality is due to the fact that $\lambda \lesssim \mathrm{tr}(\mathbf{H})$. Combining the above upper bounds for the bias and variance of SGD, we have that the output of SGD, with sample size $N_{\mathrm{sgd}} = N$ and learning rate $\gamma = 1/\mathrm{tr}(\mathbf{H})$, satisfies

$$\mathrm{SGDRisk} \lesssim \mathrm{SGDBias} + \mathrm{SGDVariance}$$

$$\lesssim \sum_{i > k^*_{\mathrm{ridge}}} \lambda_i (\mathbf{w}[i])^2 + \frac{(k^*_{\mathrm{ridge}} + 1)\|\mathbf{w}^*\|^2_{\mathbf{H}}}{N}$$

$$+ (\sigma^2 + \|\mathbf{w}^*\|^2_{\mathbf{H}}) \cdot \left( \frac{k^*_{\mathrm{ridge}}}{N_{\mathrm{ridge}}} + \frac{N_{\mathrm{ridge}}\gamma^2}{(\lambda + \sum_{i > N_{\mathrm{ridge}}} \lambda_i)^2} \cdot \sum_{i \geq k^*_{\mathrm{ridge}}} \lambda_i^2 \right)$$

$$\approx \sum_{i > k^*_{\mathrm{ridge}}} \lambda_i (\mathbf{w}[i])^2 + \frac{(k^*_{\mathrm{ridge}} + 1)\|\mathbf{w}^*\|^2_{\mathbf{H}}}{N}$$

$$+ \sigma^2 \cdot \left( \frac{k^*_{\mathrm{ridge}}}{N_{\mathrm{ridge}}} + \frac{N_{\mathrm{ridge}}\gamma^2}{(\lambda + \sum_{i > N_{\mathrm{ridge}}} \lambda_i)^2} \cdot \sum_{i \geq k^*_{\mathrm{ridge}}} \lambda_i^2 \right)$$

$$\lesssim \mathrm{RidgeBias} + \mathrm{RidgeVariance}, \tag{B.1}$$

where the last equality holds since we assume that $\|\mathbf{w}\|^2_{\mathbf{H}}/\sigma^2 = \Theta(1)$. Note that the R.H.S. of (B.1) is exactly the lower bound of the excess risk of ridge regression. Therefore, we can conclude that as long as $N_{\mathrm{sgd}} = N$, SGD with a tuned stepsize $\gamma$ will be no worse than ridge regression for all $\lambda$ (up to constant factors). This completes the proof.

$\square$

# C  Proof of Theorem B.2

In this section we always make Assumption 5.1. The results and techniques are either explicitly or implicitly presented in [7, 26]. For self-completeness, we provide a formal proof here.

**Notation.**  Following [26] and [7], we define the following notations:

- $\mathbf{v} := \mathbf{H}^{-\frac{1}{2}}\mathbf{x} \in \mathbb{R}^d$, then $\mathbf{v}$ is sub-Gaussian and has independent components.
- Let $\mathbf{X} := (\mathbf{x}_1, \ldots, \mathbf{x}_n)^\top \in \mathbb{R}^{n \times d}$. Let $\mathbf{X} = (\mathbf{X}_{0:k} \ \mathbf{X}_{k:\infty})$
- Let $\mathbf{X} = (\sqrt{\lambda_1}\mathbf{z}_1, \ldots, \sqrt{\lambda_d}\mathbf{z}_d) \in \mathbb{R}^{n \times d}$, then by Assumption 5.1, $\mathbf{z}_j$ is 1-sub-Gaussian and has independent components.
- Let $\widetilde{\mathbf{A}} := \mathbf{X}\mathbf{X}^\top = \sum_{i=1}^d \lambda_i \mathbf{z}_i \mathbf{z}_i^\top \in \mathbb{R}^{n \times n}$. Let $\mathbf{A} := \widetilde{\mathbf{A}} + \lambda_n \mathbf{I}_n = \mathbf{X}\mathbf{X}^\top + \lambda \mathbf{I}_n$.
- Let $\widetilde{\mathbf{A}}_k := \mathbf{X}_{k:\infty}\mathbf{X}_{k:\infty}^\top = \sum_{i \leq k} \lambda_i \mathbf{z}_i \mathbf{z}_i^\top \in \mathbb{R}^{n \times n}$. Let $\mathbf{A}_k := \widetilde{\mathbf{A}}_k + \lambda \mathbf{I}_n = \mathbf{X}_{k:\infty}\mathbf{X}_{k:\infty}^\top + \lambda \mathbf{I}_n$.
- Let $\widetilde{\mathbf{A}}_{-j} := \sum_{i \neq j} \lambda_i \mathbf{z}_i \mathbf{z}_i^\top \in \mathbb{R}^{n \times n}$. Let $\mathbf{A}_{-j} := \widetilde{\mathbf{A}}_{-j} + \lambda \mathbf{I}_n$.
- Let $\rho_k := \frac{\lambda + \sum_{i > k} \lambda_i}{\lambda_{k+1}}$.
- Let $\mathbf{C} := \mathbf{A}^{-1}\mathbf{X}\mathbf{H}\mathbf{X}^\top\mathbf{A}^{-1}$.
- Let $\mathbf{B} := (\mathbf{I}_d - \mathbf{X}^\top\mathbf{A}^{-1}\mathbf{X})\mathbf{H}(\mathbf{I}_d - \mathbf{X}^\top\mathbf{A}^{-1}\mathbf{X})$.
- We use $\mathbb{E}_{\mathbf{X}}[\cdot]$ and $\mathbb{E}_{\boldsymbol{\epsilon}}[\cdot]$ to denote the expectation with respect to the randomness of drawing $\mathbf{X}$ and the randomness of noise, respectively.

Under the above notations and from [7, 26], we have

$$\mathbb{E}_{\mathbf{X},\boldsymbol{\epsilon}}[\text{ridge error}] = \mathbb{E}_{\mathbf{X}}[\mathrm{RidgeBias}] + \mathbb{E}_{\mathbf{X},\boldsymbol{\epsilon}}[\mathrm{RidgeVariance}],$$

where

$$\mathrm{RidgeBias} := (\mathbf{w}^*)^\top \mathbf{B} \mathbf{w}^*, \qquad \mathrm{RidgeVariance} := \boldsymbol{\epsilon}^\top \mathbf{C} \boldsymbol{\epsilon}.$$

We next provide lower bounds for each terms respectively.

**Lemma C.1** (Variant of Lemma 10 in [7]). *There are constants $b, c \geq 1$ such that for every $k \geq 0$, with probability at least $0.1$,*

1. *for all $i \geq 1$,*

$$\mu_{k+1}(\mathbf{A}_{-i}) \leq \mu_{k+1}(\mathbf{A}) \leq \mu_1(\mathbf{A}_k) \leq c \left( \lambda + \sum_{j>k} \lambda_j + \lambda_{k+1} n \right),$$

2. *for all $1 \leq i \leq k$,*

$$\frac{1}{c} \left( \lambda + \sum_{j>k} \lambda_j \right) - c\lambda_{k+1} n \leq \mu_n(\mathbf{A}_k) \leq \mu_n(\mathbf{A}_{-i}) \leq \mu_n(\mathbf{A}),$$

3. *if $\rho_k \geq bn$, then*

$$\frac{1}{c}\lambda_{k+1}\rho_k \leq \mu_n(\mathbf{A}_k) \leq \mu_1(\mathbf{A}_k) \leq c\lambda_{k+1}\rho_k.$$

4. *if $\rho_k \geq bn$, then for all $i > k$,*

$$\mu_n(\mathbf{A}_{-i}) \geq \frac{1}{c}\lambda_{k+1}\rho_k$$

*Proof.* The first two claims are proved by noticing that $\mathbf{A} = \lambda \mathbf{I} + \widetilde{\mathbf{A}}$, $\mathbf{A}_k = \lambda \mathbf{I} + \widetilde{\mathbf{A}}_k$, $\mathbf{A}_{-i} = \lambda \mathbf{I} + \widetilde{\mathbf{A}}_{-i}$, and applying Lemma 10 in [7] to $\widetilde{\mathbf{A}}, \widetilde{\mathbf{A}}_k, \widetilde{\mathbf{A}}_{-j}$.

The third claim is proved by using the first two claims and that $\rho_k \geq bn$ to obtain that

$$\mu_1(\mathbf{A}_k) \leq c \left( \lambda + \sum_{i>k} \lambda_i + \lambda_{k+1} n \right) \leq \left( c + \frac{c}{b} \right) \cdot \left( \lambda + \sum_{i>k} \lambda_i \right),$$

$$\mu_n(\mathbf{A}_k) \geq \frac{1}{c} \left( \lambda + \sum_{i>k} \lambda_i \right) - c\lambda_{k+1} n \geq \left( \frac{1}{c} - \frac{c}{b} \right) \cdot \left( \lambda + \sum_{i>k} \lambda_i \right),$$

and by re-scaling the constants.

The fourth claim is used in Lemma 3 in [26], which can be proved under Assumption 5.1 as follows. Let $i > k$ and $\widetilde{\mathbf{A}}_{k,-i} = \sum_{j>k, j \neq i} \lambda_j \mathbf{z}_j \mathbf{z}_j^\top$. Then by Lemma 10 in [7] there is an absolute constant $c \geq 1$ such that

$$\mu_n(\widetilde{\mathbf{A}}_{-i}) \geq \mu_n(\widetilde{\mathbf{A}}_{k,-i}) \geq \frac{1}{c} \sum_{j>k, j \neq i} \lambda_j - c\lambda_{k+1} n$$

holds with probability at least $1 - 2e^{-n/c}$, which yields

$$\mu_n(\mathbf{A}_{-i}) \geq \lambda + \frac{1}{c} \sum_{j>k, j \neq i} \lambda_i - c\lambda_{k+1} n \geq \lambda + \frac{1}{2c} \sum_{j>k} \lambda_j - \left( c + \frac{1}{c} \right) \lambda_{k+1} n,$$

where the last inequality is because: (1) $\sum_{j>k, j \neq i} \lambda_j \geq \frac{1}{2} \sum_{j>k} \lambda_j$ if $i > k + 1$, and (2) $\sum_{j>k, j \neq i} \lambda_j = \sum_{j>k} \lambda_j - \lambda_{k+1}$ if $i = k + 1$. Finally, using the condition that $\rho_k \geq bn$ we obtain that for $i > k$,

$$\mu_n(\mathbf{A}_{-i}) \geq \lambda + \frac{1}{2c} \sum_{j>k} \lambda_j - (c + \frac{1}{c})\lambda_{k+1} n \geq \left( \frac{1}{2c} - \frac{c}{b} - \frac{1}{cb} \right) \cdot \left( \lambda + \sum_{j>k} \lambda_j \right),$$

which completes the proof by letting $b > 4c^2$ and $c \geq 1$

$\square$

**Variance Lower Bounds.** According to Lemma 7 in [7], and note that $\epsilon$ is independent of $\mathbf{X}$, has zero mean, and is $\sigma$-sub-Gaussian, we have that

$$\mathbb{E}_\epsilon[\text{RidgeVariance}] = \mathbb{E}_\epsilon[\epsilon^\top \mathbf{C} \epsilon] = \text{tr}\left(\mathbf{C} \cdot \mathbb{E}[\epsilon \epsilon^\top]\right) \geq \frac{1}{c}\sigma^2 \text{tr}(\mathbf{C}) \tag{C.1}$$

for some constant $c > 1$. In the following we lower bound $\text{tr}(\mathbf{C})$.

**Lemma C.2** (Variant of Lemma 8 in [7]).

$$\text{tr}(\mathbf{C}) = \sum_i \lambda_i^2 \mathbf{z}_i^\top \mathbf{A}^{-2} \mathbf{z}_i = \sum_i \frac{\lambda_i^2 \mathbf{z}_i^\top \mathbf{A}_{-i}^{-2} \mathbf{z}_i}{\left(1 + \lambda_i \mathbf{z}_i^\top \mathbf{A}_{-i}^{-1} \mathbf{z}_i\right)^2}.$$

*Proof.* This is from the proof of Lemma 14 in [26], and can be proved in the same way as Lemma 8 in [7]. $\qquad \square$

**Lemma C.3** (Variant of Lemma 14 in [7]). *There is a constant $c$ such that for any $i \geq 1$ with $\lambda_i > 0$, and any $0 \leq k \leq n/c$, with probability at least $0.1$,*

$$\frac{\lambda_i^2 \mathbf{z}_i^\top \mathbf{A}_{-i}^{-2} \mathbf{z}_i}{\left(1 + \lambda_i \mathbf{z}_i^\top \mathbf{A}_{-i}^{-1} \mathbf{z}_i\right)^2} \geq \frac{1}{cn} \cdot \left(1 + \frac{\lambda_{k+1}}{\lambda_i} \cdot \left(1 + \frac{\rho_k}{n}\right)\right)^{-2}$$

*Proof.* Let $\mathcal{L}_i$ be a random subspace if $\mathbb{R}^n$ of codimension $k$, then

$$\mathbf{z}_i^\top \mathbf{A}_{-i}^{-1} \mathbf{z}_i \geq \frac{1}{c_1} \cdot \frac{\|\Pi_{\mathcal{L}_i} \mathbf{z}_i\|_2^2}{\lambda + \sum_{j>k} \lambda_j + \lambda_{k+1} n} \qquad \text{(by Lemma C.1)}$$

$$\geq \frac{1}{c_2} \cdot \frac{n}{\lambda + \sum_{j>k} \lambda_j + \lambda_{k+1} n} \qquad \text{(by Corollary 13 in [7])}$$

$$= \frac{1}{c_2} \cdot \frac{n}{\lambda_{k+1}(\rho_k + n)},$$

where $c_1, c_2 > 1$ are constants. The above implies that

$$\frac{\lambda_i^2 \mathbf{z}_i^\top \mathbf{A}_{-i}^{-2} \mathbf{z}_i}{\left(1 + \lambda_i \mathbf{z}_i^\top \mathbf{A}_{-i}^{-1} \mathbf{z}_i\right)^2} = \left(1 + \left(\lambda_i \mathbf{z}_i^\top \mathbf{A}_{-i}^{-1} \mathbf{z}_i\right)^{-1}\right)^{-2} \cdot \frac{\left\|\mathbf{z}_i^\top \mathbf{A}_{-i}^{-1}\right\|_2^2}{\left(\mathbf{z}_i^\top \mathbf{A}_{-i}^{-1} \mathbf{z}_i\right)^2}$$

$$\geq \left(1 + \left(\lambda_i \mathbf{z}_i^\top \mathbf{A}_{-i}^{-1} \mathbf{z}_i\right)^{-1}\right)^{-2} \cdot \frac{1}{\|\mathbf{z}_i\|_2^2} \qquad \text{(by Cauchy-Schwarz's inequality)}$$

$$\geq \left(1 + c_2 \cdot \frac{\lambda_{k+1}(\rho_k + n)}{n\lambda_i}\right)^{-2} \cdot \frac{1}{\|\mathbf{z}_i\|_2^2}. \quad \text{(by the lower bound for } \mathbf{z}_i^\top \mathbf{A}_{-i}^{-1} \mathbf{z}_i)$$

According to Corollary 13 in [7], there is constant $c_3 > 1$ such that $\|\mathbf{z}_i\|_2^2 \leq \frac{1}{c_3} n$ holds with constant probability, inserting which into the above inequality and rescaling the constants complete the proof. $\qquad \square$

**Lemma C.4** (Variant of Lemma 16 in [7]). *There is constant $c$ such that for any $0 \leq k \leq n/c$ and any $b > 1$ with probability at least $0.1$,*

- *if $\rho_k < bn$, then $\text{tr}(\mathbf{C}) \geq \frac{k+1}{cb^2 n}$;*

- *if $\rho_k \geq bn$, then $\text{tr}(\mathbf{C}) \geq \frac{1}{cb^2} \min_{\ell \leq k} \left\{ \frac{\ell}{n} + \frac{b^2 n \sum_{i>\ell} \lambda_i^2}{(\lambda_{k+1} \rho_k)^2} \right\}.$*

*Proof.* This is proved by repeating the proof of Lemma 16 in [7], where we replace Lemmas 8 and 14 in [7] with our Lemmas C.2 and C.3 respectively. $\qquad \square$

**Theorem C.5** (Restatement of Theorem B.2, variance part). *There exist absolute constants $b, c, c_1 > 1$ for the following to hold: let*

$$k^* := \min\{k : \lambda + \sum_{i>k} \lambda_i \geq bn\lambda_{k+1}\},$$

*then with probability at least $0.1$:*

- *if $k^* \geq n/c_1$ then*

$$\mathbb{E}_{\boldsymbol{\epsilon}}[\text{RidgeVariance}] \geq \frac{\sigma^2}{c};$$

- *if $k^* < n/c_1$ then*

$$\mathbb{E}_{\boldsymbol{\epsilon}}[\text{RidgeVariance}] \geq \frac{\sigma^2}{c}\left(\frac{k^*}{n} + \frac{n}{\lambda + \sum_{i>k^*}\lambda_i} \cdot \sum_{i>k^*}\lambda_i^2\right).$$

*As a direct consequence, the expected ridge variance is lower bounded by*

$$\mathbb{E}_{\mathbf{X},\boldsymbol{\epsilon}}[\text{RidgeVariance}] \geq \begin{cases} \frac{\sigma^2}{10c}, & k^* \geq n/c_1 \\ \frac{\sigma^2}{10c}\left(\frac{k^*}{n} + \frac{n}{\lambda+\sum_{i>k^*}\lambda_i} \cdot \sum_{i>k^*}\lambda_i^2\right), & k^* < n/c_1. \end{cases}$$

*Proof.* The high probability lower bound is proved by (C.1), our Lemma C.4, and Lemma 17 in [7]. The expectation lower bound follows immediately from the high probability lower bound by noticing the ridge variance error is non-negative. $\qquad\square$

**Bias Lower Bound.** Recall the ridge bias error is [26]

$$\text{RidgeBias} = (\mathbf{w}^*)^\top \mathbf{B}\mathbf{w}^* = \sum_i (\mathbf{B})_{ii}(\mathbf{w}_i^*)^2 + 2\sum_{i>j}(\mathbf{B})_{ij}\mathbf{w}_i^*\mathbf{w}_j^*. \tag{C.2}$$

The following lemma shows the crossing terms are zero in expectation.

**Lemma C.6.** *For $i \neq j$,*

$$\mathbb{E}_{\mathbf{X}}[(\mathbf{B})_{ij}] = 0.$$

*Proof.* Recall that

$$\mathbf{B} := \left(\mathbf{I}_d - \mathbf{X}^\top \mathbf{A}^{-1}\mathbf{X}\right)\mathbf{H}\left(\mathbf{I}_d - \mathbf{X}^\top \mathbf{A}^{-1}\mathbf{X}\right).$$

Recall that $\mathbf{X} = \left(\sqrt{\lambda_1}\mathbf{z}_1, \ldots \sqrt{\lambda_d}\mathbf{z}_d\right)$, thus the $i$-th column of $\left(\mathbf{I}_d - \mathbf{X}^\top \mathbf{A}^{-1}\mathbf{X}\right)$ is

$$\left(\mathbf{I}_d - \mathbf{X}^\top \mathbf{A}^{-1}\mathbf{X}\right)_i = \mathbf{e}_i - \sqrt{\lambda_i}\mathbf{X}^\top \mathbf{A}^{-1}\mathbf{z}_i.$$

Moreover recall $\mathbf{H} = \text{diag}(\lambda_1, \ldots, \lambda_d)$, therefore

$$\begin{aligned}
(\mathbf{B})_{ij} = \mathbf{e}_i^\top \mathbf{B}\mathbf{e}_j &= \left(\mathbf{e}_i - \sqrt{\lambda_i}\mathbf{X}^\top \mathbf{A}^{-1}\mathbf{z}_i\right)^\top \mathbf{H}\left(\mathbf{e}_j - \sqrt{\lambda_j}\mathbf{X}^\top \mathbf{A}^{-1}\mathbf{z}_j\right) \\
&= \mathbf{e}_i^\top \mathbf{H}\mathbf{e}_j - \sqrt{\lambda_i}\mathbf{e}_j^\top \mathbf{H}\mathbf{X}^\top \mathbf{A}^{-1}\mathbf{z}_i - \sqrt{\lambda_j}\mathbf{e}_i^\top \mathbf{H}\mathbf{X}^\top \mathbf{A}^{-1}\mathbf{z}_j + \sqrt{\lambda_i\lambda_j}\mathbf{z}_i^\top \mathbf{A}^{-1}\mathbf{X}\mathbf{H}\mathbf{X}^\top \mathbf{A}^{-1}\mathbf{z}_j \\
&= \mathbf{e}_i^\top \mathbf{H}\mathbf{e}_j - \left(\sqrt{\lambda_i\lambda_j}\lambda_j + \sqrt{\lambda_i\lambda_j}\lambda_i\right)\mathbf{z}_i^\top \mathbf{A}^{-1}\mathbf{z}_j + \sqrt{\lambda_i\lambda_j}\mathbf{z}_i^\top \mathbf{A}^{-1}\mathbf{X}\mathbf{H}\mathbf{X}^\top \mathbf{A}^{-1}\mathbf{z}_j.
\end{aligned}$$

The first term is zero since $\mathbf{H}$ is diagonal and $i \neq j$. We next show the second term is zero in expectation. Indeed, let

$$F(\mathbf{z}_i) := \mathbf{z}_i^\top \mathbf{A}^{-1}\mathbf{z}_j = \mathbf{z}_i^\top \left(\mathbf{A}_{-i} + \lambda_i\mathbf{z}_i\mathbf{z}_i^\top\right)^{-1}\mathbf{z}_j,$$

where $\mathbf{A}_{-i}$ is independent of $\mathbf{z}_i$, then $F(\mathbf{z}_i) = -F(-\mathbf{z}_i)$. Also note that $\mathbf{z}_i$ follows a standard Gaussian which is symmetric, therefore $\mathbb{E}_{\mathbf{z}_i}F(\mathbf{z}_i) = 0$. In a similar manner, the third term is also zero in expectation. The proof is then completed.

$\qquad\square$

**Lemma C.7** (Part of the proof of Lemma 15 in [26]). *There exists absolute constant $c > 1$, such that with probability at least $0.1$,*

$$(\mathbf{B})_{ii} \geq \frac{1}{c} \cdot \frac{\lambda_i}{\left(1 + \frac{\lambda_i}{\lambda_{k+1}} \cdot \frac{n}{\rho_k}\right)^2}.$$

*As a direct consequence,*

$$\mathbb{E}_{\mathbf{X}}[(\mathbf{B})_{ii}] \geq \frac{1}{10c} \cdot \frac{\lambda_i}{\left(1 + \frac{\lambda_i}{\lambda_{k+1}} \cdot \frac{n}{\rho_k}\right)^2}.$$

*Proof.* This lemma summarizes part of the proof of Lemma 15 in [26]. Recall that $\mathbf{H}$ is diagonal and $\mathbf{B} := \left(\mathbf{I}_d - \mathbf{X}^\top \mathbf{A}^{-1}\mathbf{X}\right)\mathbf{H}\left(\mathbf{I}_d - \mathbf{X}^\top \mathbf{A}^{-1}\mathbf{X}\right)$, thus

$$
\begin{aligned}
(\mathbf{B})_{ii} &= \lambda_i \left\| \left(\mathbf{I}_d - \mathbf{X}^\top \mathbf{A}^{-1}\mathbf{X}\right)_i \right\|_2^2 \qquad \text{(since } \mathbf{H} \text{ is diagonal)} \\
&= \lambda_i \left\| e_i^\top - \sqrt{\lambda_i}\mathbf{z}_i^\top \mathbf{A}^{-1}\mathbf{X} \right\|_2^2 \qquad \left(\mathbf{X} = \left(\sqrt{\lambda_1}\mathbf{z}_1, \ldots, \sqrt{\lambda_j}\mathbf{z}_j, \ldots \sqrt{\lambda_d}\mathbf{z}_d\right)\right) \\
&= \lambda_i \left\| e_i^\top - \left(\sqrt{\lambda_i \lambda_1}\mathbf{z}_i^\top \mathbf{A}^{-1}\mathbf{z}_1, \ldots, \sqrt{\lambda_i \lambda_j}\mathbf{z}_i^\top \mathbf{A}^{-1}\mathbf{z}_j, \ldots \sqrt{\lambda_i \lambda_d}\mathbf{z}_i^\top \mathbf{A}^{-1}\mathbf{z}_d\right) \right\|_2^2 \\
&\geq \lambda_i \left(1 - \lambda_i \mathbf{z}_i^\top \mathbf{A}^{-1}\mathbf{z}_i\right)^2 \qquad \text{(use Pythagorean theorem)} \\
&= \frac{\lambda_i}{\left(1 + \lambda_i \mathbf{z}_i^\top \mathbf{A}_{-i}^{-1}\mathbf{z}_i\right)^2},
\end{aligned}
$$

where in the last step we use $\mathbf{A} = \mathbf{A}_{-i} + \lambda_i \mathbf{z}_i \mathbf{z}_i^\top$ and that

$$
\begin{aligned}
1 - \lambda_i \mathbf{z}_i^\top \mathbf{A}^{-1}\mathbf{z}_i &= 1 - \lambda_i \mathbf{z}_i^\top \left(\mathbf{A}_{-i} + \lambda_i \mathbf{z}_i \mathbf{z}_i^\top\right)^{-1} \mathbf{z}_i \\
&= 1 - \lambda_i \mathbf{z}_i^\top \left(\mathbf{A}_{-i}^{-1} - \lambda_i \mathbf{A}_{-i}^{-1}\mathbf{z}_i(1 + \mathbf{z}_i^\top \mathbf{A}_{-i}^{-1}\mathbf{z}_i)^{-1}\mathbf{z}_i^\top \mathbf{A}_{-i}^{-1}\right) \mathbf{z}_i \\
&= \frac{1}{1 + \lambda_i \mathbf{z}_i^\top \mathbf{A}_{-i}^{-1}\mathbf{z}_i}.
\end{aligned}
$$

Now according to Corollary 13 in [7], there exists constant $c_1 > 1$ such that

$$
\|\mathbf{z}_i\|_2^2 \leq c_1 n
$$

holds with constant probability; and according to Lemma C.1, there exists constant $c_2 > 1$ such that for any $i \geq 1$,

$$
\mu_n(\mathbf{A}_{-i}) \geq \frac{1}{c_2}\lambda_{k+1}\rho_k
$$

holds with constant probability. These two facts imply that

$$
\mathbf{z}_i^\top \mathbf{A}_{-i}^{-1}\mathbf{z}_i \leq \mu_n(\mathbf{A}_{-i})^{-1} \|\mathbf{z}_i\|_2^2 \leq c_1 c_2 \frac{n}{\lambda_{k+1}\rho_k},
$$

inserting which into the bound of $(\mathbf{B})_{ii}$, we conclude that with constant probability,

$$
(\mathbf{B})_{ii} \geq \frac{\lambda_i}{\left(1 + \lambda_i \mathbf{z}_i^\top \mathbf{A}_{-i}^{-1}\mathbf{z}_i\right)^2} \geq \frac{\lambda_i}{\left(1 + c_1 c_2 \cdot \frac{\lambda_i}{\lambda_{k+1}} \cdot \frac{n}{\rho_k}\right)^2}.
$$

Finally a rescaling of the constants completes the proof. $\qquad\square$

**Theorem C.8** (Restatement of Theorem B.2, bias part). *There exist absolute constants $b, c > 1$ for the following to hold: let*

$$
k^* := \min\{k : \lambda + \sum_{i>k} \lambda_i \geq bn\lambda_{k+1}\},
$$

*then*

$$
\mathbb{E}_{\mathbf{X}}[\text{RidgeBias}] \geq \frac{1}{c}\left(\frac{\lambda + \sum_{i>k^*}\lambda_i}{n^2} \cdot \|\mathbf{w}^*\|_{\mathbf{H}_{0:k^*}^{-1}}^2 + \|\mathbf{w}^*\|_{\mathbf{H}_{k^*:\infty}}^2\right).
$$

*Proof.* By (C.2), Lemmas C.6 and C.7, we have that,

$$
\begin{aligned}
\mathbb{E}_{\mathbf{X}}[\text{RidgeBias}] &= \sum_i (\mathbf{B})_{ii} (\mathbf{w}_i^*)^2 \\
&\geq \frac{1}{c_1} \sum_i \frac{1}{\left(1 + \frac{\lambda_i}{\lambda_{k^*+1}} \cdot \frac{n}{\rho_{k^*}}\right)^2} \cdot \lambda_i (\mathbf{w}_i^*)^2 \qquad \text{(choose } k = k^*) \\
&\geq \frac{1}{c_1 b^2} \sum_i \frac{1}{\left(\frac{1}{b} + \frac{\lambda_i}{\lambda_{k^*+1}} \cdot \frac{n}{\rho_{k^*}}\right)^2} \cdot \lambda_i (\mathbf{w}_i^*)^2,
\end{aligned}
$$

where $c_1, b > 1$ are all absolute constants. Note that for all $i \leq k^*$, we must have $\lambda + \sum_{j>i-1} \lambda_j < bn\lambda_i$,

$$\frac{\lambda_i}{\lambda_{k^*+1}} \cdot \frac{n}{\rho_{k^*}} = \frac{\lambda_i n}{\lambda + \sum_{j>k^*} \lambda_j} \geq \frac{\lambda_i n}{\lambda + \sum_{j>i-1} \lambda_j} \geq \frac{1}{b},$$

and for all $i \geq k^* + 1$, we have

$$\frac{\lambda_i}{\lambda_{k^*+1}} \cdot \frac{n}{\rho_{k^*}} \leq \frac{n}{\rho_{k^*}} \leq \frac{1}{b},$$

then

$$\mathbb{E}_{\mathbf{X}}[\mathrm{RidgeBias}] \geq \frac{1}{c_1 b^2} \sum_i \frac{1}{\left(\frac{1}{b} + \frac{\lambda_i}{\lambda_{k^*+1}} \cdot \frac{n}{\rho_{k^*}}\right)^2} \cdot \lambda_i (\mathbf{w}_i^*)^2$$

$$\geq \frac{1}{2c_1 b^2} \cdot \left( \sum_{i \leq k^*} \frac{1}{\left(\frac{\lambda_i}{\lambda_{k+1}} \cdot \frac{n}{\rho_k}\right)^2} \cdot \lambda_i (\mathbf{w}_i^*)^2 + \sum_{i > k^*} \frac{1}{(1/b)^2} \cdot \lambda_i (\mathbf{w}_i^*)^2 \right)$$

$$\geq \frac{1}{c} \left( \sum_{i \leq k^*} \frac{(\lambda_{k+1}\rho_k)^2}{n^2} \cdot \lambda_i^{-1} (\mathbf{w}_i^*)^2 + \sum_{i > k^*} \lambda_i (\mathbf{w}_i^*)^2 \right)$$

$$= \frac{1}{c} \left( \frac{\lambda + \sum_{i>k^*} \lambda_i}{n^2} \cdot \|\mathbf{w}^*\|_{\mathbf{H}_{0:k^*}^{-1}}^2 + \|\mathbf{w}^*\|_{\mathbf{H}_{k^*:\infty}}^2 \right),$$

where $c > 1$ is an absolute constant. $\square$