# OpenReview forum: "The Benefits of Implicit Regularization from SGD in Least Squares Problems"
_NeurIPS.cc/2021/Conference — NeurIPS 2021 Poster_

### Official Review · Reviewer_P3AS · 2021-07-11

**Rating:** 5
**Confidence:** 4

**Summary:**

This paper shows that for a class of least-squares problems, unregularized SGD generalizes no worse than ridge estimate given additional logarithmically more samples. On the other hand, for certain data sets, ridge estimate (positive regularization) may require quadratically more samples than SGD to achieve the same generalization performance.

**Limitations And Societal Impact:**

It is a theoretical paper. I think the main limitation is that for some regression problems, the sample size is limited. It is easier to select the number of features and the authors have not covered this part.

**Main Review:**

Overall it is an interesting paper. The main contribution is to show the advantage of unregularized SGD. On the other hand, I have some concerns about how important the contribution is and whether the comparisons are fair.
1) While the logarithm factor does not sound large, it often changes the problem from an over-parameterized regime to an under-parameterized regime.  For example in corollary 1, while the ridge problem is over-parameterized with d=O(n), the SGD problem is very under-parameterized with $d/n\rightarrow 0$. It feels a bit weird to compare two methods within two different regimes.
2) The error bounds are compared with the constant factor being omitted. So I would not claim SGD outperforms ridge based on Theorem 5.2 or Theorem 5.3 because the constant factor could be less than 1 until I read the proof. I think the authors should make it more clear why the risk of ridge estimate is worse than SGD. In addition, the term $||w^*||_H$ with subscript $k:\infty$ seem to be missed in both risk calculation in the proof of Theorem 5.2. If this term is constant (which is very likely), then this term is the dominant term of the risk.
3) Overall, it seems like comparing ridge with SGD is to compare risk at $d/n\rightarrow \gamma$ at different $\gamma$. I am not entirely convinced that SGD generalizes no worse or better than ridge is because of SGD. The results are just giving another example that we need to tune the ratio $d/n$. Specifically, Theorem 5.1 looks like comparing risk with $\gamma=0$ with $\gamma>1$, and we know given the constant signal-to-noise ratio, the two risks are comparable within a constant factor.  For Theorem 5.2 and 5.3, we know when the sample covariance matrix is very spiky, the best risk is achieved at $\gamma=0$. Adding more samples can make $d/n\rightarrow 0$, or I would suggest doing a PCR and achieve $d/n\rightarrow 0$ by decreasing d.
4) The problem is specific to linear regression which limits the contribution. Further, as shown in [26], in some cases, the optimal ridge parameter is negative. To make the results more convincing, the authors should explore this direction as well.

Overall, I have concerns about the contribution of the paper.


**Time Spent Reviewing:**

6

---

> ### Author Response · Authors · 2021-08-11
> **Response to Reviewer P3AS**
>
> Thank you for your comments. Please see the below for responses regarding the specific concerns:
>
> ---
> **Q:** "While the logarithm factor does not sound large, it often changes the problem from an over-parameterized regime to an under-parameterized regime. .... It feels a bit weird to compare two methods within two different regimes."
>
> **A:**  We would like to first point out that underparameterized setting and overparameterized setting in our paper are not significantly different since our paper is different from the proportional limit setting that assumes $d, n \rightarrow\infty$, and $d/n\rightarrow\gamma$. Our paper focuses on ``instance-wise’’ risk comparison, which only concerns the eigenspectrum of H but not the ratio $d/n$. In particular, given a certain problem instance (a certain eigenspectrum), there will not be a huge difference between the underparameterized and overparameterized regimes since the generalization error is actually a continuous function with respect to $n$. Moreover, for Corollary 5.1, we can slightly modify the example data distribution such that the problem is overparameterized or underparameterized for both SGD and ridge regression. The reason behind this is that as long as the spectrum of $H$ does not change a lot, changing the dimension $d$ will barely affect the generalization error of SGD and ridge regression.
>
> ---
>
> **Q:** "Constant factors being omitted …if the term $\\|w^*\\|\_{H\_{k:\infty}}$ is constant (which is very likely)"
>
> **A:** We would like to point out that for SGD, in general, the term $\\|w^\*\\|\_{H\_{k\_1:\infty}}^2$ is not a constant, since otherwise the solution of SGD is not in the generalizable regime (see Definition 1). To give you an example, consider the case when $w^\*$ has constant $\ell\_2$ norm (which is a common assumption in the literature), it is easy to show that $\\|w^\*\\|\_{H\_{k:\infty}}^2<\\|w^\*\\|\_2 / (n\gamma)$, which decreases at a rate of $O(1/n)$ when using constant stepsize $\gamma$. In general, $\\|w^\*\\|\_{H\_{k:\infty}}^2$ could decrease at a rate of $O(1/poly(n))$, which permits us to cancel out absolute constant by using constantly more samples, i.e., the sample inflation of SGD will only increase by a multiplicative constant.
>
> ---
>
> **Q:**  "the term $\\|w^\*\\|\_{H\_{k:\infty}}$ seem to be missed."
>
> **A:** In the proof of Theorem 5.2, we do not include $\\|w^\*\\|\_{H\_{k:\infty}}^2$ since the constructed problem instance satisfies $w^\*[i]=0$ for all $i>1$, which implies that $\\|w^\*\\|\_{H\_{k:\infty}}^2=0$ for all $k>1$.
>
> ---
>
>
> **Q:** "It seems like comparing ridge with SGD is to compare risk at $d/n\rightarrow \gamma$ at different $\gamma$..."
>
> **A:**  There is a huge misunderstanding regarding the comparison and generalization error bounds in our paper. In fact, our setting is fundamentally different from the proportional limit setting that assumes $d, n \rightarrow\infty$ and $d/n\rightarrow\gamma$. Note that our results are non-asymptotic rather than asymptotic. Furthermore, our results do not require $d/n$ to be a constant. In fact, our results even hold for infinite $d$ (provided the tail eigenvalues of the covariance matrix decay sufficiently fast), and in such a case,  $d/n = \infty$ for both SGD and ridge. So your comment that “comparing ridge with SGD is to compare risk at $d/n\rightarrow \gamma$ at different $\gamma$” is incorrect.
>
> ---
>
> **Q:** "Further, as shown in [26], in some cases, the optimal ridge parameter is negative. To make the results more convincing, the authors should explore this direction as well."
>
> **A:** Good question and thanks for raising this point! By examining the whole proof for Gaussian data distributions, you can see that we never need $\lambda$ to be non-negative. Therefore, our current results for Gaussian least squares problems can indeed cover the negative $\lambda$ case. We will emphasize this point in the revision.
>
> ---
>
> **Q:** "The problem is specific to linear regression which limits the contribution. "
>
> **A:** Extending the results to more complicated models is an important future work direction. However, even in the simple linear regression setting, there is no such kind of result before our work. Without a full comparison between SGD and ridge regression for the simplest possible problem — the linear regression — it seems unlikely that one can achieve this in more complicated models. As our work is the first work on the sample inflation comparison between SGD and ridge regression, we believe we have taken an important step along this direction and the contribution of our work is very significant.
>
> ---
>
> **Q:** "for some regression problems, sample size is limited. It is easier to select the number of features and the authors have not covered this part"
>
> **A:** We would like to emphasize that the goal of this paper is not to compare the risk of ridge regression and SGD when using the same sample size but, as mentioned in lines 157-163, is focusing on investigating how many more samples SGD requires in order to perform no worse than ridge regression. The setting you suggested is interesting and we are happy to explore it in the future.
>
> ---
>
> We hope our responses will address the possible misunderstandings and the concerns about our contributions.

---

> > ### Comment · Reviewer_P3AS · 2021-08-25
> > **Thanks for the response**
> >
> > I think dimension d in my comments replaced by effective rank will be more appropriate. I agree that the actual dimension can go to infinity as long as the eigenvalue of H decays fast enough. But the effective rank/sample size will be properly bounded. Furthermore, Corollary 5.1 is under the condition when d=O(n). In terms of $\gamma:=$effective rank/ sample size, the Ridge is still in the range of $\gamma>c>0$ for $\alpha<0.5$ while SGD has go to the $\gamma=0$ case. When $1\geq \alpha\geq0.5$, it is true that both Ridge and SGD fall in the case $\gamma=0$. Maybe it is fair to compare Ridge with SGD in this case. On the other hand, when $\alpha>1$, while both algorithms are still in the case of $\gamma=0$, SGD needs a bigger sample size more than a log factor (need to be in polynomial). It is very unclear to me what is the interesting message.
> >
> > I agree and am glad that the proof probably can be extended to $\lambda>-\lambda_{min}$ which is nice.

---

> > > ### Author Response · Authors · 2021-08-26
> > > **Response to your new comments**
> > >
> > > Thank you for your further comments.
> > >
> > > Even if replacing the dimension $d$ by the effective dimension (or effective rank), your previous comment is still incorrect. In particular, your new comment that “the ridge regression is in the range of $\gamma > c > 0$ for $\alpha<0.5$” is not true. In fact, our work concerns an instance-wise comparison (rather than comparing the worst-case guarantees of SGD and ridge regression), and we can show that for some problem instances, ridge regression can achieve $\gamma = 0$ for $0<\alpha < 0.5$. To give you an example, let’s consider a problem instance with $\sigma^2=1$, $w^*[1] =1$ and $w^*[k]=0$ for all $k>1$. Then by (6.2) in our paper, it is clear that when $\alpha<0.5$, we have $k^* = (\tilde\lambda/N)^{-1/\alpha}$ and
> > >
> > > $$
> > > RidgeRisk \gtrsim \bigg(\frac{\tilde \lambda}{N}\bigg)^2 + \frac{k^*}{N} + \frac{N}{\tilde\lambda^2}\sum_{i>k^*}\lambda_i^2
> > > = \bigg(\frac{\tilde \lambda}{N}\bigg)^2 + \bigg(\frac{N}{\tilde\lambda}\bigg)^{1/\alpha}/N + \frac{N^2}{\tilde\lambda^2}\cdot N^{-2\alpha}.
> > > $$
> > >
> > > It can be shown that in order to make the lower bound diminish as  $N\rightarrow \infty$, we have to choose regularization parameter $\lambda \eqsim \tilde \lambda =N^{1-\beta}$ for some $\beta \in (0, \alpha)$. Then it is easy to show that ratio between effective dimension/rank and the sample size is $k^*/N = N^{\beta/\alpha-1}$, which immediately suggests that $\gamma=0$ when $N\rightarrow \infty$ because $\beta<\alpha$.
> > >
> > > We hope this will clear up your confusion.

---

> > > > ### Comment · Reviewer_P3AS · 2021-08-26
> > > > **Thanks for the response**
> > > >
> > > > I think I refer to this quantity $\frac{(\sum_{i=1}^d\lambda_i)^2}{\sum_{i=1}^d\lambda_i^2}$ which is in order of d when $\lambda_i=\frac{1}{i^\alpha}$ with $\alpha<0.5$. I think there is a ``true dimension'' such that the true dimension over sample size reflects the level of complexity of the linear system. When eigenvalues do not decay and $w^*$ are random, it is simply true dimension = d. When eigenvalue decays, I think the above quantity is a good measurement of true dimension when $w^*$ is random. I agree that maybe when $w^*$ is just one eigenvector of H and the corresponding eigenvalue is bounded, the true dimension could be constant. But why it is interesting to study this kind of special case?
> > > >
> > > > The paper claims that due to the special property of SGD (i.e., the implicit regularization from SGD), it is no worse than Ridge with a small price to pay (log factor). All my arguments are trying to understand when this claim is correct in the sense of 1) the price is indeed small and 2) the benefit comes from SGD not somewhere else. To only pay a small price of a log factor, the eigenvalues can not decay too fast. Otherwise, for $\alpha>1$, we need to pay polynomial factor instead. On the other hand, if the eigenvalues decay too slow and $w^*$ is random (basically, if $w^*$ is a common one), then I think the benefits come from the regime shift (SGD is in $\gamma=0$ while Ridge is in $\gamma>c>0$). I agree by adding conditions like eigenvalues decay with median speed ($0.5\leq \alpha\leq 1$) or a special $w^*$, Ridge and SGD (after paying the log factor price) may face the same level of complexity of the linear system. But with these conditions, the results look restricted.

---

> > > > > ### Author Response · Authors · 2021-08-27
> > > > > **Thanks for your further comments**
> > > > >
> > > > > **Q:** “I refer to this quantity $\frac{(\sum_{i=1}^d \lambda_i )^2}{\sum_{i=1}^d \lambda_i^2}$....”
> > > > >
> > > > > **A:** The quantity $\frac{(\sum_{i=1}^d \lambda_i )^2}{\sum_{i=1}^d \lambda_i^2}$ you referred to is not the right quantity to measure the “dimension” in our setting, as it does not naturally appear in either the upper or lower bounds in our results for SGD and ridge regression. In order to characterize the number of “dimensions” that can be learned by certain algorithms, its definition should depend on the algorithm hyperparameters (e.g., regularization parameter for ridge regression, learning rate for SGD). In fact, the effective dimension $k^*$ used in our paper is exactly such kind of quantity to measure the “dimension” in our setting, as it is algorithmic-dependent, and naturally appears in both the upper and lower risk bounds for SGD and ridge regression (See Theorems 5.1&5.2 in [29], Theorem C.2 in Appendix page 22 for the detailed definitions). Thus, when we talk about regimes, we should use $k^*$ rather than the quantity referred by you to define the regime. With our $k^*$, we can show that both SGD and ridge regression are in the same regime for our results (i.e., $\gamma= k^*/N$, $\gamma\rightarrow 0$ when $N\rightarrow 0$).
> > > > >
> > > > > ---
> > > > >
> > > > > **Q:** “I agree that maybe when $w^*$ is just one eigenvector of H and the corresponding eigenvalue is bounded, the true dimension could be constant. But why it is interesting to study this kind of special case?”
> > > > >
> > > > > **A:** We must emphasize that the example we described in our last response ($w^\* =(1,0,\\dots)$) is just a *simple* example for refuting your previous comment that ridge regression and SGD are in different regimes. However, our results are *not* limited to this simple example! In fact, it can be easily shown that when $w^*$ has bounded $\ell_2$ norm (which is widely assumed in many papers), the lower bound of $\mathrm{RidgeRisk}$ (see (6.2) in our paper) satisfies
> > > > >
> > > > > $$
> > > > > (\tilde \lambda/N)^2 \cdot ||w^*||\_{H\_{0:k^*}^{-1}}^2 + ||w^*||\_{H\_{k^*:\infty}}^2  +\frac{k^*}{N} + \frac{N}{\tilde\lambda^2}\sum_{i>k^*}\lambda_i^2
> > > > > \lesssim  \frac{\tilde \lambda}{N} + \bigg(\frac{N}{\tilde\lambda}\bigg)^{1/\alpha}/N + \frac{N^2}{\tilde\lambda^2}\cdot N^{-2\alpha},
> > > > > $$
> > > > > where we use the definition that $\lambda_{k^*}\eqsim \tilde{\lambda}/N$.
> > > > >
> > > > > Therefore, we can still set $\lambda \eqsim \tilde \lambda =N^{1-\beta}$ for some $\beta \in (0, \alpha)$ to guarantee that such a lower bound will be diminishing as $N$ goes to infinity, and it is still in the $\gamma =  \mathrm{EffectiveDim}/N\rightarrow 0$ regime.
> > > > >
> > > > > We hope the above explanation can clear things up.

---

> > > > > > ### Comment · Reviewer_P3AS · 2021-08-27
> > > > > > **Thanks for the response**
> > > > > >
> > > > > > "you referred to is not the right quantity to measure the “dimension” in our setting, as it does not naturally appear in either the upper or lower bounds in our results for SGD and ridge regression. In order to characterize the number of “dimensions” that can be learned by certain algorithms, its definition should depend on the algorithm hyperparameters"
> > > > > >
> > > > > > I respectfully do not agree. The reason I look at this dimension is to evaluate the level of complexity of the problem (as I mentioned previously) and it is not fair to compare two algorithms when they face problems with different levels of complexity. Otherwise, I can claim OLS is the best algorithm based on the comparison that OLS achieves the minimal risk given infinite sample size while all other algorithms can not when they are given zero sample size. Of course, this is ridiculous because I looked at different levels of complexity of the problems for different algorithms (extremely easy for OLS and impossible for other algorithms).
> > > > > >
> > > > > > Since it is the complexity of the problem (linear system in this case),  the dimension should depend on the linear problem itself NOT the algorithm nor the bounds in the paper. I can design useless algorithms and derive vascious bounds which should not change how we think the problem itself is difficult or not. There is no doubt that SGD is a very useful algorithm and bounds in the paper are clearly carefully derived, but when comparing algorithms, we should not define the dimension that justifies what SGD or the bounds ``thinks'' the problem is difficult or not. This is important because we need to understand after SGD paying the price of the additional log factors, the level of complexity of the problem does not change. Otherwise, I can argue that the benefit comes from the change of level of complexity rather than the implicit regularization. Note that for many problems like matrix completion, adding an additional log factor to the sample size, the problem becomes from unsolvable to solvable by many algorithms. The reason behind it is that the additional log factor changes the difficulty of the problem significantly and is not because of some unique property of a certain algorithm.
> > > > > >
> > > > > > If the authors want to justify that $k^*$ is the right dimension to look at. Then the authors should justify why $k^*$ captures the complexity of the linear problem not from the perspective of SGD. Why $k^*$ captures the difficulty of the linear system more accurately than the quantity I propose? In addition, authors need to discuss in depth why the complexity of the problem does not change after the additional log factor.
> > > > > >
> > > > > > I agree and understand that the specific $w^*$ provided by the author is just an easy example, but as I have mentioned, as long as $w^*$ is random and independent of X (which is a common case/assumption), I consider $\frac{(\sum_{i=1}^d\lambda_i)^2}{\sum_{i=1}^d\lambda_i^2}$ the right quantity to look at and for this case, I am not convinced by the messages in the paper (about SGD is not worse than Ridge due to its implicit regularization).  Thus to suit the messages in the paper, $w^*$ has to have some structure. Then before we evaluate the message, we need to ask ourselves why such $w^*$ needs ours attentions.

---

> > > > > > > ### Author Response · Authors · 2021-08-29
> > > > > > > **Title: Thanks for your comments**
> > > > > > >
> > > > > > > **Q1:** “The reason I look at this dimension is to evaluate the level of complexity of the problem (as I mentioned previously) and it is not fair to compare two algorithms when they face problems with different levels of complexity. “   “Since it is the complexity of the problem (linear system in this case), the dimension should depend on the linear problem itself NOT the algorithm nor the bounds in the paper. .... Otherwise, I can argue that the benefit comes from the change of level of complexity rather than the implicit regularization. ”
> > > > > > >
> > > > > > >
> > > > > > > **A1:** Remember that our results are about instance-wise risk comparison. In other words, it is about the comparison of SGD and ridge regression for the *same* problem instance. In this case, it really doesn't matter what dimension is used, as long as the upper and lower risk bounds of SGD and ridge regression are tight. In our paper, the upper bound and lower bound we use for SGD and ridge regression are indeed tight (matching upper and lower bounds proved in our paper or previous work). So your comments on the complexity of the problem are really irrelevant or at least not essential at all here.
> > > > > > >
> > > > > > > ---
> > > > > > >
> > > > > > > **Q2**: “Otherwise, I can claim OLS is the best algorithm based on the comparison that OLS achieves the minimal risk given infinite sample size while all other algorithms can not when they are given zero sample size. Of course, this is ridiculous because I looked at different levels of complexity of the problems for different algorithms (extremely easy for OLS and impossible for other algorithms).”
> > > > > > >
> > > > > > > **A2:** What we’re doing in this paper is the so-called *sample-inflation* comparison, i.e., in order to achieve the same order of risk, what is the *smallest possible* sample-inflation SGD (or Ridge regression) needs to have. Back to your argument, if you choose OLS with infinite sample size, and other algorithms with zero sample size, this is a trivial and meaningless *sample-inflation* comparison, because (1) the risk of the OLS and the risk of other algorithms are not in the same order (0 vs. 1); and (2) the sample-inflation is infinite. So we don’t see how your argument here can refute our results.
> > > > > > >
> > > > > > > ---
> > > > > > >
> > > > > > > **Q3:** “Note that for many problems like matrix completion, adding an additional log factor to the sample size, the problem becomes from unsolvable to solvable by many algorithms. The reason behind it is that the additional log factor changes the difficulty of the problem significantly and is not because of some unique property of a certain algorithm.”
> > > > > > >
> > > > > > > **A3:** This is an inappropriate analogy. As you acknowledged, the log factor will cause a phase transition in matrix completion (statistically recoverable vs. statistically unrecoverable). In sharp contrast, in our results, we focus on the problem instances for which both ridge regression and SGD can generalize (this is true if $w^*$ has bounded $\ell_2$ norm), and we can show the logarithmic sample inflation for SGD. So the logarithmic factor in matrix completion and the logarithmic sample inflation in our results are fundamentally different.
> > > > > > >
> > > > > > > ---
> > > > > > >
> > > > > > > **Q4:** “If the authors want to justify that k∗is the right dimension to look at. Then the authors should justify why  $k^*$ captures the complexity of the linear problem not from the perspective of SGD. Why $k^*$ captures the difficulty of the linear system more accurately than the quantity I propose? In addition, authors need to discuss in depth why the complexity of the problem does not change after the additional log factor.”
> > > > > > >
> > > > > > > **A4:** As we mentioned in **A1:**, our result is an instance-wise risk comparison between SGD and ridge regression, and what really matters is to derive the tightest possible upper and lower bounds for SGD and ridge regression, based on which we can derive the sample-inflation results.  Indeed, based on $k^*$, we can establish matching upper and lower bounds for both SGD and ridge regression, which has already justified the use of $k^*$ in our results. If you insist the effective rank you proposed is better than $k^*$, could you use it to derive matching upper and lower bounds for SGD and/or ridge regression?
> > > > > > >
> > > > > > > ---
> > > > > > >
> > > > > > > **Q5:** “I agree and understand that the specific $w^*$ provided by the author is just an easy example, but as I have mentioned, as long as  $w^*$ is random and independent of $X$ (which is a common case/assumption), I consider ... I am not convinced by the messages in the paper (about SGD is not worse than Ridge due to its implicit regularization). Thus to suit the messages in the paper, $w^*$ has to have some structure. Then before we evaluate the message, we need to ask ourselves why such $w^*$ needs ours attentions.”
> > > > > > >
> > > > > > > **A5:** Remember that our whole paper is about instance-wise comparison, and $w^*$ is part of the problem instance, so we cannot ignore $w^*$. In fact, for one particular problem instance, $w^*$ will at least appear in the bias error, when you do variance-bias decomposition. Thus, the difficulty of one problem instance not only depends on the covariance matrix $H$, but also depends on $w^*$. This is also true even when $w^*$ is independent of $X$, though in this case the structure of $w^*$ is not important but its norm still plays an important role in the excess risk bound.
> > > > > > >
> > > > > > > ---
> > > > > > >
> > > > > > > Let us know if there is still anything unclear.

---

> > > > > > > > ### Comment · Reviewer_P3AS · 2021-08-29
> > > > > > > > **Thanks for the response**
> > > > > > > >
> > > > > > > > I think our main disagreement is on how to evaluate the level of complexity/effective dimension/effective rank. I emphasize that the level of complexity should depend on the linear system itself and should not depend on the special algorithms or derived bounds while the authors argue that the complexity should depend on the algorithm and bounds. Thanks for all the clarifications and efforts, but at this point, I do not think we will reach an agreement on the topic. As explained in the OLS example, I insist that without showing the level of complexity of the problem itself are the same for the two algorithms, we can make a trivial and ridiculous statement for this sample-inflation comparison. Hence, the goal of the example is to make the example trivial. In addition,  as explained in my second example of matrix completion, when the complexity changes, it can be the case (which is most likely the case in my opinion) that the benefit comes from the change of the complexity of the problem, not from some property of the algorithm.
> > > > > > > >
> > > > > > > > Some supporting material about my proposed quantity can be found in Bartlett's paper [6] as $R_0(\Sigma)$. The effective rank or $k^*$ defined in [6] are all in order of $n$ for $\lambda_i=\frac{1}{i^{\alpha}}$ with $\alpha<0.5$. In fact, if we just look at $k^*$ in [6], it is in order of $n$ for all $\alpha<1$ which suggests that Ridge considers the problem with the same level of complexity for all $\alpha<1$. But again, I think it is only fair to look at the complexity of the problem itself which I think $R_0(\Sigma)$ is more appropriate.
> > > > > > > >
> > > > > > > > Based on this, for many cases (I mentioned in previous comments and below), I am not convinced about the benefit of implicit regularization of SGD which is the main selling point of the paper.
> > > > > > > >
> > > > > > > >
> > > > > > > > For the second point about $w^*$, I understand it is an instance-wise comparison case, but my point is that the claim about the benefit from implicit regularization made in the paper is not convincing for a normal $w^*$. To look at other more special $w^*$, we need to justify why we want to look at it. If the claim only works for some very special $w^*$, the significance of the claim will be low and it is very unlikely to generalize to non-linear problems that we care about more. But I do realize that my point about normal $w^*$ depends on my measurement of the level of complexity and since we do not agree on that, it is also pointless to argue further on the second point.
> > > > > > > >
> > > > > > > > Thanks for the clarification and helpful discussions, but because of the main disagreement, I am not convinced about the benefit of implicit regularization of SGD.

---

> > > > > > > > > ### Author Response · Authors · 2021-08-30
> > > > > > > > > **Thank you for your reply!**
> > > > > > > > >
> > > > > > > > > We really appreciate your engagement in this discussion. Now that you have pinned down the disagreement to a very concrete issue in your last reply, we do see the hope that we can resolve this disagreement. Let’s explain it as follows.
> > > > > > > > >
> > > > > > > > > ---
> > > > > > > > >
> > > > > > > > > **Q1:** I think our main disagreement is on how to evaluate the level of complexity/effective dimension/effective rank …  it can be the case (which is most likely the case in my opinion) that the benefit comes from the change of the complexity of the problem, not from some property of the algorithm.
> > > > > > > > >
> > > > > > > > > **A1:** We would like to emphasize that the comparison is about the risk, not the complexity, so the comparison is valid and meaningful as long as the risk bounds for SGD and ridge regression are really tight, for the same problem instance. In order to derive such tight risk bounds, then we do believe the complexity should depend on both the problem and algorithms (not the risk bounds, the risk bounds rely on the complexity, but not the reverse).
> > > > > > > > >
> > > > > > > > > It is true that the **infinite sample inflation** is trivial and ridiculous, but our sample inflation result does not suffer from this issue. Moreover, as we mentioned previously, your argument regarding the matrix completion problem is not appropriate for our case, as we only focus on the problem instance that is solvable by both ridge regression and SGD (It is not like it is solvable by SGD, but not by ridge regression).
> > > > > > > > >
> > > > > > > > > ---
> > > > > > > > > **Q2:** Some supporting material about my proposed quantity can be found in Bartlett's paper [6] as $R_0(\Sigma)$. The effective rank or $k^*$ defined in [6] are all in order of $n$ for $\lambda_i=\frac{1}{i^{\alpha}}$ with $\alpha<0.5$. In fact, if we just look at $k^*$ in [6], it is in order of $n$ for all $\alpha<1$ which suggests that Ridge considers the problem with the same level of complexity for all $\alpha<1$. But again, I think it is only fair to look at the complexity of the problem itself which I think $R_0(\Sigma)$ is more appropriate.
> > > > > > > > >
> > > > > > > > > **A2:** We are glad that you are bringing up this paper [6] by Bartlett et al. so that we can have a more in-depth technical discussion.  It is true that for OLS, the effective rank and the quantity $k^*$ (Theorem 4) in [6] are in the order of $n$. But just as you mentioned, the $k^*$ is only the effective dimension for the OLS solution, but not for other algorithms. Besides, since OLS (the minimum-norm solution) does not have another hyperparameter (e.g., learning rate or regularization parameter), that’s why $k^*$ only depends on $H$ and $n$. This is not contradictory to our claim that the complexity should depend on both the problem instance and algorithms.
> > > > > > > > >
> > > > > > > > > In fact, a subset of the same authors (Bartlett et al.) have a follow-up work for ridge regression [26], and in that paper, they use a different effective dimension as follows (our paper also uses this effective dimension for ridge regression):
> > > > > > > > > $$
> > > > > > > > > k^* = \min \bigg\\{k>0: \frac{\sum_{i>k}\lambda_i +\lambda}{\lambda_{k+1}} \ge bn\bigg\\}.
> > > > > > > > > $$
> > > > > > > > > Then for ridge regression with different regularization parameters, this $k^*$ will be different (note that $k^*$ for OLS is a special case that $\lambda=0$). In fact, if the regularization parameter of ridge regression is well-tuned, then it is highly possible that $k^*=o(n)$ ($k^*$ decreases as $\lambda$ increases). This is why ridge regression (with tuned regularization) can outperform OLS.
> > > > > > > > >
> > > > > > > > > If you really insist on figuring out the complexity of a linear problem (which is independent of the algorithm), one should consider the optimal algorithm, i.e., the algorithm learns such a linear problem with the lowest price. OLS, ridge regression, and SGD are just some particular algorithms, and the $k^*$ for these algorithms can be actually regarded as **upper bound** of the *true complexity* of this linear problem.  Then according to our previous reasoning that $k^*$ of OLS is larger than $k^*$ of ridge regression, it is clear that the *true complexity* of such a linear problem should be at least smaller than the $k^*$ for ridge regression (in contrast to the effective rank pointed out by you or $k^*$ for OLS).
> > > > > > > > >
> > > > > > > > > ---
> > > > > > > > >
> > > > > > > > > **Q3:** For the second point about $w^*$ ... But I do realize that my point about normal $w^*$ depends on my measurement of the level of complexity and since we do not agree on that, it is also pointless to argue further on the second point.
> > > > > > > > >
> > > > > > > > > **A3:** Let’s discuss it more once we agree on the **A1** and **A2**.

---

> > > > > > > > > > ### Comment · Reviewer_P3AS · 2021-08-31
> > > > > > > > > > **Thanks for the response**
> > > > > > > > > >
> > > > > > > > > > First of all, I still consider that level of complexity should depend on the problem instead of a particular algorithm. It is true that my examples of OLS or matrix completion are not the exact case for SGD and Ridge. However, the logic behind the examples still holds. Specifically, without showing the complexity of the problem does not change, we can not draw the conclusion that the benefit comes from some property of an algorithm, it can come from the change of complexity. It is even harder to convince me that the claim can be generalized to non-linear problems which I think is the main motivation to look at this linear problem in the first place.
> > > > > > > > > >
> > > > > > > > > > About k^* in the follow-up paper, BENIGN OVERFITTING IN RIDGE REGRESSION, first of all, I want to emphasize that I am against using $k$ as a measure. For the sake of friendly discussion, let us focus on $k$. Note that in both [6] and Theorem 7 in their follow-up paper (link below), $k$ is defined by comparing between $n\lambda_{k+1}$ and $\sum_{i>k}\lambda_i$. Thus it does not depend on regularizer $\lambda$. If the authors are referring to Theorem 5 about when upper bound matches lower bound. I agree that $k$ depends on $\lambda$ and optimal $k$ is considered to let $\rho_k$ being constant. However, to have k=o(n), we require $\lambda = \Omega(n\lambda_{k+1}) >> n^{1-\alpha}$ when $\lambda_i=\frac{1}{i^{\alpha}}$ with $\alpha<1$. I think this is not the optimal $\lambda$ for a random true parameter with d>n, and the optimal $\lambda$ for the common cases should be $O(n^{1-\alpha})$. For optimal $\lambda$, we have $\sum_{i>k}\lambda_i = \Omega(\lambda)$ and we are still comparing between $\sum_{i>k}\lambda_i$ and $n\lambda_{k+1}$.
> > > > > > > > > >
> > > > > > > > > > Again, the above is for a friendly discussion and my trying to walk in the authors' shoes. From my own perspective, I do not think the effective rank should depend on an algorithm. I agree with the following concept of $k$ that mentioned in the follow-up paper that a good $k$ is to measure the spiky part of the eigenvalues of H. I think $r_k(\Sigma)$ or $R_k(\Sigma)$ are serving this purpose.
> > > > > > > > > >
> > > > > > > > > > the follow-up paper: https://arxiv.org/pdf/2009.14286.pdf

---

> > > > > > > > > > > ### Author Response · Authors · 2021-09-01
> > > > > > > > > > > **Thanks for the response**
> > > > > > > > > > >
> > > > > > > > > > > It looks that there is still disagreement about the complexity used in our paper. Yet we don’t want to argue it anymore because we believe we have already made it crystal clear in our previous responses why the complexity does not matter in our results, as long as the upper/lower bounds of SGD and ridge regression are tight.
> > > > > > > > > > >
> > > > > > > > > > > With that being said, we have to point out a mistake in your latest comment regarding the optimal $\lambda$ and clarify it as follows.
> > > > > > > > > > >
> > > > > > > > > > > **Q:** “...to have k=o(n), we require $\lambda = \Omega(n\lambda_{k+1}) >> n^{1-\alpha}$ when $\lambda_i=\frac{1}{i^{\alpha}}$ with $\alpha<1$. I think this is not the optimal $\lambda$ for a random true parameter with d>n, and the optimal $\lambda$ for the common cases should be $O(n^{1-\alpha})$. For optimal $\lambda$, we have $\sum_{i>k}\lambda_i = \Omega(\lambda)$ and we are still comparing between $\sum_{i>k}\lambda_i$ and $n\lambda_{k+1}$”.
> > > > > > > > > > >
> > > > > > > > > > > **A:** We do not agree that the optimal ridge regression parameter is $\lambda = \sum_{i>k^*}lambda_i$, otherwise ridge regression will have no advantage compared to OLS. In addition, your comment that “the optimal $\lambda$ for the common cases should be $O(n^{1-\alpha})$” is incorrect.
> > > > > > > > > > >
> > > > > > > > > > > In detail, given the polynomial decaying spectrum $\lambda_i = i^{-\alpha}$ for $0<\alpha<1$ and bounded $w^*$ (i.e., $\\|w^*\\|_2=O(1)$), then by Theorem 1 and Theorem 5 in [26], the following bound holds
> > > > > > > > > > >
> > > > > > > > > > > $$
> > > > > > > > > > >  Excess risk \lesssim \frac{\tilde \lambda}{N} + \bigg(\frac{N}{\tilde\lambda}\bigg)^{1/\alpha}/N + \frac{N^2}{\tilde\lambda^2}\cdot N^{-2\alpha}.
> > > > > > > > > > > $$
> > > > > > > > > > >
> > > > > > > > > > > Thus, if $\alpha<1/2$, the last term will dominate the second one, then we only need to  find $\lambda$ that minimizes
> > > > > > > > > > >
> > > > > > > > > > > $$
> > > > > > > > > > >  \frac{\tilde \lambda}{N} + \frac{N^2}{\tilde\lambda^2}\cdot N^{-2\alpha},
> > > > > > > > > > > $$
> > > > > > > > > > >
> > > > > > > > > > > which gives $\lambda \eqsim \tilde\lambda \eqsim N^{1-2\alpha/3}$.
> > > > > > > > > > >
> > > > > > > > > > > If $1/2 <\alpha<1$, then the second  term will dominate the last one, the optimal $\lambda$ should minimize
> > > > > > > > > > >
> > > > > > > > > > > $$
> > > > > > > > > > >  \frac{\tilde \lambda}{N} + \bigg(\frac{N}{\tilde\lambda}\bigg)^{1/\alpha}/N,
> > > > > > > > > > > $$
> > > > > > > > > > >
> > > > > > > > > > > which gives $ \lambda \eqsim \tilde\lambda = N^{1/(1+\alpha)}$.
> > > > > > > > > > > Combining the above two cases, for $0<\alpha<1$, it is clear that the optimal $\lambda$ is larger than $N^{1-\alpha}$. Thus, your comment that “the optimal $\lambda$ for the common cases should be $O(n^{1-\alpha})$” is incorrect.

---

> > > > > > > > > > > > ### Comment · Reviewer_P3AS · 2021-09-01
> > > > > > > > > > > > **Thanks for the response**
> > > > > > > > > > > >
> > > > > > > > > > > > I agree that we won't reach an agreement on the level of complexty.
> > > > > > > > > > > >
> > > > > > > > > > > > For the optimal $\lambda$, I do not agree with authors calculation. First, when $\theta^*$ is random and $k<<n$, the bias term which contains $\|\theta^*_{k:\infty}\|_{\Sigma_{k:\infty}}^2$ will be always at least in order of $\frac{n^{1-\alpha}}{n}||\theta^*||_2^2$ and dominates $\frac{\tilde{\lambda}}{n}||\theta^*||_2^2$. Hence, the authors optimization over $\lambda$ is not correct. The best solution is to increase $\lambda$ large enough such that Variance is at most the order of Bias and I agree it seems that $\lambda>> n^{1-\alpha}$ when we assume $||\theta^*||_2^2=1$ and $\sigma^2=1$. But I want to point out that this is not the right way to scale the true parameter and noise level. In fact, we can find that the signal to noise ratio is  $\frac{n^{1-\alpha}}{n}||\theta^*||_2^2/\sigma^2$ which goes to 0 if $||\theta^*||_2^2=1$ and $\sigma^2=1$. It does make sense that for large enough noise we should choose much larger regularization but that also makes the problem less interesting. However, for a constant signal to noise ratio, we can find that $\lambda=\Theta(n^{1-\alpha})$ and $k=\Theta(n)$ achieves the same order of risk which is $\frac{n^{1-\alpha}}{n}||\theta^*||_2^2$. Hence, the optimal lambda should be tuned to achieve the best constant omitted in the bounds rather than the order.
> > > > > > > > > > > >
> > > > > > > > > > > > Another way to look at the proper scaling of $\lambda$ is to look at the estimate which is $\hat{\theta}=(X^TX+\lambda)^{-1}X^TX\theta^*$ assuming $y=X\theta^*$ for simplicity. Note that when $d=\Theta(n)$,  $\frac{n^{\alpha}}{n}X^TX$ has benign spectrum because $\frac{n^{\alpha}}{i^{\alpha}}$ weakly converge to a distribution. Hence, let $S=\frac{n^{\alpha}}{n}X^TX$ and we can rewrite the estimate as $\hat{\theta}=(S+\frac{\lambda}{n^{1-\alpha}})^{-1}S\theta^*$. Hence, to achieve a proper estimate, the scaling of $\lambda$ should be in order of $n^{1-\alpha}$ otherwise, if $\lambda>>n^{1-\alpha}$, $||\hat{\theta}||_2^2/||\theta^*||_2^2\rightarrow 0$ which is only optimal when signal to noise goes to 0.

---

> > > > > > > > > > > > > ### Author Response · Authors · 2021-09-01
> > > > > > > > > > > > > **Thanks for your further response**
> > > > > > > > > > > > >
> > > > > > > > > > > > > Thanks for your reply!
> > > > > > > > > > > > >
> > > > > > > > > > > > > We would like to point out yet another mistake in your latest comment (See below).
> > > > > > > > > > > > >
> > > > > > > > > > > > > ---
> > > > > > > > > > > > >
> > > > > > > > > > > > > **Q1:** “when $w^*$ is random and $k^* << N$, the bias term which contains $\\|w^*_{k^*:\\infty}\\|\_{H\_{k^*:\\infty}}^2$ will be always at least in order of  $\\frac{N\^{1-\\alpha}}{N}\\|w^\*\\|\_2\^2$ and dominates $\\frac{\\tilde \\lambda}{N}\\|w^\*\\|\_2\^2$.”
> > > > > > > > > > > > >
> > > > > > > > > > > > > **A1:** Your comment that “$\\frac{N\^{1-\\alpha}}{N}\\|w^\*\\|\_2\^2$ and dominates $\\frac{\\tilde \\lambda}{N}\\|w^\*\\|\_2\^2$ is wrong.  According to the definition of $k\^*$, it is clear that for any $\\tilde \\lambda$, we have
> > > > > > > > > > > > >
> > > > > > > > > > > > > $$
> > > > > > > > > > > > > \\|w^*_{k^*:\\infty}\\|\_{H\_{k^*:\\infty}}^2\\le\\lambda\_{k^*} \cdot \\|w^*_{k^*:\\infty}\\|\_2^2\\le \\frac{\\tilde \\lambda}{N}\\|w^\*\\|\_2\^2
> > > > > > > > > > > > > $$
> > > > > > > > > > > > > In other words, no matter how $\tilde \lambda$ is chosen or optimized, we always have $\\frac{\\tilde \\lambda}{N}\\|w^\*\\|\_2\^2$ be larger than $\\|w^*_{k^*:\\infty}\\|\_{H\_{k^*:\\infty}}^2$, not the other way around.
> > > > > > > > > > > > >
> > > > > > > > > > > > > ---
> > > > > > > > > > > > >
> > > > > > > > > > > > > **Q2:**: “I want to point out that this is not the right way to scale the true parameter and noise level. In fact, we can find that the signal to noise ratio is $\\frac{N\^{1-\\alpha}}{N}\\|w^\*\\|\_2\^2 / \\sigma\^2$ which goes to 0 if $\\|w^\*\\|\_2\^2 =1$ and $\\sigma\^2 = 1$.”
> > > > > > > > > > > > >
> > > > > > > > > > > > > **A2:** We would like to emphasize that the signal-to-noise ratio in our paper is $\\|w
> > > > > > > > > > > > > \^*\\|\_{H} / \\sigma\^2$, which is clearly stated in line 211. We wonder how did you come up with the signal-to-noise ratio in the form of $\\frac{N\^{1-\\alpha}}{N}\\|w^\*\\|\_2\^2 / \\sigma\^2$? The signal-to-noise ratio in our paper follows the standard definition [1] for a noisy model $y = s + n$ (with $s$ and $n$ being signal and noise respectively), which is defined as $E[s\^2]/E[n\^2]$. Note that in our model $s = \langle w^\*, x\rangle$, then it is clear that $E[s\^2]/E[n\^2] = E[ (\langle w^\*, x\rangle)\^2] / \\sigma^2 = \\|w^\*\\|\_H\^2/\sigma\^2$, which may not go to zero as you mentioned.
> > > > > > > > > > > > >
> > > > > > > > > > > > >
> > > > > > > > > > > > >
> > > > > > > > > > > > > [1] https://en.wikipedia.org/wiki/Signal-to-noise_ratio

---

> > > > > > > > > > > > > > ### Comment · Reviewer_P3AS · 2021-09-01
> > > > > > > > > > > > > > **Thanks for the response**
> > > > > > > > > > > > > >
> > > > > > > > > > > > > > To clarify, all my calculation is based on simple and standard assumption that $w_i^*$ or $\theta^*_i$ are i.i.d. Gaussian random variables.
> > > > > > > > > > > > > >
> > > > > > > > > > > > > > I agree that the signal to noise is $||w^*||_{H}^2/\sigma^2$. Then with the random assumption given above, with high probability, the signal to noise ratio is in order of its expectation which is $E(w_i^*)^2Tr(H) /\sigma^2 \approx \frac{\\|w^*\\|_2^2}{d}d^{1-\alpha} /\sigma^2$ for $\lambda_i=\frac{1}{i^{\alpha}}$.
> > > > > > > > > > > > > >
> > > > > > > > > > > > > > In previous reply, the authors assume $||w^*||_2^2=1$ (not $||w^*||_H^2=1$). Then for a constant signal to noise ratio, we need $\sigma^2\approx \frac{d^{1-\alpha}}{d}\approx N^{-\alpha}$. For $\sigma^2=1$, the signal to noise ratio is going to be 0.
> > > > > > > > > > > > > >
> > > > > > > > > > > > > > Back to the Bias Variance upper bound. I apologize that when the authors want $\lambda>> N^{1-\alpha}$, the dominant term of bias is in fact $\frac{\lambda}{N}||w^*||_2^2 \approx \frac{\lambda}{N}$ (I was using my assumption that $\lambda=O(N^{1-\alpha})$, I apologize). However, the variance term which is $\sigma^2 \bigg( \bigg(\frac{N}{\tilde\lambda}\bigg)^{1/\alpha}/N + \frac{N^2}{\tilde\lambda^2}\cdot N^{-2\alpha}\bigg)$ and $\sigma^2\approx N^{-\alpha}$ for a constant signal to noise ratio. Hence, the authors have missed this $N^{-\alpha}$ in their optimization. With the correct Variance, the authors should find that $\lambda=\Theta(N^{1-\alpha})$ is the optimal choice.
> > > > > > > > > > > > > >
> > > > > > > > > > > > > > Again, the above calculation is based on a common $w^*$ with no structure assumption, and we analyze an average case. If the claim does not hold on the standard cases, it is hard to believe it can be generalized.

---

> > > > > > > > > > > > > > > ### Author Response · Authors · 2021-09-01
> > > > > > > > > > > > > > > **Thanks for the reply**
> > > > > > > > > > > > > > >
> > > > > > > > > > > > > > > Thanks for the clarification.
> > > > > > > > > > > > > > >
> > > > > > > > > > > > > > >
> > > > > > > > > > > > > > > We would like to clarify that the example where $\\| w^\*\\|_2 \eqsim 1$ and $\sigma^2 \eqsim 1$ is introduced to explain the calculation of $k^*$ and $\lambda$. It is possible that the SNR will go to zero for certain $w^\*$ (e.g., random $w^\*$). We also want to emphasize that there are also many problem instances that have structured $w^\*$ such that $\\|w^*\\|_2\eqsim 1$, $\\|w^\*\\|_H\eqsim 1$ and  $\sigma^2 \eqsim 1$ (which will have a constant SNR).
> > > > > > > > > > > > > > >
> > > > > > > > > > > > > > >
> > > > > > > > > > > > > > > More importantly, we would like to point out that the example you pointed: $w^*$ is random, $\\|w^*\\|_2=1$, $\lambda_i = i^{-\alpha}$, $\sigma^2 = N^{-\alpha}$ can be covered by our Corollary 5.1. Our Corollary 5.1 is actually a worst-case sample inflation bound for *any* problem instance with $\lambda_i=i^{-\alpha}$ and constant SNR, which is a larger class of problem instances that contains your example with random $w^\*$. In fact, for your example, we can derive an even stronger result for sample inflation comparison between SGD and ridge regression. More specifically, setting $N\_{sgd} \ge N\_{ridge}$ is sufficient for $risk\_{sgd} \lesssim risk\_{ridge}$. To see this, note that $k^\* \eqsim N\_{ridge}$ and the optimal parameter is $\lambda = N^{1-\alpha} \eqsim Tr(H)$ for your example. For SGD, by Lemma 6.1, we can simply set the stepsize to be $\gamma = 1/\lambda = N^{\alpha-1} \eqsim 1/{Tr(H)}$ and $k_1= k_2 \eqsim k^\*$, then Lemma 6.1 implies that the SGD risk matches that of ridge regression. We will add a discussion on this example in the final version.

---

### Official Review · Reviewer_zkCv · 2021-07-15

**Rating:** 5
**Confidence:** 3

**Summary:**

Prior work showed that SGD has implicit regularization. This paper comperes the generalization performance of the implicit regularization (namely SGD without regularization) versus $ \ell_2 $ regularization (i.e. ridge regression), which is a popular explicit regularization technique. The generalization performance is measured in excess risk. Using bounds on the excess risk, the authors show that SGD without regularization generalize at least as ridge regression, if not better.

**Limitations And Societal Impact:**

See above

**Main Review:**

The article is well written and well organized.
Here are my comments:

1.	The comparison made in the paper between single pass SGD and ridge regression is not impartial. Specifically, single pass SGD sees each sample only once and does not minimize the empirical risk. In contrast, for ridge regression, the authors considered the unique minimizer of the regularized risk. This equivalent to multi pass SGD with explicit regularization. In other words, for unregularized objective they use single pass SGD, and for regularized objective they use multi pass SGD. Why?
2.	The implicit regularization of multi pass SGD for overparametrized linear regression is well understood. In this case, the obtained solution minimizes the Euclidean distance from initialization (this also cited by the authors). Considering my first remark, I think that the article could have been more convincing if they show advantages of implicit regularization over explicit in the simple (and well understood) case of multi pass SGD.
3.	It is quite popular in statistical learning to use bounds that hold for some absolute coefficient. However, I am not sure to what extent these bounds meaningful.

I note that due to time limitations, I have not checked the proofs.


**Time Spent Reviewing:**

6 Hours

---

> ### Author Response · Authors · 2021-08-10
> **Response to Reviewer zkCv**
>
> Thanks for your comments.
>
> ---
> **Q:** Comparison between single-pass SGD for unregularized objective and multi-pass SGD for regularized objective (Re: comments 1 and 2)
>
> **A:**  It looks that the reviewer has misunderstood the key question we want to answer in this paper. We emphasize that our aim is to investigate the effect of implicit bias of one-pass SGD (instead of multi-pass SGD!), so it is natural to compare the performance of one-pass SGD with the ridge solutions as we are making a comparison between the implicit regularization of one-pass SGD with the explicit regularization of ridge regularization. Besides, since we can already get the closed-form solution of ridge regression, it is not necessary to use multi-pass SGD for solving ridge regression. We will emphasize the central goal of our paper in the revision.
>
> ---
>
> **Q:** “It is quite popular in statistical learning to use bounds that hold for some absolute coefficient. However, I am not sure to what extent these bounds are meaningful.”
>
> **A:** As you have acknowledged, it is indeed a common practice of statistical learning to use bounds that hold for some absolute coefficient. Note that as long as the solutions are generalizable (Definition 1), the effect of absolute constants can be safely neglected provided a sufficiently large sample size $N$. This is indeed the case of modern machine learning applications, where the sample size is huge. Therefore, we think a rate comparison can well capture the fundamental behaviors of different algorithms.
>
> ---
>
> We hope our replies would address your concerns.

---

> > ### Comment · Reviewer_zkCv · 2021-08-31
> > **Reply**
> >
> > As I understand it, this article is about the benefits of implicit regularization over explicit, as its title says. However, the existence of explicit regularization is not the only difference between the two cases studied in it (regularized and unregularized). Specifically, in the unregularized case, the objective function is not considered to be minimized. This is in contrast to the regularized setting where the authors studied the unique minimizer of the loss function. Note that stopping SGD before convergence, i.e. early stopping, is a well known regularization technique. Therefore, it is not clear to me why the authors attribute the benefits to the implicit regularization of SGD, instead of early stopping.
> > To be clearer, assuming limited number of iterations, if the authors claim that implicit regularization is better than explicit $ \ell_2 $ regularization, then they should show it using **single pass SGD with $ \ell_2 $ regularization, and not via ridge regression solution**.

---

> > > ### Author Response · Authors · 2021-09-01
> > > **Thanks for your reply**
> > >
> > > Thank you for raising those additional questions. We will answer them as follows.
> > >
> > > **Q1:** “As I understand it, this article is about the benefits of implicit regularization over explicit, as its title says. .... Note that stopping SGD before convergence, i.e. early stopping, is a well-known regularization technique. Therefore, it is not clear to me why the authors attribute the benefits to the implicit regularization of SGD, instead of early stopping.”
> > >
> > > **A1:** We view early stopping as a kind of implicit regularization (as has been demonstrated in many prior works). We did not try to highlight early stopping since it is part of constant-stepsize online SGD with iterate averaging studied in our paper.
> > >
> > > ---
> > >
> > > **Q2:** “To be clearer, assuming a limited number of iterations, if the authors claim that implicit regularization is better than explicit $ \ell_2 $ regularization, then they should show it using **single pass SGD with $ \ell_2 $ regularization, and not via ridge regression solution**.”
> > >
> > > **A2:** If you study single pass SGD with $\ell_2$ regularization, then the regularization effect could be a mixture of “implicit regularization” by single-pass SGD and the “explicit regularization” by $\ell_2$ regularization. Then comparison would become implicit regularization vs. implicit+explicit regularizations, which is not the focus of our paper.

---

### Official Review · Reviewer_dmQy · 2021-07-16

**Rating:** 7
**Confidence:** 3

**Summary:**

This paper compares the excess risk of averaged SGD with that of ridge regression. The comparison is in the setting where the expected noise is zero and also the excess risks of the optimally tuned algorithms are smaller than the optimal population risk. The overall conclusion is that the performance of SGD is close to ridge regression in every problem in the class of interest and can even be better than ridge regression based on the eigenspectrum of the input and the signal-to-noise ratio.

**Limitations And Societal Impact:**

Limitations are addressed. Societal Impact is not applicable.

**Main Review:**

I'm voting to accept the submission as these results can be a step towards understanding optimization and generalization of overparameterized models with SGD. The reliance of the results on signal-to-noise ratio also matches the previous observation that the behavior of SGD can be vastly different on real data versus random labels [28]. The main weakness is that, although the paper mentions that SGD requires at most logarithmically more samples, the constants in the bound can be huge.

In the instance-wise comparison (Theorem 5.1), the difference between SGD and ridge regression depends on the inverse of the smallest eigenvalue of the input. If the data is not well-conditioned, this factor can easily dominate the bound. The paper then correctly mentions that the constant cannot be removed, but still, with a large multiplicative factor, given the current bound, SGD can vastly underperform ridge regression. Do the authors have any empirical evidence that the smallest eigenvalue is large enough in common datasets?

The paper motivates its results as a study of implicit regularization and generalization of SGD, yet as l215 clarifies, there is an interplay of optimization and generalization at work, and the convergence rate and generalization ability together determine the risk. I suggest that the authors mention the role of convergence rate in the introduction and in other parts of the paper, for example in l256 that describes when SGD "generalizes better" than ridge regression.

Minor comments:

The Gaussian assumption and Assumption 5.1 are made for simplicity yet it is not clear to me how relaxing these assumptions would change the results and if extra factors would be introduced.

What is lambda in Theorem 5.1. Does the result hold for any lambda?

**Time Spent Reviewing:**

10

---

> ### Author Response · Authors · 2021-08-10
> **Response to Reviewer dmQy**
>
> Thank you for acknowledging our contributions as well as the valuable comments and suggestions!
>
> ---
> **Q:** "...the difference between SGD and ridge regression depends on the inverse of the smallest eigenvalue of the input....”
>
> **A:** It is true that for the class of problem instances with very large $\kappa(n)$, the SGD bound in Theorem 5.1 is less interesting. However, we would like to point out that Theorem 5.1 is a worst-case guarantee that allows $w^*$ to be arbitrary. In fact, Theorem 5.3 suggests that the (inverse of) small eigenvalues will only dominate the ``sample inflation’’ if a large fraction of $w^*$ lies in the directions corresponding to the small eigenvalues of $H$. In the statistically interesting setting, $w^*$ should have good properties (e.g., $w^*$ aligns well with the larger eigenvalue directions of $H$), then Theorem 5.3 complements Theorem 5.1, and guarantees that SGD can still be better than ridge regression regardless how small $\lambda_{\min\\{d, n\\}}$ is.
>
> ---
>
> **Q:** Do the authors have any empirical evidence that the smallest eigenvalue is large enough in common datasets?
>
> **A:** This is a good point. But we feel that it would be difficult to empirically evaluate the smallest eigenvalue when it’s small. In particular, $\kappa(n)$ remains in the constant order if $\lambda_{\min\\{d, n\\}} = O(1/n)$. Since we only have $n$ training data, it would be difficult to precisely evaluate this quantity since the estimation error could be as large as $O(1/n^{1/2})$.
>
> ---
>
> **Q:** Generalization and convergence rate
>
> **A:** Thanks for your suggestion! We will highlight it in the revision.
>
> ---
>
> **Q:** “The Gaussian assumption and Assumption 5.1 are made for simplicity yet it is not clear to me how relaxing these assumptions would change the results and if extra factors would be introduced.”
>
> **A:** Our results can be generalized to sub-Gaussian cases as well. Please refer to footnote 1 on page 5. The results only need to be modified up to some absolute constant factors.
>
> ---
>
> **Q:** “What is lambda in Theorem 5.1. Does the result hold for any lambda?”
>
> **A:** $\lambda$ is the regularization parameter of the ridge regression. Yes, Theorem 5.1 holds for any $\lambda$ (both positive and negative). We will emphasize it in the revision.

---

> > ### Comment · Reviewer_dmQy · 2021-08-25
> > **Thank you**
> >
> > Thanks for the response.
> >
> > While it is true in my experience that the top principal components capture most of the information about w*, I suggest adding citations or evidence that this is the statistically interesting setting and examples like line 223 are uncommon in practice.
> >
> > Also after another read I think the motivating example for the generalizable setting (line 155) is weak as the setting in the rest of the paper is infinite dimensional. What is the generalizable setting like when d approaches infinity?

---

> > > ### Author Response · Authors · 2021-08-26
> > > **Response to your additional comments**
> > >
> > > Thank you for your suggestions! We will be sure to add citations/evidence to justify the statistically interesting setting in the final version.
> > >
> > > Regarding the motivating example when $d$ approaches infinity for the generalizable setting, here is an example: let’s consider ridge regression solution in the infinite-dimensional setting, then we can say that the solution $w_{Ridge}(N;\lambda)$ is generalizable on the problem instance with polynomial decay spectrum $\lambda_i = i^{-2}, i=1,2,\ldots$, provided that $N$ is greater than some constant and $\lambda=\Theta(1)$ (according to the upper bound of ridge regression solution in Theorem 1 of [25]). We will also add this motivating example in the final version.

---

### Official Review · Reviewer_uG8V · 2021-07-19

**Rating:** 6
**Confidence:** 4

**Summary:**

The paper considers the problem of understanding implicit regularization of SGD for the problem of linear regression. The authors compare the excess risk guarantees of SGD solution to the one obtained by solving ridge regression exactly. The main contribution is to identify some conditions on the data distribution and the optimal solution under which:

- The SGD solution (with log(n) more samples) is at least as good as the one obtained by ridge regression.
- There exists a problem setting where the excess risk of the ridge regression solution is quadratically worse than the SGD solution.

The above separation result between SGD and ridge regression is obtained by explicitly computing **instance dependent** upper and lower bounds on their respective excess risk guarantees, and by comparing them under various assumptions on the data distributions. The presented results are shown for the setting when the feature vectors are one-hot or Gaussian.


**Main Review:**

I am a bit unsatisfied with the results and do not support acceptance. My reasons are:
(1. ) Nature of the results:
For now, let us restrict to the Gaussian case (Main section of the paper, Section 5). The authors show that there are some conditions (call them A) such that for any learning problem that satisfies A:
        -SGD is almost as good as Ridge regression.
        -There exists a problem setting that satisfies A for which ridge regression is quadratically worse than SGD.

As a result, the authors conclude that SGD has a better implicit regularization than ridge regression. However, this jump is wrong !!!

The results presented in the paper do not rule out the following: There exists some other conditions B (different from A) on the data distribution such that for any learning problem that satisfies B:
        -Ridge regression is almost as good as SGD
        -There exists a problem that satisfies B (but not A) for which SGD is quadratically worse than ridge regression.

B could be as natural / realistic as A, and the results in Theorem 5.1 shows that this may be possible. Thus, there could be some other conditions where the solutions obtained by ridge regression are better (or as good as) that of SGD.

In order to claim that SGD is better than ridge regression, one needs to show that (a) SGD works as well as ridge regression whenever ridge regression works, (b) there exists some problem when ridge regression is strictly worse than SGD.

Note: the experiments shown in Figure 1 in the appendix support the above claim and show that there exist settings where ridge regression requires much less samples than SGD (see the blue line). This shows that the conditions B above may also be easy to characterize.

(2.)  Step size for SGD:
The step size required for SGD to match the performance of ridge regression (as in Corollary 5.1 and Theorem 4.1) depends on the data distribution. It is not clear how to estimate it from data. Thus, it is not clear how to run SGD algorithm in practice to match the rates of ridge regression; limiting the scope of this result.

The choice of step size should not depend on the data distribution. If one has access to the distribution, we can simply run the following trivial algorithm: Given a distribution D, choose the starting point w_0 as the optimal solution for the test loss w.r.t. D. Set the step size to be 0 and run SGD. Here, the final output will be the optimal solution and always better than ridge regression.

(3.) What about other regularizers? One can have regularizers (potentially data dependent) such that the RERM solution with this regularizer generalizes as well as SGD. I think the classical results which compare SGD solutions with ridge regression solutions are for the worst case distributions only.

(4.) While the paper is technically rich, many of the instance dependent upper / lower bounds presented in the paper were known in earlier papers (or at least their proof techniques) (e.g. [29], [6]). Thus, the new technical contributions are limited.

Minor issues / typos:
       - Line 118 has a typo in H_{k: \infty} in the aligned equation.
       - Line 209 has a typi in the definition of a. What is N?
       - The illustrative example from lines 222-228 is circular and uses k(N_{ridge}) in the definition.

Rest of the paper is well written!

**Time Spent Reviewing:**

8-10 hours

---

> ### Author Response · Authors · 2021-08-10
> **Response to Reviewer uG8V**
>
> Thank you for your positive comments!
>
> ---
> **Q:** It is not appropriate to claim that SGD has a better implicit regularization than ridge regression.
>
> **A:** We agree that there exists a class of problem instances such that ridge regression can be better than SGD. For example, consider the problem instances with  $w^*$ that has a large fraction on the small eigen directions of $H$, and with a large condition number $\kappa_n$. For this class of problem instances, ridge regression performs well but SGD will encounter “numerical optimization” issues and performs poorly, according to Theorems 5.1 and 5.3. However such problems are less practical and less interesting in statistical learning (see discussions in line 215-228, and Theorem 5.3). In the revision, we will emphasize that our focus is to compare SGD with ridge regression for those problem instances that are interesting in statistics.
>
> ---
> **Q:** “Figure 1 shows that there exist settings where ridge regression requires much less samples than SGD”
>
> **A:** We disagree with the reviewer’s interpretation of the experimental results shown in Figure 1.  In fact, the blue line does not show that ridge regression is significantly better than SGD, since that curve is nearly linear, i.e.,  there is at most a constant factor gap between the required sample sizes of SGD and ridge regression. This is very consistent with our theory since Theorem 5.1 basically states that SGD could be at most logarithmically worse than ridge regression.
>
> ---
> **Q:** Stepsize for SGD could depend on the data distribution
>
> **A:** Thank you for raising this concern! We would like to point out that the choice of stepsize in our theorems does not require the full information of the data distribution (i.e., the full eigenspectrum of $H$ and $w^*$); In fact, from the proof, it only depends on $tr(H)$ and the regularization parameter $\lambda$ for ridge regression, which is easy to estimate based on samples. Moreover, the stepsize for SGD and the regularization parameter $\lambda$ for ridge regression should be considered as the hyperparameters of the algorithms. Then our theorems (e.g., Theorem 5.1)  read that: for any $\lambda$, there is a stepsize (determined by $\lambda$ and $tr(H)$) for SGD such that SGD can achieve comparable or even better generalization ability than ridge regression with certain sample size. We believe such results clearly illustrate how to run SGD algorithm in practice to match the performance of ridge regression, since the step size can be directly calculated from the training data and ridge regression parameter. We will update the statement of the theorems to emphasize that choosing the stepsize of SGD does not require the full knowledge of the data distribution. Based on the above explanation, the trivial algorithm suggested by you utilizes more information than SGD and cannot refute the implication of our theorems.
>
> ---
> **Q:** Other regularizers
>
> **A:** Good question! The aim of this paper is to show that SGD could have certain implicit regularization by comparing its generalization ability to the solution of ERM with explicit $\ell\_2$ regularization. We agree that other regularizers (e.g., data-dependent regularizers) may give better generalization ability compared to simple $\ell\_2$ regularization ([26] has shown that a weighted $\ell\_2$ regularization could be optimal for over-parameterized linear regression under certain conditions on the data distribution). This is definitely an interesting future work direction since it can reveal more mysteries regarding the implicit regularization of SGD, and we believe our results on the SGD vs. ridge regression can shed light on this direction!
>
> ---
> **Q:** Technical contributions are limited given existing works.
>
> **A:** It is true that the instance dependent bounds in this paper are built upon existing works. However, none of these works provides a “sample inflation” comparison between SGD and ridge regression, which is one of our main contributions. Besides, our paper also provides a nearly complete analysis for the one-hot case, which cannot be covered by the existing works for Gaussian (or sub-gaussian) problems. Moreover, our work provides an expected lower bound result for ridge regression, which does not require the ground-truth $w^*$ to have a coordinate-independent prior distribution [6].

---

### Decision · Program_Chairs · 2021-09-27

**Decision:**

Accept (Poster)

**Comment:**

The paper considers the problem of understanding if SGD can be seen as a solution to implicit regularized empirical risk minimizer for the problem of  linear regression. The authors compare the excess risk guarantees of SGD solution to the one obtained by solving ridge regression exactly. The authors show that SGD solution is always almost as good as the one obtained by ridge regression. But on the other hand, there exists a problem setting where the excess risk of the ridge regression is substantially worse than the SGD solution. This shows that SGD cannot be seen as implicitly regularized ridge regression.

This paper generated a large number of discussions between the reviewers and also between reviewers and authors. While for some points there isnt total agreements, given the reviews and the discussion, it seems clear to me that the paper should be accepted first because despite some of the mentioned possible shortcomings the paper still is interesting and second, the subtler discussion points are exactly why we need to accept the paper so these issues are more widely discussed.